



# Identification and Quantification of $CH_4$ Emissions from Madrid Landfills using Airborne Imaging Spectrometry and Greenhouse Gas Lidar

Sven Krautwurst[1], Christian Fruck[2], Sebastian Wolff[2], Jakob Borchardt[1], Oke Huhs[1], Konstantin Gerilowski[1], Michał Gałkowski[3,4], Christoph Kiemle[2], Mathieu Quatrevalet[2], Martin Wirth[2], Christian Mallaun[5], John P. Burrows[1], Christoph Gerbig[3], Andreas Fix[2], Hartmut Bösch[1], and Heinrich Bovensmann[1]

[1]University of Bremen, Institute of Environmental Physics, Bremen, Germany
[2]Deutsches Zentrum für Luft- und Raumfahrt, Institut für Physik der Atmosphäre, Oberpfaffenhofen, Germany
[3]Department Biogeochemical Signals, Max Planck Institute for Biogechemistry, Jena, Germay
[4]Faculty of Physics and Applied Computer Science, AGH University of Kraków, Kraóow, Poland
[5]Deutsches Zentrum für Luft- und Raumfahrt, Flugexperiemente, Oberpfaffenhofen, Germany

**Correspondence:** Sven Krautwurst (sven.krautwurst@iup.physik.uni-bremen.de)

**Abstract.**

Methane ($CH_4$), alongside carbon dioxide ($CO_2$), is a key driver of anthropogenic climate change. Reducing $CH_4$ is crucial for short-term climate mitigation. Waste-related activities, such as landfills, are a major $CH_4$ source, even in developed countries. Atmospheric concentration measurements using remote sensing offer a powerful way to quantify these emissions. We study waste facilities near Madrid, Spain, where satellite data indicated high $CH_4$ emissions. For the first time, we combine passive imaging (MAMAP2DL) and active lidar (CHARM-F) remote sensing aboard the German research aircraft HALO, supported by in situ instruments, to quantify $CH_4$ emissions. Using the $CH_4$ column data and ECMWF ERA5 model wind information validated by airborne measurements, we estimate landfill emissions through a cross-sectional mass balance approach. Strong emission plumes are traced up to $20\,\mathrm{km}$ downwind on the 4[th] August 2022, with the highest $CH_4$ column anomalies observed over active landfill areas in the vicinity of Madrid, Spain. Total emissions are estimated at $\sim 13\,\mathrm{t\,h^{-1}}$. Single co-located plume crossings from both instruments agree well within $1.2\,\mathrm{t\,h^{-1}}$ (or $13\,\%$). Flux errors range from $\sim 25$ to $40\,\%$, mainly due to boundary layer and wind speed variability. This case study not only showcases the capabilities of applying a simple but fast cross-sectional mass balance approach, as well as its limitations due to challenging atmospheric boundary layer conditions, but also demonstrates the, to our knowledge, first successful use of both active and passive airborne remote sensing to quantify methane emissions from hot spots and independently verify their emissions.

## 1 Introduction

Methane ($CH_4$) is the second most important anthropogenic greenhouse gas after carbon dioxide ($CO_2$). It has an effective radiative forcing (ERF) of $\sim 0.54\,\mathrm{W\,m^{-1}}$ or one quarter of the ERF of $CO_2$ (Forster et al., 2021). It is a more potent greenhouse



gas than $CO_2$ by a factor of 81 per unit mass on a time horizon of 20 years (Forster et al., 2021) and its atmospheric lifetime,
which is dominated by the oxidation agent hydroxyl (OH) and transport and oxidation in the stratosphere, is relatively short
($\sim 12$ years, Szopa et al., 2021). For the above reasons, Shindell et al. (2012) proposed that the reduction of $CH_4$ emissions
was a potentially valuable short-term strategy to reduce the impact of anthropogenic emissions on the climate. This objective
became part of international environmental policy through the Global Methane Pledge, an initiative launched by the European
Union (EU) and the United States (U.S.) having the goal of reducing anthropogenic $CH_4$ emissions by $30\%$ from 2020 to 2030
(EU-US, 2021).

Landfills and waste related activities are estimated to account for one-fifth of anthropogenic $CH_4$ emissions (Saunois et al.,
2020). Within landfills, $CH_4$ (but also $CO_2$ and other gases such as precursors of short-lived climate pollutants and greenhouse
gases such as non-methane hydrocarbons) are produced by anaerobic decomposition of organic matter by microbes (e.g.,
Eklund et al., 1998). This methane has been and is released to the atmosphere nearly unhindered from unmanaged landfills.
Alternatively, in the context of greenhouse gas mitigation, measures exist to reduce these emissions by, e.g., installing gas
collection systems to recover a large fraction of the $CH_4$ (e.g., Parameswaran et al., 2023) for possible energy generation in
gas-fired power plants or flaring, and/or by deploying special covers, which partly oxidise $CH_4$ to the less potent greenhouse
gas $CO_2$ (e.g., Bogner et al., 1997). Despite these management, mitigation and reduction efforts, which are typically only
available in the developed world (Kumar et al., 2023; Kaza et al., 2018), reported $CH_4$ emissions from waste still account for
$\sim 24\%$ of the anthropogenic emissions in the European Union in 2022 (EEA, 2024).

Of relevance to this study, Tu et al. (2022) have investigated landfill sites and related facilities in Madrid, Spain. There,
significant amounts of $CH_4$ have been identified to be released to the atmosphere. Based on satellite observations, acquired
between May 2018 and December 2020, Tu et al. (2022) would suggest an underestimation in the reported emissions by the
European Pollutant Release and Transfer Register (E-PRTR, EEA, 2024) by a factor of $\sim 3$. Their estimated emissions would
correspond to $\sim 4\%$[1] of Spain's national $CH_4$ emissions in 2020 reported to the United Nations Framework Convention on
Climate Change (UNFCCC, EEA, 2023).

Landfill facility emissions need to be reported to the authorities (E-PRTR) in the European Union to meet the objective
of EU directives (European-Parliament, 2006). This reporting is usually fulfilled by using bottom-up estimates of methane
emissions described in IPCC (2006, 2019). However, emissions based on these bottom-up estimates may be underestimated
due to inaccurate model parameters (Wang et al., 2024) and often differ from those using atmospheric measurements (top-down,
e.g., Lu et al., 2022; Maasakkers et al., 2022; Duren et al., 2019).

In the past, different approaches have been used from different platforms to provide independent validation of waste facility
emissions. Commonly used measurement techniques are ground based measurements of the gases by closure chambers, scat-
tered across the landfill surface (e.g., Xie et al., 2022; Jeong et al., 2019; Trapani et al., 2013), greenhouse gas in situ analyser
measurements downwind of landfills with (e.g., Monster et al., 2014a, b) and without tracer (e.g., Liu et al., 2023; Xia et al.,
2023), as well as vertical or horizontal scanning lidar observations (e.g., Innocenti et al., 2017; Zhu et al., 2013) or Fourier-
transform infrared (FTIR) spectrometer measurements (Sonderfeld et al., 2017). Another strategy involves airborne (e.g., Ren

---

[1]Applying 28 as conversion factor for $CO_{2,eq}$ to $CH_4$.



et al., 2018; Krautwurst et al., 2017; Cambaliza et al., 2015, 2014; Peischl et al., 2013; Mays et al., 2009) or drone (Fosco et al.,
2024, and references therein) observations collecting in situ $CH_4$ concentrations downwind of a landfill. Comprehensive com-
parisons of these techniques are given by Mønster et al. (2019) and Babilotte et al. (2010). Recently, passive remote sensing
imaging instruments have been deployed, that map $CH_4$ column amounts of the plumes leaving a landfill (e.g., Cusworth et al.,
2024, 2020) in addition to airborne thermal imagers (e.g., Tratt et al., 2014). These allow not only precise leakage detection,
but also emission quantification. Moreover, nowadays, high spatial resolution (in the order of several tens of meters) satellite
instruments are exploited in terms of $CH_4$ column observations for a more regular investigation of landfills (e.g., McLinden
et al., 2024; Maasakkers et al., 2022) than was possible with irregular campaign deployments in the past. However, also satel-
lite observations having a coarse spatial resolution of some kilometres were used to constrain landfill emissions (e.g., Balasus
et al., 2024; Nesser et al., 2024) but not to a detail possible by their high spatial counterparts.

  Beside the mentioned predominately passive remote sensing approaches, there is currently no satellite mission using ac-
tive $CH_4$ remote sensing in orbit and we are not aware of any studies utilising active airborne remote sensing to measure
landfill emissions. However, Amediek et al. (2017) have quantified local $CH_4$ emissions from coal mine ventilation shafts,
demonstrating the capabilities of active airborne remote sensing measurements for such endeavours.

  In the analysis described in this manuscript, we use a data set of unique observations collected by passive imaging and active
airborne remote sensing instruments of, generally speaking, atmospheric $CH_4$ column gradients combined with auxiliary in
situ measurements of $CH_4$, $CO_2$ and 3D winds in support of the remote sensing data. It was the first time that this payload was
flown aboard the same aircraft acquiring spatially and temporally collocated active and passive remote sensing measurements
side-by-side for the acquisition of atmospheric $CH_4$ column observations. The greenhouse gas lidar CHARM-F ($CO_2$ and $CH_4$
Atmospheric Remote Monitoring Flugzeug) is an airborne demonstrator for the future satellite mission MERLIN (Methane
Remote Sensing LIDAR Mission, Ehret et al., 2017). The passive imaging MAMAP2DL (Methane Airborne Mapper 2D
- Light) remote sensing instrument demonstrates the applicability of the $CH_4$ proxy retrieval (see below) at scales probed
by CO2M (Copernicus Anthropogenic Carbon Dioxide Monitoring, Sierk et al., 2021) and TANGO (Twin Anthropogenic
Greenhouse Gas Observer, SRON, 2024). The observations were collected in 2022 as part CoMet 2.0 (Carbon Dioxide and
Methane) Arctic mission in Canada (CoMet, 2022). Prior to the transfer to Canada, an initial research flight was carried out
to test all the instruments. This test flight was performed on the 4[th] August over Madrid to investigate the unexpected high
landfill emission rates reported by Tu et al. (2022) and in a webstory from the European Space Agency (ESA) from October
2021 (ESA, 2021).

  In Sect. 2, we provide a brief summary of the CoMet 2.0 mission and introduce the main instruments MAMAP2DL and
CHARM-F used in this study (Sect. 2.1). This also includes a description of the algorithms used to infer $CH_4$ columns from
the measurements (Sect. 2.2), additional steps necessary to achieve comparability between the passive and active observations
(Sect. 2.3), and the cross-sectional flux method, which is used to quantify the $CH_4$ emissions (Sect. 2.4). Section 3 describes
the observed $CH_4$ plumes over Madrid from both remote sensing instruments. This data is used to pin-point the exact source
locations within the landfill area (Sect. 3.1), followed by a rigorous comparison of the active and passive data (Sect. 3.2), the





resulting emission fluxes (Sect. 3.3), and a comprehensive discussion of potential uncertainties (Sect. 3.4). We close the paper by discussing our fluxes in a broader context (Sect. 4) and summarising our findings (Sect. 5).

## 2 Methods and Data

### 2.1 Campaign and Instrumentation

In the following, we provide an overview of the observations, the target, used instruments and applied algorithms to retrieve $CH_4$ columns and eventually fluxes through our cross-sections, which are related to the emission rates of the waste treatment facilities in the measurement area.

#### 2.1.1 CoMet 2.0 Arctic

The analysed data was collected in the framework of the Carbon Dioxide and Methane Mission CoMet 2.0 Arctic, executed in Canada in 2022. The main objective of CoMet 2.0 was to investigate the influence of the contribution of major Arctic wetlands to the greenhouse gas budget in summer and how these compare to anthropogenic emissions from, e.g., fossil fuel exploitation and production sites or landfills. To achieve this goal, a comprehensive suite of instruments was installed aboard the German research aircraft HALO (High Altitude and Long Range Research Aircraft, operated by the DLR, Deutsches Zentrum für Luft-
und Raumfahrt, type: Gulfstream G550). HALO was based in Edmonton, Canada, during the period from the 7[th] August to the 16[th] September. The measurement flight over Madrid, Spain, on the 4[th] August 2022, was planned as a test flight to ensure that all instrumentation worked nominally prior to the transfer to Canada but also enabled us to investigate the emissions from landfills. This study focuses on the atmospheric $CH_4$ column observations from the passive MAMAP2DL and the active CHARM-F airborne remote sensing instruments. Additionally, auxiliary data from the in situ sensor JIG (Jena Instrument
for Greenhouse gases) for $CO_2$ and $CH_4$ concentrations and of the basic data acquisition system of HALO, BAHAMAS (Basic Halo Measurement and Sensor System) including SHARC (Sophisticated Hygrometer for Atmospheric ResearCh) were used. The HALO basic data acquisition suite collects various atmospheric parameters such as the 3D wind field, temperature, humidity, and aircraft attitude data that is used for interpretation of the remote sensing data. The main instrumentation and the flight strategy are described in more detail below.

### 2.1.2 Target Description and Flight Strategy

The targets under consideration were the Mancomunidad del Sur landfill in the municipality Pinto (40.264°N, 3.633°W; here-after: Pinto landfill) and the Valdemingómez technology park (VTP, 40.332°N, 3.586°W) in the south-east of Madrid, Spain. The latter is a waste treatment complex accepting around 4000 tons of waste daily and housing several waste treatment facilities including the largest biomethane plant in Spain (Calero et al., 2023), which is also one of the largest in Europe (UABIO, 2022).
Additionally, it contains landfill sites. The non-operating Valdemingómez landfill (40.331°N, 3.580°W) equipped with a gas recovery system and an active landfill site (40.325°N, 3.591°W; hereafter: Las Dehesas landfill) next to the waste treatment





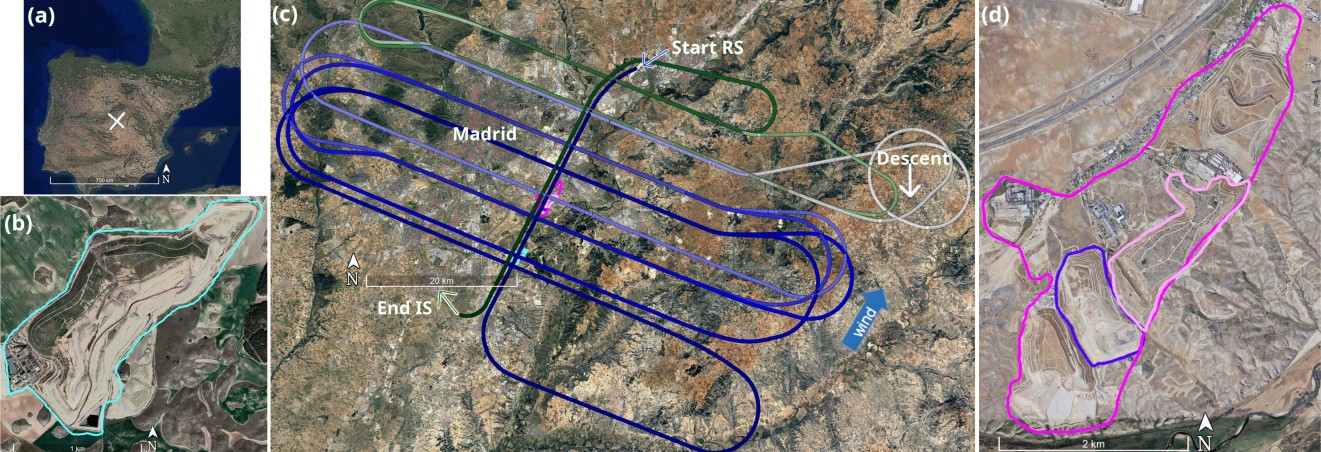

**Figure 1.** Top-view of the flight path of the HALO aircraft during the test flight over Madrid. An overview map in Google Earth of Spain is shown in **(a)** and Madrid is marked by the white cross. A zoom to the Pinto landfill is shown in **(b)** and to the Valdemingómez Technology Park (VTP) including the closed Valdemingómez and open Las Dehesas landfills in **(d)**. The Pinto landfill and the VTP are marked by the cyan and dark pink solid lines, respectively. Bright pink and purple mark the closed and open landfills, respectively, in the VTP. The flight path is shown in **(c)** whereby bluish colours represent the remote sensing (RS) part at $\sim 7.7\,\mathrm{km}$ a.g.l. and greenish colours the in situ (IS) part at $\sim 1.6\,\mathrm{km}$ a.g.l. (above ground level) of the flight. For better visualisation, the greenish in situ part is slightly shifted to the north-west because otherwise part of the legs would be hidden by RS legs. The map underneath is provided by Google Earth (Image © Landsat/Copernicus, Maxar Technologies).

plant Las Dehesas. The northern half of the Las Dehesas landfill, where certain areas (i.e. cells) are already full and therefore closed, are also equipped with a gas recovering system (Sánchez et al., 2019). The two landfills in the technology park spread over an area of $\sim 0.9$ and $0.6\,\mathrm{km}^2$ for the inactive Valdemingómez and the active Las Dehesas sites, respectively. More de-
tails about the different facilities are in the Annual Report for 2022 for the VTP (Madrid, 2022). The Pinto landfill, further to the south, stretches over $\sim 1.5\,\mathrm{km}^2$. It opened in 1987, is still operational with around 800,000 tons of waste being dumped yearly (Rodriguez, 2022), and the already closed parts of the landfills are equipped with gas recovering system (MdS, 2024). In E-PRTR, combined annual reported $CH_4$ emissions for the two facilities "DEPOSITO CONTROLADO DE RESIDUOS UR-BANOS DE PINTO" and "VERTRESA-URBASER, S.A. UTE (UTE LAS DEHESAS)", labelled as "landfills", are $0.2\,\mathrm{t\,h}^{-1}$
in 2022. We assume that these reported values are representative for the two areas we investigate, which includes landfills and waste treatment plants, as no other sources are given.

To properly investigate emissions from these two landfills, dedicated flight patterns where the aircraft is levelled (so called flight legs) were aligned perpendicular to the forecasted wind direction (Fig. 1). The overflight time was between 13:00 and 15:40 local time (11:00 to 13:40 UTC) on the $4^{\mathrm{th}}$ August 2022. This time window was chosen using knowledge of the weather
forecast predicting stable winds around noon, which also favoured the observations by the passive remote sensing instrument due to the high position of the sun.



Prevailing wind direction during the flight was from approximately SSW, aligned with the two waste treatment areas. For later emission rate estimates (Sect. 2.4), flight legs were mostly flown perpendicular to the mean wind direction at several distances from the sources downwind at altitudes of $\sim 7.7$ and $1.6\,\mathrm{km}$ a.g.l. (above ground level) as depicted in Fig. 1. The higher flight altitudes were flown to optimise passive and active remote sensing observations, whereas the lower altitudes were used to primarily collect in situ observations within the boundary layer (BL) in and out side of the emission plumes as well as high spatial resolution $CH_4$ imaging data. Remote sensing observations were collected upwind of each of the landfills to account for potential inflow of $CH_4$ and to separate emissions from the two waste treatment areas.

Moreover, the flight pattern started with one straight leg against the wind direction directly overflying the landfills at remote sensing altitude at $\sim 7.7\,\mathrm{km}$ a.g.l. to identify emission hot spots using the imaging capabilities of MAMAP2DL. Then, perpendicular remote sensing legs were flown in an alternating order due to the large turning radius of the aircraft. Three legs were repeated twice. Afterwards, the aircraft descended to the altitudes optimal for in situ measurements, flying four legs downwind of both areas. Lastly, the flight pattern was closed with a straight leg overflying both landfills directly at in situ altitude. The in situ flight was performed after the remote sensing part towards the afternoon when a fully developed BL favours these measurements.

### 2.1.3 Passive MAMAP2D-Light Remote Sensing Imaging Instrument

MAMAP2DL (Methane Airborne Mapper 2D - Light) is a light-weighted airborne imaging greenhouse gas sensor for mapping atmospheric column concentration anomalies of $CH_4$ and $CO_2$ (in $\mathrm{molec\,cm^{-2}}$ or in % relative to the given background column). It builds on the heritage of MAMAP (Gerilowski et al., 2011) and is a passive remote sensing instrument collecting backscattered solar radiation mainly from the ground, which has been modified by absorption from atmospheric gases. Using absorption spectroscopy, the depth of these absorption lines is interpreted as column gas concentrations in the atmosphere (for details, see Sect. 2.2.1). MAMAP2DL comprises a grating spectrometer and records spectra in the range between 1558 and $1689\,\mathrm{nm}$, where prominent absorption features of $CH_4$ and $CO_2$ exist (Krings et al., 2011), having a spectral resolution of around $1\,\mathrm{nm}$ with a spectral sampling of $\sim 3$ to 4 pixels per FWHM (full width at half maximum). The front optics maps the measurement scene via 28 glass fibres onto a 2D sensor consisting of 384 pixels in horizontal and 288 pixels in vertical direction. The horizontal direction maps onto the spectral axis and the vertical direction onto the spatial axis (see also Fig. 2). Each glass fibre is mapped onto around 6 usable lines on the chip which are binned to increase the signal-to-noise-ratio before further analysis (see Sect. 2.2.1). For the Madrid flight, the exposure time for a single readout was between 40 and $45\,\mathrm{ms}$. This would result in a ground scene size of $\sim 110\,\mathrm{x}\,8.5\,\mathrm{m}^2$ (across x along flight direction). To achieve quadratic ground scenes, we therefore bin 13 ground scenes in along flight direction after the retrieval of the column anomalies.

The instrument was built at the Institute of Environmental Physics (IUP) at the University of Bremen (UB) and its design has its heritage in the non-imaging greenhouse gas sensor MAMAP (Gerilowski et al., 2011) built at IUP UB in 2006. MAMAP2DL shares many of the optical concepts developed in MAMAP but uses a spectrometer consisting of lenses instead of mirrors and a 2D-detector array allowing for imaging of emission plumes. MAMAP's column observations have been proven to be of high data quality achieving a single-measurement precision of $\sim 0.2$ % for the background normalised column anomaly (Krautwurst



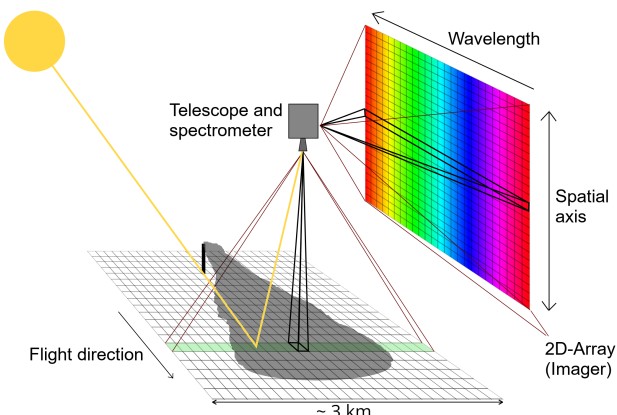

**Figure 2.** The schematic diagram shows the measurement principle of MAMAP2DL. The instrument simultaneously acquires 28 ground scenes across track with a swath width of $\sim 3\,\mathrm{km}$ at a flight altitude of $\sim 7.7\,\mathrm{km}$ a.g.l.. The final ground scene size is $\sim 110\,\mathrm{x}\,110\,\mathrm{m}^2$.

et al., 2021). Its observations have been used successfully to estimate $CO_2$ emissions from single power plants (Krings et al., 2011), power plant clusters (Krings et al., 2018), and were part of a model validation study for power plant emissions (Brunner et al., 2023). $CH_4$ emissions from coal mine ventilation shafts (Krautwurst et al., 2021; Krings et al., 2013) and landfills (Krautwurst et al., 2017) were determined, as well as upper limits of emissions from offshore geological $CH_4$ seeps (Krings
et al., 2017; Gerilowski et al., 2015) were estimated.

### 2.1.4    Active CHARM-F Remote Sensing Instrument

CHARM-F ($CO_2$ and $CH_4$ Remote Monitoring - Flugzeug), developed and operated by DLR, is an integrated-path differential-absorption (IPDA) lidar instrument that consists of a pulsed laser transmitter and a receiver system. The transmitter is based on two optical parametric oscillators (OPOs) which are pumped by means of diode-pumped, injection seeded, and Q-switched
Nd:YAG lasers in a master-oscillator power-amplifier configuration. Installed on an aircraft, the nadir oriented lidar emits laser pulses at two precisely tuned wavelengths in the near infrared at $\sim 1645\,\mathrm{nm}$ for $CH_4$ and $\sim 1572\,\mathrm{nm}$ for $CO_2$. These two laser pulses propagate through the atmosphere until they are backscattered at a surface. From the backscattered intensities entering the detector, absolute column-averaged mixing ratios of carbon dioxide ($XCO_2$ in ppm) and methane ($XCH_4$ in ppb) below the aircraft are derived (see Amediek et al., 2017, and Appendix C1). A schematic illustration of the IPDA measurement
principle is shown in Figure 3.

The generation of narrow band wavelength is realised by injection seeding the OPOs with continuous wave (cw) radiation from stabilised distributed feedback (DFB) lasers. In order to fulfil the stringent requirements on frequency stability for the online and offline wavelengths, a sophisticated locking scheme has been developed that is based on DFB lasers referenced to a multi-pass absorption cell and offset locking techniques (Amediek et al., 2017; Quatrevalet et al., 2010). The online and offline
laser pulses are emitted as double pulses with a temporal separation of $500\,\mu\mathrm{s}$ and a repetition rate of $50\,\mathrm{Hz}$.



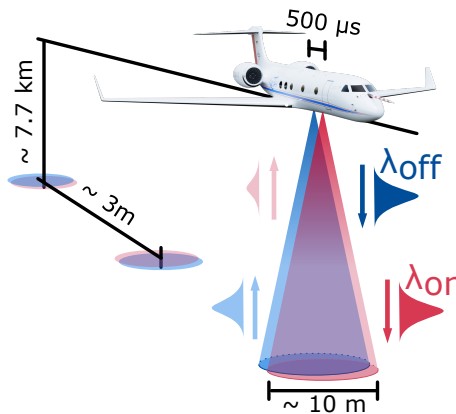

**Figure 3.** The measurement geometry of CHARM-F installed on the HALO aircraft. Two laser pulses are emitted towards the Earth with a delay of $500\,\mu s$. The laser pulse with the online wavelength is denoted as $\lambda_{on}$, while the one with the offline wavelength is denoted as $\lambda_{off}$. The concentration in the surveyed column can be derived from the backscattered intensities. As the footprints are larger than the distance between consecutive pulse pairs, they actually overlap. For the visualisation above, they were pulled apart. The order in which the on-off pairs are sent out alternates.

CHARM-F's receiving system consists of four receiving telescopes, two for each greenhouse gas, with a diameter of 20 and 6 cm, and equipped with InGaAs pin diodes and InGaAs avalanche photo diode (APD), respectively. This redundant measurement capacity proved to be very valuable for an independent quality assessment of the data. The received signals are sampled using fast digitisers and processed by means of a home-built data acquisition system. Two digital cameras (in the VIS and NIR spectral range) provide additional context information of the ground scene.

In the context of this study, we only make use of the ($X\mathrm{CH}_4$) measurement, which is fully independent of the $\mathrm{CO}_2$ channels. The $\mathrm{CH}_4$ wavelengths, at which CHARM-F operates, are at 1645.55 and 1645.86 nm for on- and offline wavelength, respectively.

Previous work has shown that CHARM-F measurements are suitable for quantifying $\mathrm{CH}_4$ and $\mathrm{CO}_2$ emission sources (Amediek et al. (2017); Wolff et al. (2021)). Furthermore, CHARM-F serves as a technology demonstrator for the MERLIN spaceborne methane lidar that will measure methane columns globally starting in the late 2020s (Ehret et al., 2017).

### 2.1.5 Auxiliary Data

In support of the remote sensing data, we use additional measurements from in situ instruments aboard the HALO aircraft and model data. To adapt radiative transfer model simulations (RTMs) used later during the retrieval process of MAMAP2DL data (for details, see Sect. 2.2.1) to prevailing atmospheric background conditions, we use $\mathrm{CH}_4$ and $\mathrm{CO}_2$ in situ observations from JIG (operated by Max-Planck-Institute for Biogeochemistry, MPI-BGC, Gałkowski et al., 2021; Chen et al., 2010) and $\mathrm{H}_2\mathrm{O}$ from the BAHAMAS suite (Giez et al., 2023), recorded at a frequency of 1 Hz ($\mathrm{CH}_4$), 1 Hz ($\mathrm{CO}_2$), and 10 Hz ($\mathrm{H}_2\mathrm{O}$). $\mathrm{CH}_4$ and $\mathrm{CO}_2$ are measured with a precision and accuracy of 1 ppb and 2 ppb, and 0.1 ppm and 0.2 ppm, respectively. Measurements





of $H_2O$ have an uncertainty of up to $\sim 5\%$. Furthermore, for the correct georeferencing of the remote sensing observations,

positioning and attitude data of the HALO aircraft also measured by the BAHAMAS suite at $10\,Hz$ is used.

A critical parameter for the emission rate quantification is the wind in the BL, where the exhaust plumes are located. The BAHAMAS system delivers highly-accurate in situ wind measurements at $10\,Hz$. The uncertainty of the horizontal wind speed and direction usually is $\sim 0.14\,m\,s^{-1}$ and $\sim 2.9°$, respectively, for low flying altitudes (Giez et al., 2023). A special data analysis for the Madrid flight shows slightly increased errors due to the replacement of the static pressure sensor and the strong

turbulence, but the wind measurements are still of very high quality with uncertainties of $\sim 0.2\,m\,s^{-1}$ and $\sim 4°$ for the relevant altitude levels.

During the remote sensing measurements, the wind information within the BL is needed but not measured, as HALO was flying well above at $\sim 7.7\,km$ a.g.l.. Therefore, we use the wind measurements from the BAHAMAS system to verify the quality of the European Center for Medium-Range Weather Forecast (ECWMF) reanalysis v5 (ERA5) model (Hersbach et al.,

2020) in that area on that day. We use ERA5 data with a temporal and horizontal spatial resolution of one hour and $31\,km$, respectively, and 137 altitude levels. The comparison is found in Appendix D1 and shows a very good agreement between measurements and model with averaged deviations of $0.05\,m\,s^{-1}$ and $0.8°$ within the BL and thus gives confidence for the use of the ERA5 winds in our study.

We also use airborne in situ observations to validate the boundary layer height (BLH) from ERA5. The analysis is given in

Appendix D2 and reveals that the observed boundary layer heights during the flight are up to $15\%$ lower than those given in ERA5.

## 2.2    Retrieval Algorithms

The following subsections describe how atmospheric columns are derived from the measured spectra in the case of the passive instrument, and from the backscattered laser pulses in the case of the active instrument.

### 2.2.1    $CH_4$ Column Anomalies by MAMAP2DL

For the analysis of the MAMAP2DL spectral data, the WFM-DOAS (Weighting Function Modified - Differential Absorption Spectroscopy) approach is used. It was originally developed for the spaceborne instrument SCIAMACHY aboard ENVISAT (Buchwitz et al., 2000) and later adapted to airborne geometry for the MAMAP sensor (Krings et al., 2011). The latest version of the algorithm is described in Krautwurst et al. (2021) and has been applied to the imaging data from MAMAP2DL. The

results are background normalised column anomaly maps of $CH_4$ or just $CH_4$ column anomalies.

For the current study, the single measurement precision of the $CH_4$ column anomalies, derived from MAMAP2DL columns in areas not (or only little) influenced by emissions, is around $0.4\%$ $(1\text{-}\sigma)$ for $\sim 110\,x\,110\,m^2$ ground scenes. The accuracy of the $CH_4$ column anomalies is estimated to be around $0.14\%$ possibly not correctable by the applied normalisation processes. Further details about the algorithm setup and uncertainties associated with it are given in the Appendix B.

The retrieved anomaly maps are also orthorectified (also known as georeferencing). A correction is applied along the lines as described in Schoenhardt et al. (2015) to account for the orientation of the aircraft (e.g., pitch, roll, yaw) which would lead to





spatially incorrectly projected ground scenes and would prohibit proper source allocation. For that, attitude data provided by the BAHAMAS system at $10\,\text{Hz}$ resolution has been used. Visual inspection of measured intensity maps overlaid on Google Earth yields a relative accuracy to Google Earth imagery of $\sim 110\,\text{m}$ (or approx. one MAMAP2DL ground scene, see Appendix B4
for details).

### 2.2.2   $X\text{CH}_4$ by CHARM-F

IPDA lidars, such as CHARM-F, directly measure the differential absorption optical depth between the online and offline wavelength (DAOD), from backscattered signals without the need of auxiliary information. The DAOD is converted into a weighted column average of the dry-air molar mixing ratio of the trace gas in question by applying the so-called weighting
function (see Appendix C1).

The weighting function depends, apart from precise spectral information, also on external information about the state of the atmosphere below the aircraft such as temperature, pressure and humidity, vertically resolved. For spectroscopic reasons, the sensitivity of CHARM-F for methane is highest close to the ground, but varies by only a few percent within the lower troposphere (Ehret et al., 2008, 2017).
As mentioned in Sect. 2.1.4, CHARM-F is equipped with two detector channels for $\text{CH}_4$. For this study, $X\text{CH}_4$ measurements from both detectors are combined in a weighted average, where the inverse variance due to noise is used as weights.

For the conditions present during the Madrid measurements the statistical uncertainty (1-$\sigma$ error) of a single $X\text{CH}_4$ measurement (averaging both available detectors), based on one online and one offline pulse, is on the order of $5\%$. The main contributing random sources of error are shot and detector noise, as well as random variations in the speckle and albedo pattern
(Ehret et al., 2008).

When averaging along the flight track over multiple double-pulse measurements, this uncertainty decreases, as expected, with one over the square root of the number of measurements, until, systematic drifts and offsets start to dominate. For the $3\,\text{s}$ averaging, which corresponds to a distance of about $500\,\text{m}$ on the ground and which is used in the plots, visualisations, and flux computations that are shown in the following, the statistical measurement uncertainty is roughly $10\,\text{ppb}$ or $0.5\%$.
Due to the background normalisation that is performed as part of the flux calculation conducted in this context, the results are largely unaffected by constant offsets and slow drifts in the methane column. Our conversion into total columns and comparison with predicted values from the Copernicus Atmosphere Monitoring Service (CAMS) global inversion model (Segers and Houweling, 2023) suggest an offset of less than $0.5\%$. See Appendix C3 for more details.

### 2.3   Common Columns

In order to allow for a better comparison between active and passive remote sensing measurements and the application of a uniform approach for computing cross-sectional fluxes with both instruments, CHARM-F partial columns (pc, below the aircraft) $X\text{CH}_4$ have been converted into total column (tc) relative enhancements (column anomalies). This conversion requires assumptions about the composition and structure of the atmosphere that are not directly accessible from CHARM-F measurements alone. A detailed formalised description of this conversion can be found in Appendix C2.



In order to estimate a relative column anomaly, the methane concentration from the CAMS global inversion model (Segers and Houweling, 2023) is used as a reference. For the partial column between the aircraft and the ground, $XCH_4$ measured by CHARM-F is compared to the corresponding value calculated based on CAMS and the CHARM-F weighting function. For the partial column above the aircraft the anomaly is zero by definition.

For the partial column below the aircraft a small correction (corresponding to a 2-3 % relative scaling effect on the column
anomaly) for the effect of the weighting function has to be applied to the column anomaly computed using CHARM-F measurements. As explained in Sect. 2.2.2, due to the spectroscopic properties of methane and the choice of lidar wavelengths, CHARM-F is somewhat more sensitive close to the ground than in the upper troposphere. As a correction factor for the anomaly of the partial column, we use the ratio between, the average weighting function for the full column below the aircraft, and the average column only within the BL. This assumes that methane emitted from the landfills is only dispersed within the BL[2].

Finally, the anomalies of the partial columns above and below the aircraft are combined in a weighted average with number density of (vertically summed) air molecules per area as weights.

## 2.4   Flux Computation

Already during the planning activities for the Madrid flight (see Sect. 2.1.2), the position and orientation of the flight legs were designed for the application of a cross-sectional mass balance approach or flux method. To account for instrument spe-
cific properties, two slightly different methods are applied and described in the following. Both follow the widely applied approach for in situ (Klausner et al., 2020; Peischl et al., 2018; Cambaliza et al., 2015; Lavoie et al., 2015) or remote sensing (Fuentes Andrade et al., 2024; Wolff et al., 2021; Reuter et al., 2019; Krings et al., 2018; Varon et al., 2018; Frankenberg et al., 2016) observations, where the mass of molecules that is transported through an imaginary curtain or cross-section, is computed by

$$F_{\text{cs}} = f \cdot \sum_i \Delta V_i \cdot \Delta x_i \cdot u_i \cdot \cos(\alpha_i), \tag{1}$$

where $F_{\text{cs}}$ is the resultant and areal integrated $CH_4$ mass flux or the $CH_4$ mass flow rate in $\text{t hr}^{-1}$ of one cross-section. In the following, we use the term 'flux' when talking about mass flow rates through a cross-section and 'emission rate' if the flux is attributed to a certain source or source area. $f$ is a conversion factor[3] to transform to units of $\text{t hr}^{-1}$, $\Delta V_i$ is the retrieved $CH_4$ column anomaly in $\text{molec cm}^{-2}$, $\Delta x_i$ the valid length element for the corresponding $\Delta V_i$ in metres, $u_i$ the absolute wind
speed (or effective wind speed valid for the plume) in $\text{m s}^{-1}$, and $\alpha_i$ is the angle between the normal of the length element and the wind direction in degrees to calculate the wind fraction perpendicular to the length element. The sum indicates the summation over all observations $i$ within the plume.

---

[2]The way the weighting function is constructed ensures correct values for the average column concentration for a homogeneous methane mixing within the column below the aircraft. Any deviation from homogeneity (column anomaly), as we deal with here by assuming the concentration enhancement from the sources to affect the PBL only, requires a correction factor like described in the text.

[3]E.g. including the conversion from number of $CH_4$ molecules per $cm^2$ to mass of $CH_4$ per $m^2$.





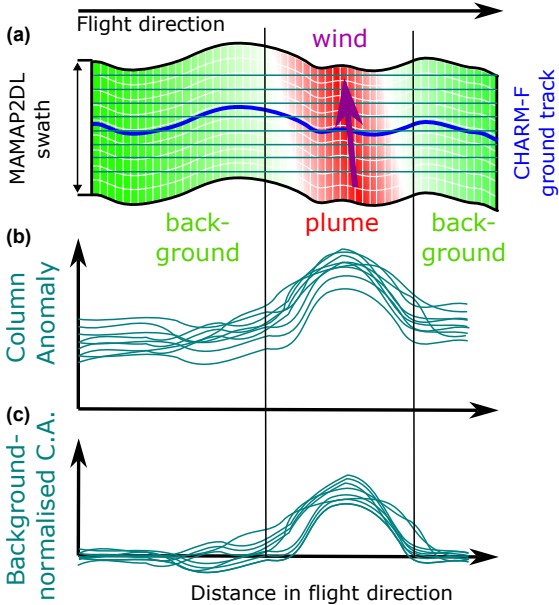

**Figure 4.** These schematic diagrams explain the principle used to estimate the $CH_4$ fluxes from the measured MAMAP2DL anomaly maps and CHARM-F anomalies. **(a)** shows schematically the MAMAP2DL flight leg with the flight direction parallel to the x-axis. The wind direction is approximately perpendicular to the flight direction. Horizontal dark cyan lines indicate the added cross-sections for which the fluxes are computed. Vertical black lines, parallel to the y-axis, separate the plume and background areas. The latter is used to normalise the entire cross-section and to compute the $CH_4$ anomalies within the plume area. The (10 m wide) CHARM-F ground track is depicted in blue. **(b)** shows the column anomalies along the cross-sectional lines from **(a)**. **(c)** shows the normalised cross-sectional lines from **(b)** normalised by the background observation of the respective cross-section.

The first modification of Eq. 1 accounts for the characteristics of the imaging data from the MAMAP2DL sensor. The retrieved $CH_4$ anomaly maps consist of strips with a swath width of $\sim 3\,km$ and 28 ground scenes across track (see Fig. 2),

which are, however, additionally distorted by the movement of the aircraft (see schematic diagram in Fig. 4, a). In a first step, the leg is aligned parallel to the x-axis (x-axis, thus, corresponds to flight direction in Fig. 4, a, from left to right).

Next, we apply $n$ cross-sections parallel to the x-axis (dark cyan solid lines in Fig. 4, a) evenly distributed across the swath, and define plume and background areas as indicated in Fig. 4 based on visual inspection of the plume signal across the entire swath similar to the approach taken in other publications (e.g., Krings et al., 2018; Krautwurst et al., 2017; Frankenberg et al.,

2016). To compute the $CH_4$ anomalies along one cross-section, it is normalised by the observations in the local background area (i.e. the cross-section is divided by values from a straight line which has been fitted to observations in the background only). This approach also accounts for smooth atmospheric concentration gradients or other systematic effects not considered during the retrieval.

The process of estimating the $CH_4$ background normalised column anomalies is shown schematically in Fig. 4 (a) to (c).

The objective of this sampling approach is to determine representative fluxes of one leg by considering as much available





information as possible. Therefore, the number $n$ of cross-sections is chosen such that the swath is well covered (i.e. every $10\,\mathrm{m}$) and the number has basically no effect on the average flux of one leg calculated later. In the same manner, one cross-section is sampled with a sufficient number of points (i.e. every $10\,\mathrm{m}$) so that changing this sampling has also effectively no effect on the flux anymore. As a result, Eq. 1 simplifies to become:

$$F_{\mathrm{M2D,\,cs}} = f \cdot \Delta x \cdot u \cdot \cos(\alpha) \cdot \sum_i \Delta V_i \qquad (2)$$

as wind speed $u$, angle $\alpha$ and length element $\Delta x$ are constant for one cross-section. The wind speed and direction is calculated at the position and overflight time of every leg from ECMWF ERA5 fields (see Sect. 2.1.5 or Appendix D for validity of ERA5 data during the flight). We assume effective mixing of the emissions in the boundary layer and, thus, average the wind over all layers in ERA5 from the bottom to the top of the BL. Each layer is weighted by its number of air molecules. For an individual leg, the same winds are applied to the MAMAP2DL and, later, also to the CHARM-F observations for the flux calculation. The average flux $F_{\mathrm{M2D,\,leg}}$ for an entire MAMAP2D leg is computed by

$$F_{\mathrm{M2D,\,leg}} = \frac{\sum_i F_{\mathrm{M2D,\,cs}_i}}{n}. \qquad (3)$$

The errors of fluxes for one cross-section $F_{\mathrm{M2D,\,cs}}$ and of the average flux of one leg $F_{\mathrm{M2D,\,leg}}$ are then computed by error propagation using the errors on the individual parameters used in Eq. 2, as explained in Appendix F.

However, the approach above does not allow for a 1-to-1 comparison of fluxes between those determined form the measurements made by the imaging MAMAP2DL and those by the 1D CHARM-F instruments. Both datasets are actually distorted by the aircraft movement (i.e. predominantly the aircraft roll). The straight cross-sections introduced above for MAMAP2DL do not follow the distorted CHARM-F ground track. However, as seen in Fig. 4 (a), the CHARM-F ground track follows one fixed MAMAP2D viewing angle approximately in the middle of the swath because the effect of the distortion is the same for both instruments. Therefore, Eq. 1 is directly applied to the measurements, with each parameter being evaluated individually for one measurement i.e. the wind speed, direction and length segment are not constant anymore. The resulting CHARM-F flux is then representative for this one leg. Definition of plume and background areas remain the same.

Independent of the applied approach for MAMAP2DL or CHARM-F data described above, the fluxes from several legs, computed by Eq. 3, are then again averaged to derive mean emission rates $F_{\mathrm{M2D,\,ar\text{-}aver}}$ and $F_{\mathrm{CHARM\text{-}F,\,ar\text{-}aver}}$[4] of certain areas in the measurement area for the respective instrument:

$$F_{\mathrm{M2D\ or\ CHARM\text{-}F,\,ar\text{-}aver}} = \frac{\sum_i F_{\mathrm{M2D\ or\ CHARM\text{-}F,\,leg}_i}}{p}, \qquad (4)$$

where $p$ is the number of legs. This applies, for example, to the area in the lee of the two waste treatment areas, which is representative of the total emissions from the measurement area.

---

[4] 'ar-aver' stands for areal averages.



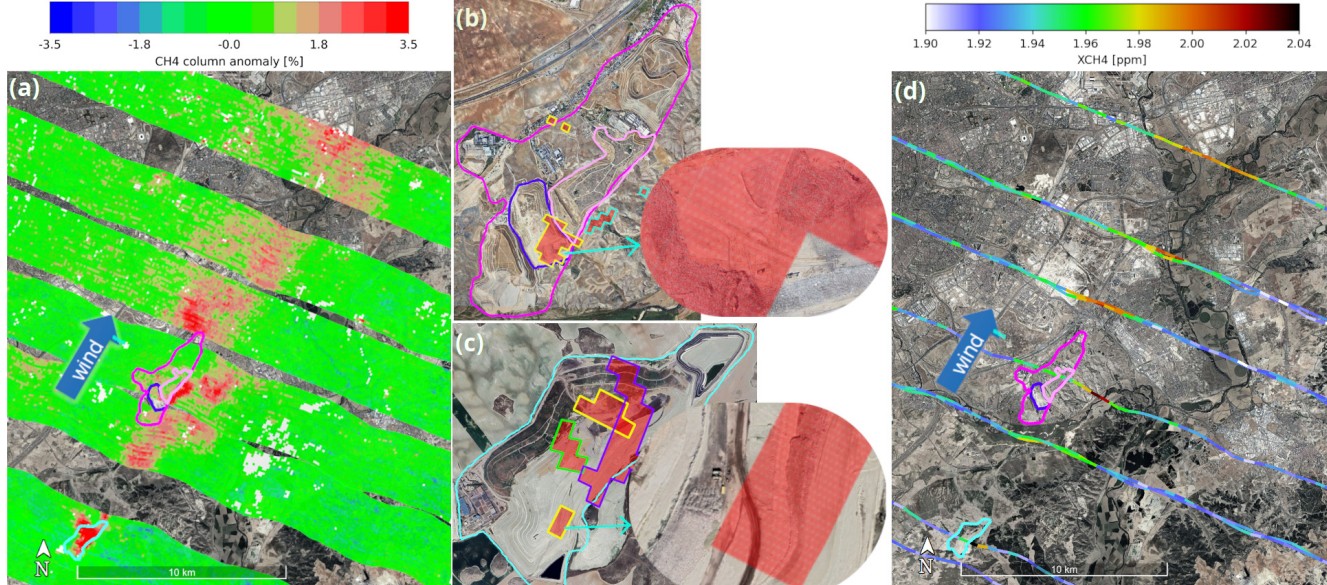

**Figure 5. (a)** and **(d)** show the retrieved $CH_4$ column anomalies from MAMAP2DL and the $XCH_4$ from CHARM-F, respectively. **(b)** and **(c)** are zoomed pictures of the two landfills including the MAMAP2DL observations with the largest anomalies only ($\sim 4\%$ for VTP or Las Dehesas, **b**, and $\sim 8\%$ for Pinto, **c**). The small insets in **(b)** and **(c)** zoom in further detailing some activities across the areas with largest observed enhancements. The shown Google Imagery was recorded in August 2022. The different colours of the borders around the observations mark the enhancements observed in different flight legs. The waste treatment areas are encircled by different coloured solid lines: Cyan for the Pinto landfill; dark pink for the VTP; bright pink for the Valdemingómez landfill; purple for the Las Dehesas landfill. The map underneath is provided by Google Earth (Image © Landsat/Copernicus, Maxar Technologies).

## 3 Results

### 3.1 Observed Column Enhancements over Madrid and Source Attribution

Figure 5 visualises the retrieved and orthorectified $CH_4$ column anomaly maps derived from the MAMAP2DL measurements (a, as described in Sect. 2.2.1) and the $XCH_4$, given as 3 seconds averages, derived from CHARM-F data (d, as described in Sect. 2.2.2) for the different remote sensing legs acquired at a flight altitude of $\sim 7.7\,\mathrm{km\,a.g.l.}$. Both data sets clearly show $CH_4$ enhancements (in red) located at or downwind of the waste treatment areas, whereas upwind or south-west of the Pinto landfill in the bottom left corner, there are no indications of inflow of external enhanced $CH_4$ in the measurement area. These observations are also confirmed by two legs flown in along wind direction at two different flight altitudes (see Appendix A and Fig. A1 for details). Especially for the Pinto landfill, there is a clear plume visible in both overflights, $\sim 2.5\,\mathrm{h}$ apart from each other.





The highest $CH_4$ concentrations are observed at or close to landfills. $CH_4$ hot spots, with peak enhancements of around
$17\%$, are located at the eastern part of the Pinto landfill according to the MAMAP2DL imaging data. This hot spot is also
captured by the CHARM-F instrument with $XCH_4$ of up to $2.28\,\mathrm{ppm}$.

The insets (b) and (c) show more details of the individual landfills including the locations of the highest column anomalies,
which were identified in different overflights. Marked regions in the southeast of the landfills are areas which are, most proba-
bly, responsible for a large fraction of the observed emissions. The Google Earth imagery recorded in August 2022 (the same
month and year as our flight) clearly shows that these hot spots are directly located over active landfill areas where waste is
deposited. However, we cannot exclude that other parts, for example, closed cells of the landfills or facilities located in these
waste treatment areas also contribute to the observed $CH_4$ plume.

### 3.2   Column Comparison between MAMAP2DL and CHARM-F

Figure 5 shows a good visual agreement between the column anomalies of the passive MAMAP2DL and the $XCH_4$ of the
active CHARM-F instruments. In order to perform a more rigorous comparison between the two types of atmospheric $CH_4$
columns, we convert the $XCH_4$ partial columns derived from CHARM-F to total column anomalies (see Sect. 2.3). We then
identify the ground scene in the MAMAP2DL swath which corresponds to the CHARM-F measurements, which are approx-
imately located in the middle of the MAMAP2DL swath. This procedure ensures the selection of observations, where both
instruments see similar ground scenes and air masses.

Figure 6 shows a typical example comparison for one leg. The two different types of observations have been processed as
explained previously, i.e the plume anomalies have been processed as described in Sect. 2.3 and the $CH_4$ fluxes have been
estimated as described in Sect. 2.4. The shown background normalised column anomalies agree well within their respective
errors inside and outside of the plume. Even more pronounced structures in the $CH_4$ concentration, as encountered on the right
hand side ($\sim 6$ to $15\,\mathrm{km}$ distance), are identified by both instruments. The fluxes from the two shown cross-sections deviate by
only $0.1\,\mathrm{t\,h^{-1}}$ or $1\,\%$.

More generally, when comparing fluxes estimated using measurements of MAMAP2DL with those derived from CHARM-F
observations from all flight legs (see Fig. E1), the averaged absolute difference between them is $\sim 1.2\,\mathrm{t\,h^{-1}}$ or $\sim 13\,\%$ excluding
the flight legs upwind and directly over the Pinto landfill (see Sect. 3.3 for reasoning). This difference is in part attributed
to different opening angles of the two instruments. The widths ($110\,\mathrm{m}$ vs. $10\,\mathrm{m}$), as we noted, are different. Consequently,
observed air masses are different. Typically, the errors of the fluxes are around or below $30\,\%$ of the respective flux and are
similar for MAMAP2DL and CHARM-F.

### 3.3   Derived Landfill Emission Rates

Using imaging MAMAP2DL observations, we also computed the fluxes within the different legs (see Sect. 2.4). The results are
summarised in Fig. 7, which also includes the cross-sectional fluxes derived from the CHARM-F instrument already computed
and introduced in Sect. 3.2.



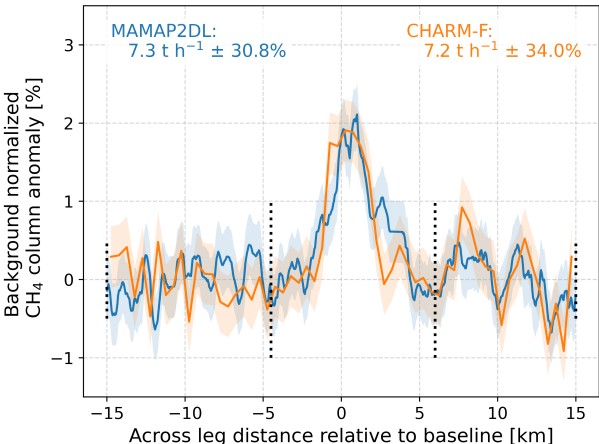

**Figure 6.** The background normalised $CH_4$ column anomalies for CHARM-F (orange) and the co-located MAMAP2DL (blue) observations for one flight leg collected between 11:42 and 11:46 UTC are shown. Vertical dotted lines separate the plume and background areas. Shaded areas represent the random error (single measurement precision) of the retrieved column anomalies of the respective instrument. The computed fluxes for the cross-sections according to Eq. 1 and the corresponding errors (MAMAP2DL: Eq. F1, and CHARM-F: Eq. F10) are given by the text insets. For graphical presentation only, the MAMAP2DL data has been smoothed by a $500\,\text{m}$ kernel to match the spatial resolution of CHARM-F in along flight direction. Flux, error and uncertainty range is, however, based on the $\sim 110\,\text{x}\,110\,\text{m}^2$ data.

Based on the MAMAP2DL observations, the fluxes exhibit a step-wise increase at the location of landfills as expected (from left - upwind, to right - downwind). The upwind leg at $-5\,\text{km}$ shows no significant inflow of enhanced $CH_4$ and a steep increase directly over the Pinto landfill. Between the Pinto and the VTP the flux or emission rate stabilises at $4.2\,\text{t}\,\text{h}^{-1}$ ($\pm 38\,\%$) before increasing to around $12.1\,\text{t}\,\text{h}^{-1}$ ($\pm 27\,\%$) on average at and after the Las Dehesas landfill. However, the cross-sectional
fluxes show some variability from flight leg to flight leg (see the bold horizontal coloured lines, representing averaged values over one MAMAP2DL leg) and variability within one leg (see the thin solid coloured lines). Adding the fluxes derived from the CHARM-F observations to the figure (coloured stars) reveals very good agreement between active and passive remote sensing (thin solid coloured lines) data as already indicated in Sect. 3.2. Computing average fluxes or emission rates from the CHARM-F observations alone yields $5.2\,\text{t}\,\text{h}^{-1}$ ($\pm 37\,\%$) for the Pinto landfill and $13.3\,\text{t}\,\text{h}^{-1}$ ($\pm 26\,\%$) for both waste treatment
areas combined.

For the averaged flux between the two landfills, the flight leg directly over the Pinto landfill (i.e. the green lines and star in Fig. 7) has been omitted. There, the plume might be still restricted to the surface and the wind speed is highly biased due to the strong vertical wind gradient (see Fig. D2, a). Over the Las Dehesas landfill, although there are new emissions emerging at the bottom, the plume from Pinto is assumed to be already well-mixed. Therefore, this leg is included in the flux average.



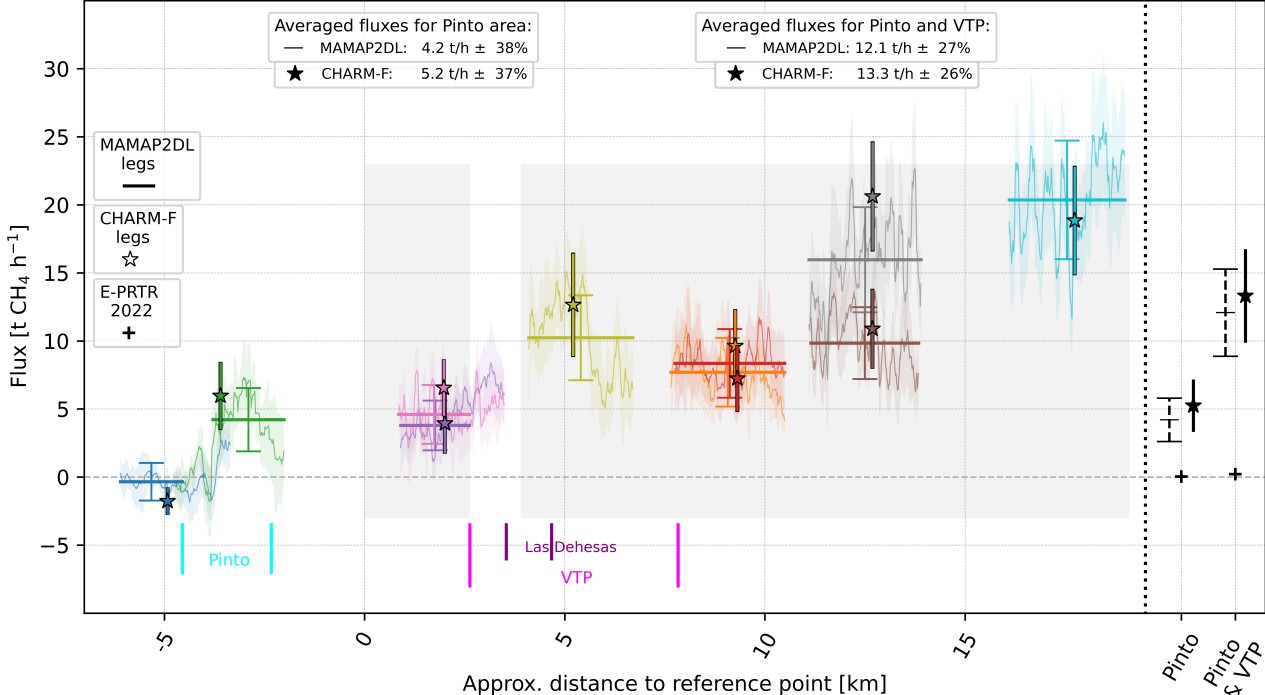

**Figure 7.** This plot shows the evolution of the $CH_4$ flux values upwind of the waste treatment areas (-5 km) to downwind (> 8 km). Cyan, purple and magenta vertical lines identify the locations of the two investigated waste treatment areas. The coloured thin solid lines are the values of the cross-sectional fluxes across the different MAMAP2DL legs and exhibit a high variability most likely due to atmospheric variability and turbulence on that day. Corresponding shaded coloured areas show the errors (estimated using Eq. F1). The averaged flux and error (Eq. F3) of one leg is given by the coloured bold horizontal lines and error bars, respectively. The averaged fluxes or emission rates and their errors (Eq. F6) estimated using MAMAP2DL observations for the two areas (in between Pinto and the Valdemingómez technology park, VTP, and in the lee of the VTP) are the black dashed lines in the right panel of the figure. Coloured stars and vertical bars give the fluxes and errors (Eq. F10) estimated using the CHARM-F measurements, respectively. Black stars and bars in the right panel are the averaged fluxes or emission rates and their errors (Eq. F11) over the same areas as for the MAMAP2DL observations. The two areas over which emission rates are computed are indicated by the grey shading. The pluses in the right panel indicate additionally the reported emissions for the Pinto area, and both the Pinto area and the VTP for the year 2022 assuming constant emission during the year.

## 3.4 Uncertainties Discussion

The estimation of errors or uncertainties is extensively discussed in the Appendix F and Table 1 lists the uncertainties for the different components, we assumed in our error analysis. Table 2 summarises the effect of the these components on the computed fluxes.





**Table 1.** Summary of relevant error sources used during the error analysis described in Appendix F. See Table F1 for further explanation of error sources.

| Parameter | Assumed uncertainty |
|---|---|
| $\delta F_u$ | $0.1\,\mathrm{ms}^{-1}$ |
| $\delta F_\alpha$ | $10°$ |
| $\delta F_{\mathrm{blh}}$ | $20\,\%$ on BLH |
| | translating to $\sim 0.8\,\mathrm{ms}^{-1}$ on wind speed |
| | (see Appendix D1) |
| $\delta F_{\mathrm{bg}}$ | up to $50\,\%$ variation of background area(s) |
| $\delta F_{\mathrm{col\text{-}pr}}$ | $\sim 0.4\,\%$ MAMAP2DL |
| | (see Sects. 2.2.1 and B2) |
| | $\sim 0.5\,\%$ CHARM-F |
| | (see Sect. 2.2.2) |
| $\delta F_{\mathrm{col\text{-}ac}}$ | $0.14\,\%$ (only MAMAP2DL) |
| | (see Sects. 2.2.1 and B2) |
| $\delta F_{\mathrm{col\text{-}cf}}$ | $1.2\,\%$ (only MAMAP2DL) |
| | (see Appendix B3) |

**Table 2.** Summary of computed error components for the averaged flux downwind of the two waste treatment areas according to Appendix F. Values are given as percentages of the respective downwind fluxes: $12.1\,\mathrm{t}\,\mathrm{h}^{-1}$ for MAMAP2DL and $13.1\,\mathrm{t}\,\mathrm{h}^{-1}$ for CHARM-F. 'X' stands for MAMAP2DL or CHARM-F according to nomenclature in Appendix F2 and F3, respectively.

| Parameter | MAMAP2DL [%] | CHARM-F [%] |
|---|---|---|
| $\delta F_u$ | 2 | 2 |
| $\delta F_\alpha$ | 5 | 5 |
| $\delta F_{\mathrm{blh}}$ | 20 | 20 |
| $\delta F_{\mathrm{col\text{-}cf}}$ | 1 | - |
| $\delta F_{\mathrm{X, atm, legs}}$ | 15 | 15 |
| $\delta F_{\mathrm{X, legs}}$ | 7 | 6 |
| components of $\delta F_{\mathrm{X, legs}}$ according to Eqs. F8 and F13 | | |
| $\delta F_{\mathrm{M2D, css}}$ | $< 1$ | - |
| $\delta F_{\mathrm{M2D, atm, css}}$ | 7 | - |
| $\delta F_{\mathrm{bg}}$ | 14 | 13 |
| $\delta F_{\mathrm{col\text{-}pr}}$ | - | 1 |



### 3.4.1 Individual Error Components

The uncertainties of our estimated fluxes are on the order of 25 to 40 % of the respective flux for the different spatial scales
(single cross-sections/legs or areal averages) and therefore quite similar for the different spatial scales. This is due to the fact
that the major error source, BLH ($\delta F_{\mathrm{blh}}$: $\sim 20\,\%$ error on the flux, see Table 2) consequently affecting the averaged wind speed
over the BL, is systematic and, therefore, cannot be reduced by averaging over several cross-sections or legs. The error on
the wind speed ($\delta F_u$: $\sim 2\,\%$) and wind direction ($\delta F_\alpha$ : $\sim 5\,\%$) itself, although also a systematic one, have only a limited
influence. The other two important error sources are plume distortions caused by atmospheric turbulence ($\delta F_{\mathrm{X, atm, legs}}$: $\sim 15\,\%$)
and the limits for the background area ($\delta F_{\mathrm{bg}}$: $\sim 14\,\%$). The latter is especially pronounced on the scales of legs as it reduces by
averaging over several legs. The column single measurement precision ($\delta F_{\mathrm{col-pr}}$: $< 1\,\%$) of the two instruments, or the remaining
systematic offset ($\delta F_{\mathrm{col-ac}}$: $< 2\,\%$) and the conversion factor error ($\delta F_{\mathrm{col-cf}}$: $< 2\,\%$) of MAMAP2DL lead to negligible errors on
the computed fluxes due to the relatively large spatial extent and large enhancements of the observed plume signals.

The major error source is the uncertainty in BLH, which has a significant influence on the averaged wind speed applied in the
flux computation. As stated above (Sect. 2.1.5 and Appendix D), we used atmospheric measurements of wind speed, direction,
and potential temperature collected during one ascent and one descent to validate and correct the ERA5 model estimates. Based
on the two measured profiles and the overestimation of the BLH in ERA5 compared to these profiles, we apply a correction
reducing the ERA5 BLH by $\sim 17\,\%$ on average. We assume that this correction is also applicable to ERA5 data up to 2 hours
earlier when the remote sensing measurements started. Due to the strong vertical gradient in wind speed, this also reduces the
averaged wind speed by $\sim 24\,\%$ and leads to the same relative reduction in the fluxes. The uncertainty of the BLH estimates
itself is 20 %, which consequently translates into an wind speed error of $0.8\,\mathrm{m\,s^{-1}}$ on average.

Additionally, to estimate the accuracy of the ERA5 wind data, we compare it to the BAHAMAS measurements. The averaged
deviations are only $0.05\,\mathrm{m\,s^{-1}}$ and $0.8°$. Therefore, we assume that the error on the modelled wind speed within the BL is
$0.1\,\mathrm{m\,s^{-1}}$. For the wind direction, we compare the modelled one with the visually observed plumes in Google Earth imagery
and concluded an uncertainty of $\sim 10°$.

Other important error sources are the limits for the background area and plume distortions caused by atmospheric turbulence.
Depending on the spatial scale, they are reduced by averaging the estimated $CH_4$ fluxes from multiple cross-sections. For
example, the effect of the atmospheric variability reduces if more independent legs or cross-sections are collected (either
spatially or temporally separated). This variability is quantified by the standard deviation over all legs of one area where a
constant flux is expected, or over the individual cross-sectional fluxes within one leg. We assume that the fluxes are independent
for different legs as they are recorded at different times and/or locations but have a correlation length of around $\sim 400\,\mathrm{m}$ within
one MAMAP2DL leg resulting in 7 independent fluxes across one leg.

Even if correlation between all fluxes of one leg is assumed, the relative error on the averaged downwind flux would only
increase from 27 % to 28 %. The effect would be slightly larger for the averaged flux between the two landfills (38 % vs 45 %)
and for single legs (7 % vs. 18 %). The errors are still dominated by the systematic wind errors. As we use the standard deviation





to quantify the variability, it might also be partly influenced by measurement error and the error introduced by the background normalisation.

For MAMAP2DL, also the uncertainty from the conversion factor related to the magnitude and change of the BLH during the
flight time is a systematic error source and scales with the retrieved anomalies. This is not reduced by multiple cross-sections or legs and has the same influence on the cross-sectional fluxes within one MAMAP2DL leg as well as on the averaged total flux from the two waste treatment areas.

### 3.4.2 Potential Additional Sources

The most downwind leg in Fig. 7 (cyan leg) shows a high variability in the computed fluxes from the MAMAP2DL observations
across the first two-third of the leg. Whereas the last third, which is also located downwind of the position of the CHARM-F observation, shows a consistently more stable and higher flux. This might be related to potential additional $CH_4$ emissions from an industrial area located there (40.433°N, 3.491°W), which also includes a 'Planta de Combustible' (fuel plant), several storage tanks, and a waste-water treatment plant. Excluding this latter part of the MAMAP2DL leg would reduce the mean flux of this MAMAP2DL leg from 20.4 to $19.1\,\mathrm{t\,h^{-1}}$ and the average over the entire area, however, only by $0.1\,\mathrm{t\,h^{-1}}$. Furthermore,
there are additional urban waste-water treatment plants distributed in the measurement area according to E-PRTR. While these could produce and emit $CH_4$, none stands out in our column observations.

### 3.4.3 Potential Plume Accumulation Effects

As discussed in Appendix D1 and shown in Fig. D2 (b), before the start of the remote sensing part of the flight at 11:00 UTC, the wind direction changed from around 130° to 210°. Additionally, during the entire flight time, there was a very strong vertical
wind gradient with $\sim 1\,\mathrm{m\,s^{-1}}$ at the ground and up to $\sim 10\,\mathrm{m\,s^{-1}}$ at the top of the BL (Fig. D2, a). Especially the turn in wind direction directly before our measurement started could, potentially, have created an area with enhanced $CH_4$ concentrations due to accumulation (a '$CH_4$ puff'), which would have subsequently been advected in wind direction. Surveying such a puff would also lead to increased fluxes.

The grey and cyan leg in Fig. 7 would indicate these enhanced fluxes compared to the remaining legs. Assuming that
during the time of the remote sensing measurement, a mean wind speed of $\sim 3.9\,\mathrm{m\,s^{-1}}$ prevailed, and that these legs were acquired around 90 minutes after the start at 11:00 UTC, would lead to a travel distance of $21\,\mathrm{km}$ of the observed air masses. $21\,\mathrm{km}$ would roughly correspond to the southern part of the Pinto waste treatment area. Excluding the two legs from the downwind average would lead to a mean flux of $9.6\,\mathrm{t\,h^{-1}}$ instead of $12.7\,\mathrm{t\,h^{-1}}$. Additionally, the change in wind direction also caused some residual plume structures over the city of Madrid. A potential influence of this residual plume on our background
determination is covered by the respective error $\delta F_{\mathrm{bg}}$.

To investigate these effects further and to verify our assumption of $CH_4$ accumulation would, however, require more sophisticated model simulations and is not possible with a simple and fast mass balance approach. Applying model-inversion based flux-estimation methods is beyond the scope of this publication, but will be addressed in a follow-up paper that is currently in preparation.





## 4 Comparison of emission rates determined in this study with other estimates

The waste treatment areas Pinto and VTP have reported emission rates of $0.35\,\mathrm{kt\,yr^{-1}}$ (or $0.04\,\mathrm{t\,h^{-1}}$ assuming constant emissions throughout the year) and $1.58\,\mathrm{kt\,yr^{-1}}$ (or $0.17\,\mathrm{t\,h^{-1}}$) in E-PRTR for the year 2022, respectively. Our observations were collected within 2 hours on the 4th August 2022. This represents a snapshot of estimated emission rates and they should not be lightly extrapolated to annual averages. Landfill emissions usually exhibit some temporal variability and are modulated by, e.g., emissions caused by leakages, activities across the landfill when waste is deposited, atmospheric parameters such as pressure changes, temperature, wind speed, or temperature and humidity conditions within the landfill (e.g., Cusworth et al., 2024; Kissas et al., 2022; Delkash et al., 2016; Xu et al., 2014; Trapani et al., 2013; Poulsen and Moldrup, 2006).

However, over the past years, other studies using observations of a limited period derived similar emission rate estimates as observed by us. The most recent is the webstory from ESA (2021) using satellite data from TROPOMI and GHGSat in August and October 2021. They reported total emission rates of $8.8\,\mathrm{t\,h^{-1}}$ with one of the sources emitting $5.0\,\mathrm{t\,h^{-1}}$, without mentioning landfill names. However, in the GHGSat images on the website, the Pinto and Las Dehesas landfills are identified as part of their target area. Although these estimates are from the preceding year, partly from the same season, they agree well with our results. Additionally, based on the available imagery, one main plume appears to originate, at least partly, from the already closed and covered area of the Las Dehesas landfill. Although we cannot exclude outgassing from closed parts of this landfill, our $CH_4$ hotspots are predominantly located over the active areas of the landfills.

In 2018, another study used ground-based and satellite observations to also estimate emissions of Madrid's landfills (Tu et al., 2022). Their ground-based observations were collected between the end of September and beginning of October 2018 and their resulting flux is $\sim 3.5\,\mathrm{t\,h^{-1}}$. This flux was assigned to the Valdemingómez waste plant. Satellite data were analysed over the period May 2018 to December 2020. Estimated emission rates are $7.1\,\mathrm{t\,h^{-1}}$ ($\pm 0.6\,\mathrm{t\,h^{-1}}$) for the entire area.

A ground-based investigation in that area was undertaken from the 1st to 3rd March in 2016. Sánchez et al. (2019) used specifically designed flux chambers to measure $CH_4$ emission from the already full and closed parts (or cells) of the Las Dehesas landfill north of the still active area. They have estimated $1.1\,\mathrm{t\,h^{-1}}$ on average for this part which accounts for approximately half of the total designated landfill area of $\sim 0.6\,\mathrm{km^2}$. The values for the 95 % confidence interval are given with 0.4 to $2.8\,\mathrm{t\,h^{-1}}$. Their averaged value would correspond to around 9 % of our total emission rate, however, derived for the entire area also including the Pinto landfill.

Over the past years, all these estimates indicate consistently high emission rates of up to 7 to $9\,\mathrm{t\,h^{-1}}$ for both waste treatment areas, although they are made over short periods (with the exception of the estimates using satellite observations[5]). Our estimated emission rate for the two areas are at the upper end of this range ($12.7\,\mathrm{t\,h^{-1}}$ or $9.3\,\mathrm{t\,h^{-1}}$ if the $CH_4$ puff hypothesis is applicable) and also indicates disagreement with the reported values in E-PRTR (see Table 3).

The locations of high human activity and waste deposition correlate with the highest observed column concentrations. We infer that these locations on the landfill are the main origin of our observed emissions. These active areas were also identified by Cusworth et al. (2024) as $CH_4$ emission sources. However, it is unclear whether these emission hot spots exist only during

---

[5]But especially the TROPOMI satellite data have issues to allocated sources precisely due to the large footprint of $7.0\,\mathrm{x}\,5.5\,\mathrm{km^2}$.



**Table 3.** Reported $CH_4$ emission rates in E-PRTR for the Pinto area and the VTP in $t\,h^{-1}$ assuming constant emissions throughout the year.

| Year | 2016 | 2017 | 2018 | 2019 | 2020 | 2021 | 2022 |
|------|------|------|------|------|------|------|------|
| Pinto | 1.10 | 1.19 | 1.28 | 1.37 | 1.47 | 1.55 | 0.04 |
| VTP | 1.02 | 0.70 | 0.82 | 0.94 | 0.29 | 0.33 | 0.18 |

the day, when work is done on the landfill, or also at night. The degree of correlation between emissions and activity is unclear and these emissions should actually cease when a cell is completed and closed. Local process-based bottom-up modelling of
emissions of waste deposition is challenging due to the unpredictability of exact locations and practices. This may explain some of the discrepancies between the inventory and the top-down estimates (Balasus et al., 2024).

## 5 Summary and Conclusion

The reduction of anthropogenic $CH_4$ emissions has been proposed as target for climate mitigation strategies, due to $CH_4$'s relatively short tropospheric life time. In spite of this objective, knowledge of the $CH_4$ emissions from many anthropogenic
sources and in particular landfills, even though these emissions account for a significant fraction of the global anthropogenic $CH_4$ budget, are still uncertain. Relevant examples are the recent discussions of the emissions from landfill sites in Madrid, the capital of Spain. Exceptionally high $CH_4$ emission rates have been reported using both ground based and satellite borne observations in the year 2021 and before.

To examine this $CH_4$ source and to estimate its emissions, we undertook a measurement flight on the 4[th] August 2022 as
part of the CoMet 2.0 mission. In this study, for the first time, the passive imaging MAMAP2DL and active lidar CHARM-F remote sensing instruments flew aboard the same platform, the German research aircraft HALO, and successful co-located and independent measurements were made. During the first part of the flight, remote sensing column observations were acquired. MAMAP2DL collected 28 ground scenes having a spatial resolution of $\sim$110 x 110 $m^2$ within a $\sim$3 km swath for a flight altitude of 7.7 km a.g.l.. CHARM-F recorded ground tracks with a spatial resolution of $\sim$500 m in flight direction, due to
averaging, and $\sim$10 m across.

In total, 10 flight legs, aligned perpendicular to the prevailing wind direction, were flown at several distances up- and downwind of the two waste treatment areas Pinto and VTP including the Las Dehesas landfill. Exploiting the design of the flight plan, emissions from the two landfill sites were separated and estimated by combining the retrieved $CH_4$ column anomalies with model wind data from ECMWF ERA5. Additionally, from the overflights above the landfill areas in combination with
$CH_4$ imaging data, potential source locations on the landfills were identified.

The BL was physically characterised by the measurements of vertical atmospheric profiles of meteorological parameters and trace gases within the BL. This supported our analysis of the remote sensing data and was used to validate the ERA5 model data for that day. As the remote sensing data was acquired well above the BL, we relied on models for (wind) data within the BL.





The emissions from the two landfill sites were well separated by the two remote sensing instruments with an observed emission rate of $\sim 5\,\mathrm{t\,h^{-1}}$ for the Pinto area, while the combined emission rate of Pinto and VTP was $\sim 13\,\mathrm{t\,h^{-1}}$. The error on these $CH_4$ emission rate estimates are around 26 to $38\,\%$ of the given fluxes (or 1.9 to $3.5\,\mathrm{t\,h^{-1}}$) and are dominated by the knowledge of the BLH in combination with a strong vertical wind gradient and the separation between plume and background areas. Moreover, the measured fluxes and emission rates are influenced by atmospheric turbulence. This results in the flux

variation in different legs expressed as standard deviation over all legs in the downwind area of up to $\sim 5\,\mathrm{t\,h^{-1}}$. We conclude that a sufficient number of independent flight legs are required to minimise the error from turbulent flow in the estimation of the the fluxes from observed plumes.

This was the first time that emissions were observed and quantified by two different and independent remote sensing techniques. The comparison of fluxes retrieved using the measurements of the active and passive remote sensing instruments shows

that the two estimates are in very good agreement. To ensure comparability of the flux estimation using the different remote sensing approaches, we also used identical wind speeds for individual legs. Absolute differences are $13\,\%$ of the respective fluxes on average. These differences may be explained by the different ground scene sizes observed by the two instruments, which are $10\,\mathrm{m}$ and $110\,\mathrm{m}$ for CHARM-F and MAMAP2DL in across flight direction. Consequently, they measure different but overlapping air masses in the plume. The agreement between the two different techniques also increases our confidence

that the emission rates are as high as our estimates. The complementarity of the active and passive instruments shows good prospects for their joint deployment also on spaceborne platforms.

For source attribution, the imaging data of the MAMAP2DL instrument was utilised. The determination of the exact source location is limited by a combination of the ground scene size of $\sim 110\,\mathrm{x}\,110\,\mathrm{m^2}$ and the accuracy of the orthorectification process itself and has been estimated to be better than $110\,\mathrm{m}$. Highest column enhancements, identifying the origin of the

emissions, were observed over active parts of the landfills, where the garbage is deposited, towards the south-east for Las Dehesas and in the eastern part of Pinto. In the same regions CHARM-F observes the largest column enhancements. This implies significant emissions from areas which are not yet managed during nominal operations but probably also not sufficiently covered by the reporting. Nevertheless, the question remains about night time and weekend emissions, when there is less or no activity on the landfill.

A crucial parameter for the estimation of emission rates is the wind speed, which is particular challenging to determine for remote sensing instruments as they typically fly above the plumes and the BL. Here, we used modelled ERA5 data, which were validated by airborne measurements within the BL. On average, wind speed and direction disagree by only $0.05\,\mathrm{m\,s^{-1}}$ and $0.8°$, respectively.

However, larger deviations occurred for the BLH in ERA5, which was consistently lower in the comparison of ERA5 to

the two measured profiles. Correcting for this discrepancy led to a decrease in the average wind speed used for the cross-sectional fluxes of $\sim 24\,\%$ due to the strong vertical wind gradient (present in both ERA5 model data and BAHAMAS wind measurements). This reduction in wind speed directly changes proportionately the estimated emission rates.

Our analysis shows the importance of knowledge and understanding of the characteristics of the BL during a measurement. While we had the privilege to compare in situ wind measurements with model data, even though at a later time of the day,



emission estimates based on satellite data rely on atmospheric parameters from models. Moreover, there is usually no possibility
to validate the conditions during measurement times. Systematic errors such as the BLH in combination with the strong vertical
wind speed gradient, influences the estimated emission rates. They need to be identified and taken into account to minimise
their impact.

Our calculated emission rates are in good agreement with previous top-down estimates, even though, strictly speaking, they
are only valid for the time of the overflight. The prevailing winds in combination with the vertical distribution of the $CH_4$
emissions in the BL could introduce a common error in our emission rate estimate but not to an extent that we approach
reported values assuming constant emission throughout the year, at least on the 4[th] August 2022 during our flight. The fact
that our emission estimates are a factor of 40 to 50 higher than reported values (assuming constant emissions) supports the
inference that a major part of the emissions are unreported, especially as the reported emissions in E-PRTR fell by a factor of
10 from 2021 to 2022.

An additional analysis is currently being studied. This makes use of a transport model to constrain the influence of the
(changing) wind field and the (vertical) mixing of the $CH_4$ plume during the measurement flight. We consider the use of
a transport model will resolve some of the issues encountered when the direction of the wind changes i.e. residual plume
structures over the city of Madrid and potential $CH_4$ accumulations. These are difficult to account for using the simple cross-
sectional mass balance approach.

*Data availability.* The MAMAP2DL $CH_4$ column anomalies and the CHARM-F observations are available from the authors upon request.
The HALO BAHAMAS in situ dataset (including SHARC and JIG) can be directly inquired from the authors or can be downloaded from the
HALO database (https://halo-db.pa.op.dlr.de/, Deutsches Zentrum für Luft- und Raumfahrt, 2024). The ECMWF ERA5 data can be directly
inquired from the authors.

**Appendix A: Further Flight Legs**

Figure A1 supplements Fig. 5 with two additional flight legs, which were flown in along wind direction. Therefore, they were
not used for any flux estimates. However, they reveal further insights into possible source regions.

The flight leg in (a) was acquired at the same flight altitude as the legs shown in Fig. 5. The leg shown in (b), on the other
hand, was collected after the in situ part of the flight at around 13:34 UTC at a flight altitude of $\sim 1.6\,\mathrm{km}$ a.g.l.. The reduced
flight altitude also reduced the swath width of the MAMAP2DL imaging data from $\sim 3\,\mathrm{km}$ to $700\,\mathrm{m}$ and also the ground scene
size from $\sim 110\,\mathrm{x}\,110\,\mathrm{m}^2$ to $24\,\mathrm{x}\,24\,\mathrm{m}^2$.

Interestingly, in the lower flight leg (b), $CH_4$ enhancements are observed at similar positions as in the leg flown at higher
altitudes (a) and in the perpendicular legs in Fig. 5 (a) for the Pinto landfill in the south. However, no enhancements are
visible across the VTP Fig. A1 (b). This is in-line with the legs flown perpendicular to the wind direction, in which the highest
anomalies were observed in the south-eastern part of the Las Dehesas landfill, however, not covered by the low flying leg. As




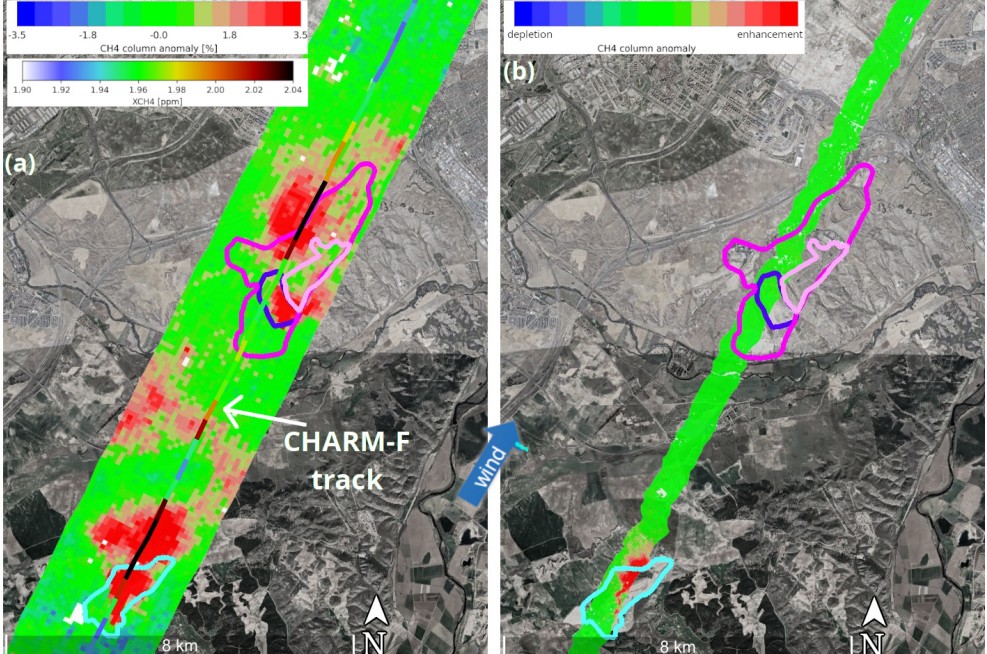

**Figure A1.** Similar to Fig. 5 but for the along wind legs at flight altitudes of $\sim 7.7$ km a.g.l. **(a)** and $\sim 1.6$ km a.g.l. **(b)**. In **(a)**, the retrieved $CH_4$ column anomalies from MAMAP2DL data are overlaid by the $XCH_4$ from CHARM-F data. There is no CHARM-F data available for low flying altitudes due to saturation of the detectors. The map underneath is provided by Google Earth (Image © Landsat/Copernicus, Maxar Technologies).

the flight leg, acquired at lower flight altitude (b), was within the BL and, therefore, within the plume, caution needs to be taken with an quantitative interpretation of the shown MAMAP2DL column anomalies.

## Appendix B: MAMAP2DL Retrieval

### B1 WFMD-DOAS

The WFM-DOAS retrieval has been extensively described in other publications (Krings et al., 2011; Krautwurst et al., 2021) and thus, we focus here on the aspects, which are important for the quality of the retrieved $CH_4$ column anomalies. The core of the retrieval is based on radiative transfer model (RTM) simulations (in our case with SCIATRAN v3.8, Mei et al., 2023; Rozanov et al., 2014) of radiances, which describe the general state of the atmosphere at the time of the measurement flight to our best knowledge. Differences between the modelled radiance and the measured radiance are described by fitting

weighting functions[6] to the model and minimising the difference between measurement and modified model. An example for

---

[6]Here, a weighting function describes the change of radiance due to a change of one parameter and must not be confused with CHARM-F's weighting function used to describe its altitude sensitivity, Appendix C1.



such differences are deeper absorption lines due to enhanced $CH_4$ from an emission plume in the atmosphere. The resulting fit factors are called profile scaling factors (PSFs) and are representative for the observed atmospheric $CH_4$ and $CO_2$ columns. The weighting functions, one for each fit parameter (in our case for $CH_4$, $CO_2$, $H_2O$ and temperature), describe the change of radiance due to a change of one of the listed parameters. Furthermore, we apply a 1D look-up table approach for the topography
to account for strong variations in surface elevation during the retrieval process.

**Table B1.** Boundary conditions for the radiative transfer model simulations (RTMs) for the remote sensing (RS) part at $\sim 7.7\,\mathrm{km}$ a.g.l. of the flight over Madrid.

| Flight day | 04.08.2022 |
|---|---|
| Time for RS pattern (local time, UTC) | |
| start [hh:mm] | 13:00, 11:00 |
| end [hh:mm] | 14:40, 12:40 |
| Mean solar zenith angle (sza) [a)] [°] | 25.8 |
| Flight altitude [b)] [m a.s.l.] | 8371 |
| Surface elevation along flight track [c)] | |
| min [m a.s.l.] | 441 |
| max [m a.s.l.] | 1026 |
| Mean column mole fractions [d)] | |
| $CH_4$ [ppb] | 1876 |
| $CO_2$ [ppm] | 417.0 |
| $H_2O$ [ppm] | 4127 |
| Aerosol scenario [e)] [−] | urban |
| Albedo [f)] [−] | 0.30 |

[a),b),c),e),f)] are estimated similarly to Krautwurst et al. (2021).

[d)]The vertical atmospheric profiles are taken from the U.S. standard atmosphere (USCESA, 1976), which are adapted to and replaced by the in situ observations collected by BAHAMAS ($H_2O$, temperature, pressure) and JIG ($CH_4$, $CO_2$) for the measurement flight at altitudes between $\sim 1.6$ and $7.7\,\mathrm{km}$ a.g.l..

To represent the atmosphere by the modelled radiances as realistically as possible, vertical concentration profiles of the gases ($CH_4$, $CO_2$, $H_2O$), pressure and temperature are needed. The model takes the properties of the reflecting surface into account. Multiple scattering by aerosols in the atmosphere is considered. Finally, geometrical factors such as flight altitude, surface elevation and solar zenith angle (sza) are included in the calculations (see Table B1 for details on the parameters used). Fig. B1
shows one example fit for the two fit windows which we use in this study. These are 1590.0 to 1635.0 nm for $CO_2$ and 1625.0 to 1672.5 nm for $CH_4$.

Then, the $CH_4$ column anomalies are computed from the retrieved PSFs as follows:





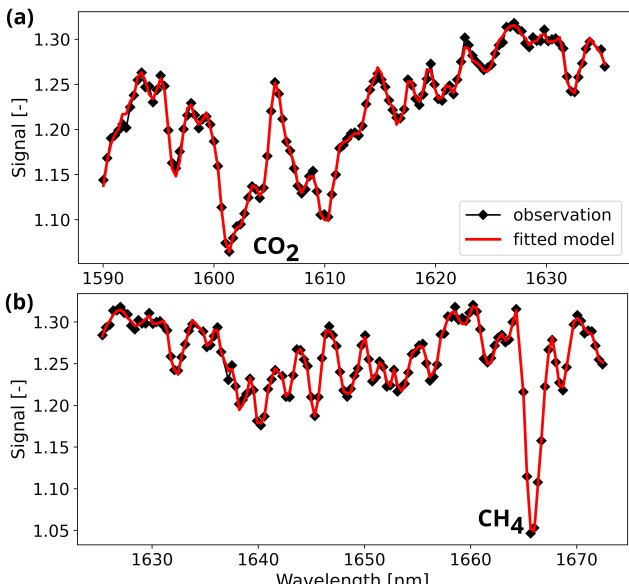

**Figure B1.** Example fits for the two fit windows used in the WFM-DOAS retrieval: **(a)** $CO_2$ and **(b)** $CH_4$ fit window. The black diamonds denote the measurement and the red solid line the fitted model.

$$\Delta V_{CH_4} = \left( \frac{PSF_{ratio}}{\overline{PSF_{ratio}}} - 1 \right) \cdot CH_4^{abs\,col} \cdot k \tag{B1}$$

where

$$PSF_{ratio} = \frac{PSF_{CH_4}}{PSF_{CO_2}}, \tag{B2}$$

where $PSF_{ratio}$ is the unitless ratio of the two retrieved PSFs for $CH_4$ and $CO_2$, which is also called the proxy method (Krings et al., 2011, 2013), $\Delta V_{CH_4}$ is the $CH_4$ column anomaly in $molec\,cm^{-2}$ (for visualisation purposes displayed as % relative to the given background column), k is a unitless conversion factor (see Appendix B3), $CH_4^{abs\,col}$ is the assumed background column of $CH_4$ in $molec\,cm^{-2}$ as used in the RTM simulation, and $\overline{PSF_{ratio}}$ denotes a normalisation process using observations

from the local background (see Sect. 2.4).

    As an example, a PSF for $CH_4$ ($PSF_{CH_4}$) of larger than 1 would, compared to the modelled background radiances, indicate more $CH_4$ in the measured spectrum (and thus in the atmosphere) due to an emission plume and vice versa. However, the absorption depth in a measurement is not only determined by the amount of gases but also by other effects like variations in sza, surface elevation, flight altitude or aerosol composition, which might not be modelled accurately. These effects lead to

a light path error, which affects the PSFs of $CH_4$ and $CO_2$ in a similar way due to their spectral proximity. Using the proxy method as indicated in Eq. B2, which is only possible if there are no major variation in the atmospheric $CO_2$ concentration field expected, reduces these apparent column variations due to light path errors significantly (Krings et al., 2013, 2011). Another





important step is the normalisation by the local background, indicated by $\overline{\mathrm{PSF_{ratio}}}$, and the consideration of the conversion factor, before obtaining the $CH_4$ column anomalies in $\mathrm{molec\,cm^{-2}}$ (or better suited for visualisation purposes in $\%$) used during the cross-sectional flux method (Sect. 2.4).

## B2  Errors of the $CH_4$ Column Anomalies

Previous sensitivity studies have shown (Borchardt et al., 2021; Krautwurst et al., 2021; Krings et al., 2011) that false assumptions in the input parameters for the RTM simulations can cause significant systematic errors in the retrieved single columns or the PSFs of $CH_4$ and $CO_2$. Most of these systematic errors are related to light path errors and are thus significantly reduced by the proxy method as described in Sect. B1. Remaining systemic errors (e.g., a changing sza, or a constant offset caused by an inaccurate $CO_2$ profile, temperature or aerosol profiles) vary either smoothly with time or are approximately constant over the measurement area. These effects are therefore corrected by the additional normalisation with observations outside of a plume. Exceptions could be changes in surface elevation and surface spectral reflectance, which can occur pseudo-random and over short distances (see Krautwurst et al., 2021, for details). Additionally, potentially co-emitted $CO_2$ from landfills affect the proxy method. Krautwurst et al. (2017) have shown that the reduction on the $CH_4$ anomalies could be around $5\,\%$ on average. This effect is however, not considered further here. The sensitivity of the retrieval to parameters for the Madrid flight are summarised in Table B2 and could potentially lead to a remaining systematic offset of the retrieved $CH_4$ column anomalies of around $0.14\,\%$ after correction (Krings et al., 2011) by the conversion factor (0.763) resulting from the basic scenario used in Table B2.

**Table B2.** Sensitivity of the retrieved PSFs to the input parameters for the RTM simulations according to expected variations during the flight on the 4[th] August 2022. The deviations for the PSFs of $CH_4$, $CO_2$ and the ratio $CH_4$ over $CO_2$ are given relative to the background column. The parameters for the basic scenario used during the retrieval are given in Table B1 using a surface elevation of $0.734\,\mathrm{km}$. Not all values deviate symmetrically around $0\,\%$, therefore, the worst case scenario is always selected.

| variation in parameter | Expected deviation of PSF [%] | | |
|---|---|---|---|
| | $CH_4$ | $CO_2$ | ratio |
| Solar zenith angle [$\pm\,3^\circ$] | $\pm\,1.31$ | $\pm\,1.29$ | $\pm\,0.02$ |
| Surface elevation [$\pm\,50\,\mathrm{m}$] | $\pm\,0.83$ | $\pm\,0.93$ | $\pm\,0.10$ |
| Flight altitude [$\pm\,5\,\mathrm{m}$] | $\pm\,0.02$ | $\pm\,0.01$ | $\pm\,0.01$ |
| Aerosol [desert, background] | $\pm\,0.12$ | $\pm\,0.32$ | $\pm\,0.21$ |
| Albedo [$0.1-0.50$] | $\pm\,0.98$ | $\pm\,1.14$ | $\pm\,0.16$ |
| $H_2O$ [$\pm\,50\,\%$] | $\pm\,0.01$ | $\pm\,0.00$ | $\pm\,0.01$ |
| $CO_2$ [$\pm\,1\,\%$] | $\pm\,0.00$ | $\pm\,1.00$ | $\mp\,1.00$ |
| $CH_4$ [$\pm\,1\,\%$] | $\pm\,1.00$ | $\pm\,0.00$ | $\pm\,1.00$ |
| Temperature [$\pm\,5^\circ\mathrm{C}$] | $\pm\,1.60$ | $\pm\,1.80$ | $\pm\,0.21$ |





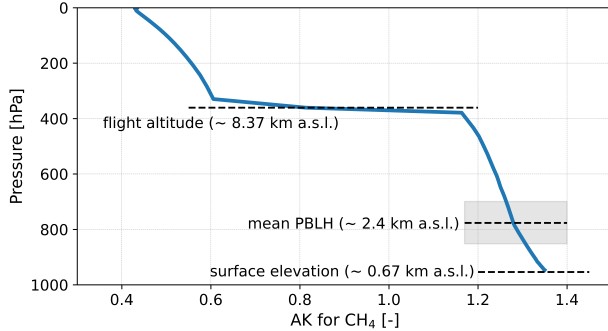

**Figure B2.** Averaging kernel ($AK$) of $CH_4$ for the atmospheric and geometrical conditions as encountered during the Madrid flight for MAMAP2DL. The step in the profile at $8.37\,km$ a.s.l. (or $7.7\,km$ a.g.l.) marks the flight level of the aircraft. The shaded area of the BLH represents the growth and uncertainty of the BLH during the remote sensing flight according to ERA5 and measured vertical profiles (also see Appendix D2).

In addition to the systematic effects described above, random effects like measurement noise produce random column errors. They are not separated further and estimated together as the single-measurement precision, which is directly computed from the retrieved $CH_4$ column anomalies outside of the plume (as, e.g., done in McLinden et al., 2024; Chulakadabba et al., 2023; Borchardt et al., 2021; Krautwurst et al., 2021; Krings et al., 2013). Moreover, the measurement precision can cover some of the remaining small scale systematic offsets. For Madrid's landfills flight, it has been estimated to be $\sim 0.4\,\%$. This is slightly
worse than that of its precursor instrument MAMAP (see Sect. 2.1.3) and possibly related to the coarser spectral sampling ($\sim 3$ to 4 pixels vs. $\sim 10$ pixels). However, this is compensated by simultaneously acquiring 28 observation in across flight direction for a swath width of $\sim 3\,km$ at a flight altitude of $\sim 7.7\,km$ a.g.l..

**B3    Averaging Kernels and Conversion Factor**

An important parameter of the WFM-DOAS retrieval applied to MAMAP2DL observations is the so-called averaging kernel
($AK$). It describes the sensitivity of the retrieval to $CH_4$ column changes in different altitude levels. It is computed by retrieving the $CH_4$ column from simulated measurements in which the $CH_4$ concentration at various altitude levels has been perturbed. Figure B2 shows the $AK$ based on a RTM simulation mimicking the atmospheric and geometrical condition during the flight over Madrid as used in Sect. B1 for analysis of the MAMAP2DL observations.

An $AK$ of unity at a certain altitude or pressure level would indicate that the retrieval is able to retrieve the perturbed
$CH_4$ concentration correctly. However, due to the measurement geometry the retrieval overestimates $CH_4$ changes below the aircraft close to the surface. This effect is related to the light path in the atmosphere. An idealised light beam covers the air masses below the aircraft twice before reaching the sensor leading to apparent enhanced absorption by $CH_4$. This effect must be corrected for, otherwise, the retrieval would overestimate potential enhancements below the aircraft.





Assuming that observed variations in the $CH_4$ column originate from air masses inside the BL, a conversion factor $k$ is computed as mean of the averaging kernels $\overline{AK}_{below}$ within the BL and then applied in Eq. B1 for the computation of the column anomaly:

$$k = \frac{1}{\overline{AK}_{below}}. \tag{B3}$$

The altitude resolved $AK(z)$ is defined as variation of the retrieved total $CH_4$ column $\partial V_{\text{retrieved}}$ as a result of a perturbation of the true $CH_4$ subcolumn $\partial v_{\text{true}}(z_j)$ at altitude $z_j$ (Krings et al., 2011):

$$AK(z_j) = \frac{\partial V_{\text{retrieved}}}{\partial v_{\text{true}}(z_j)}. \tag{B4}$$

The conversion factor $k$ for the Madrid flight is 0.763 for an average BLH of $2.4\,\text{km a.s.l.}$ as encountered during the remote sensing flight. Computing the BLH as described in Appendix D2 indicates an increase in the BLH at the position and time of the different flight legs from approx. 1.9 to $2.7\,\text{km a.s.l.}$. Considering additionally the given uncertainty of the BLH estimate would lead to an error of the estimated conversion factor of $1.2\,\%$.

## B4 Orthorectification

In order to correctly deduce source positions of the $CH_4$ emission plumes across the two landfills, the $CH_4$ anomaly maps from MAMAP2DL have to be accurately projected on the ground. Important parameters for this procedure are the attitude information of the aircraft: pitch, roll, and yaw, which define the line of sight of the instrument. Moreover, the aircraft's flight altitude and the surface elevation at the position of the projected ground scene, in combination with the viewing angle of the instrument, determine the across leg ground scene size and, thus, the width of the entire swath and, finally, the aircraft position itself. The position and attitude data (pitch, roll, yaw, flight altitude, location) are provided by the BAHAMAS system introduced in Sect. 2.1.5 at a resolution of 10 Hz. Topography data is derived from the SRTM (Shuttle Radar Topography Mission, Farr et al., 2007) digital elevation model (DEM) also used for the estimate of the surface elevation applied in the RTMs in Appendix B1. Next, the orthorectification is performed along the lines as described in Schoenhardt et al. (2015).

As main RGB imagery source, we use Google Earth data, overlaid with the $CH_4$ column anomaly maps (as kmz-files, see Fig. 5). Therefore, the accuracy of the orthorectification is validated against Google Earth imagery. Coincidentally, at least some of the Google Earth images of Madrid were taken in August 2022, which is in the same month as the measurements were acquired. Therefore, we assume that the state of the landfill during the overflight on the 4th August is very similar to that shown in the Google Earth images. For the validation process, we use high-resolution (not spatially binned) intensity maps of the measured on chip signal strength around $1.6\,\mu\text{m}$ with a spatial resolution of $\sim 8.5\,\text{m}$ in along flight direction as they mimic to a certain degree the surface properties and structures seen in the Google Earth RBG maps. The deviations (or better gradients) of distinct features such as rivers or streets are then used to verify the accuracy of the orthorectification. In along flight direction the accuracy is better than $\sim 20\,\text{m}$, whereas in across flight direction, it is determined by the coarse spatial



resolution of $\sim 110\,\mathrm{m}$. Overall, we estimate an accuracy of the column anomaly maps of better than $110\,\mathrm{m}$, or around one
695  ground scene, for the Madrid dataset, in generally limited by the final ground scene size.

## Appendix C: CHARM-F Retrieval

### C1  Retrieval and Weighting Function

The quantity that is independently measured by CHARM-F is the differential absorption optical depth (DAOD, $\Delta\tau$), calculated
from the received signals by

$$\Delta\tau = \frac{1}{2}\ln\left(\frac{S_{\mathrm{off}}/E_{\mathrm{off}}}{S_{\mathrm{on}}/E_{\mathrm{on}}}\right). \tag{C1}$$

$S_{\mathrm{on}}$, $S_{\mathrm{off}}$ are backscattered signal and $E_{\mathrm{on}}$, $E_{\mathrm{off}}$ are internal energy reference measurement at the on- and offline wavelength,
respectively. $\Delta\tau$ can on the other hand also be described in terms of the molecular absorption cross-section at both wavelengths
($\sigma_{\mathrm{on}}$ and $\sigma_{\mathrm{off}}$) and the number density $n_{\mathrm{CH_4}}$ or the dry-air mixing ratio of methane $r_{\mathrm{CH_4}}$.

$$\Delta\tau = \int_{h_0}^{h_1} n_{\mathrm{CH_4}} \underbrace{(\sigma_{\mathrm{on}} - \sigma_{\mathrm{off}})}_{=\Delta\sigma} \ \mathrm{d}h, \tag{C2}$$

$$= \int_{h_0}^{h_1} \frac{r_{\mathrm{CH_4}}}{(1+r_{\mathrm{H_2O}})} \cdot n_{\mathrm{air}} \cdot \Delta\sigma \ \mathrm{d}h. \tag{C3}$$

The integral runs over the air column between aircraft ($h_1$) and ground ($h_0$). In Eq. C3 the number density of greenhouse-gas
molecules has been expressed in terms of the number density of air molecules $n_{\mathrm{air}}$ and the dry-air molar mixing ratio of the
greenhouse-gas species, while also accounting for the dry-air molar mixing ratio of water

$$n_{\mathrm{air}} = (1+r_{\mathrm{H_2O}}) \cdot n_{\mathrm{dry\ air}}. \tag{C4}$$

710  Using the general gas equation ($n_{\mathrm{air}} = \frac{p}{k_{\mathrm{B}}\cdot T}$) $n_{\mathrm{air}}$ can be expressed in terms of pressure $p$ and temperature $T$. Furthermore,
the integral over altitude can be transformed into a pressure integral ($\mathrm{d}h = \frac{\mathrm{d}h}{\mathrm{d}p}\cdot\mathrm{d}p = -\frac{k_{\mathrm{B}}\cdot N_{\mathrm{A}}\cdot T}{p\cdot M_{\mathrm{air}}\cdot g}\ \mathrm{d}p$).

$$\Delta\tau = \int_{p_1}^{p_0} \frac{r_{\mathrm{CH_4}}\cdot N_{\mathrm{A}}}{(1+r_{\mathrm{H_2O}})\cdot M_{\mathrm{air}}\cdot g} \cdot \Delta\sigma \ \mathrm{d}p. \tag{C5}$$



Here, $k_{\mathrm{B}}$ is the Boltzmann constant, $N_{\mathrm{A}}$ the Avogadro constant, $g$ the gravitational acceleration and $M_{\mathrm{air}}$ the average molar mass of air. Next, molar mass is expressed as molecular mass ($m_{\mathrm{air}} = \frac{M_{\mathrm{air}}}{N_{\mathrm{A}}}$) and dry air is discriminated from water vapour.

$$\Delta\tau = \int_{p_1}^{p_0} \frac{r_{\mathrm{CH_4}}}{(1 + r_{\mathrm{H_2O}})} \cdot \frac{\Delta\sigma}{\underbrace{m_{\mathrm{air}}}_{= \frac{m_{\mathrm{dry\ air}} + r_{\mathrm{H_2O}}\ m_{\mathrm{H_2O}}}{1 + r_{\mathrm{H_2O}}}} \cdot g} \ \mathrm{d}p, \tag{C6}$$

$$= \int_{p_1}^{p_0} \frac{r_{\mathrm{CH_4}}}{(m_{\mathrm{dry\ air}} + r_{\mathrm{H_2O}} \cdot m_{\mathrm{H_2O}})} \cdot \frac{\Delta\sigma}{g} \ \mathrm{d}p. \tag{C7}$$

Finally, $r_{\mathrm{CH_4}}$ is pulled out of the integral by replacing it with a (weighted) column average $X\mathrm{CH_4}$, which is thus defined.

$$\Delta\tau = X\mathrm{CH_4} \int_{p_1}^{p_0} \frac{\sigma_{\mathrm{on}}(p,T) - \sigma_{\mathrm{off}}(p,T)}{g\,(m_{\mathrm{dry\ air}} + r_{\mathrm{H_2O}} \cdot m_{\mathrm{H_2O}})} \ \mathrm{d}p, \tag{C8}$$

$$= X\mathrm{CH_4} \int_{p_1}^{p_0} W(p,T) \ \mathrm{d}p. \tag{C9}$$

The quotient remaining in the integral is the so-called weighting function

$$W(p,T) = \frac{\sigma_{\mathrm{on}}(p,T) - \sigma_{\mathrm{off}}(p,T)}{g \cdot (m_{\mathrm{dry\ air}} + m_{\mathrm{H_2O}} \cdot r_{\mathrm{H_2O}})}, \tag{C10}$$

which carries the terms that are pressure/altitude dependent but known to some degree. In the standard data analysis routines of CHARM-F, the absorption cross-sections are calculated based on the spectroscopic data sets GEISA20 and Vasilchenko (Delahaye et al., 2021; Vasilchenko et al., 2023), and the state of the atmosphere (vertical structure at measurement location) is extracted from the ECMWF Integrated Forecasting System (IFS).

The partial-column weighted-average molar mixing ratio is therefore described by

$$X\mathrm{CH_4} = \frac{\Delta\tau}{\int_{p_1}^{p_0} W(p,T) \ \mathrm{d}p}. \tag{C11}$$

## C2    Column Anomalies from CHARM-F Measurements

This section describes the details of how we convert $X\mathrm{CH_4}$, as measured by CHARM-F, into a column anomaly of the dry-air $\mathrm{CH_4}$ molar mixing ratio for the total column. The reference for the calculation of anomaly is the methane concentration from the Copernicus Atmosphere Monitoring Service (CAMS) global inversion model (Segers and Houweling, 2023). Based on the dry-air molar mixing ratio from CAMS ($r_{\mathrm{CH_4,CAMS}}$) we calculate a column-averaged molar mixing ratio between surface (sfc) and flight altitude (flh), using number density of air molecules from CAMS as weight $n_{\mathrm{air}}$:

$$X\mathrm{CH_4_{CAMS}} = \frac{\int_{\mathrm{sfc}}^{\mathrm{flh}} r_{\mathrm{CH_4,CAMS}} \cdot n_{\mathrm{air}} \ \mathrm{d}z}{\int_{\mathrm{sfc}}^{\mathrm{flh}} n_{\mathrm{air}} \ \mathrm{d}z}. \tag{C12}$$

On this basis, the partial column anomaly $A_{\mathrm{pc}}$ is calculated.





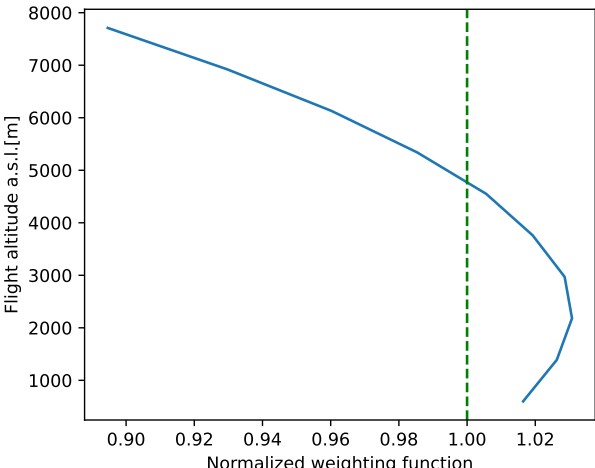

**Figure C1.** Typical altitude dependency of the CHARM-F weighting function for methane. The weighting function has been normalised to the particle-number-averaged value of the column between the ground and the flight altitude of about 8 km. The dashed green line indicates the unity weighting used for the CAMS reference column concentration.

Due to the temperature and pressure dependence of the spectroscopic properties of methane, $X\mathrm{CH}_{4\,\mathrm{CHARM-F}}$ and $X\mathrm{CH}_{4\,\mathrm{CAMS}}$ are weighted somewhat differently along the column (see Fig. C1). Nevertheless, in the hypothetical case, where methane is homogeneously distributed along the surveyed column, they give the exact same result by definition of the weighting function. To compensate for this small bias (few percent), which only affects the anomalous part of the column concentration, based on
740 the CHARM-F weighting function $W(p,T)$, we calculate a correction factor.

$$C_W = \frac{\frac{\int_{\mathrm{sfc}}^{\mathrm{flh}} W(p,T) n_{\mathrm{air}}\ \mathrm{d}z}{\int_{\mathrm{sfc}}^{\mathrm{flh}} n_{\mathrm{air}}\ \mathrm{d}z}}{\frac{\int_{\mathrm{sfc}}^{\mathrm{blh}} W(p,T) n_{\mathrm{air}}\ \mathrm{d}z}{\int_{\mathrm{sfc}}^{\mathrm{blh}} n_{\mathrm{air}}\ \mathrm{d}z}}\ . \tag{C13}$$

Here, we assume that methane plumes from a nearby source at the surface lead to a mole-fraction enhancement only below the top of the atmospheric BL. No enhancement is expected within the free troposphere, up to the flight altitude (flh). $C_W$ quantifies the ratio between how a homogeneously distributed methane enhancement would be perceived by CHARM-F versus
745 an enhancement that is also homogeneous, but restricted to the BL.

The partial column anomaly for the column below the aircraft can therefore be calculated as follows:

$$A_{\mathrm{pc}} = C_W \cdot \frac{X\mathrm{CH}_4 - X\mathrm{CH}_{4\,\mathrm{CAMS}}}{X\mathrm{CH}_{4\,\mathrm{CAMS}}}\ . \tag{C14}$$

Finally, also the column from the aircraft to the top of the atmosphere (toa) has to be considered. That region, where the anomaly is zero by definition, is combined with the column below by averaging, using particle number density of air molecules



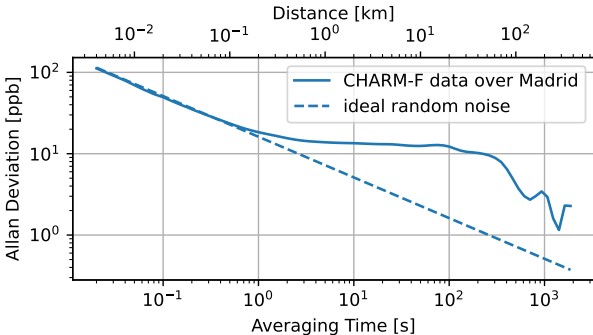

**Figure C2.** Allan deviation plot of the CHARM-F measurements over the Madrid landfills. In the case of pure ideal noise, the scatter reduces from single pulse-pair measurements with one over the square root of the number of measurements that are averaged. Deviations indicate either drifts, or like in this case, mostly real methane gradients.

as weight.

$$
A_{\text{tc}} = \frac{A_{\text{pc}} \cdot \int_{\text{sfc}}^{\text{flh}} n_{\text{air}} \ \mathrm{d}z}{\int_{\text{sfc}}^{\text{toa}} n_{\text{air}} \ \mathrm{d}z} . \tag{C15}
$$

Thus, $A_{\text{tc}}$ is the closest approximation of the $CH_4$ anomaly in terms of mole fraction for the total column along flight tracks, comparing CHARM-F measurements with CAMS reanalysis data as reference.

### C3 Uncertainties of CHARM-F Measurements

The measurement uncertainties of $X CH_4$ retrieved with CHARM-F can be categorised into statistical and systematic uncertainties. Statistical uncertainties are introduced through the measurement of DAOD, or more precisely, the measurement of the four signal intensities that contribute to DAOD. These are associated with a certain degree of noise, largely independent between individual laser pulses. The main noise sources are photon statistics, detector noise, and speckle noise (Ehret et al., 2008). The exact magnitude can be estimated from the system parameters or directly determined from measurement statistics. The influence of these uncertainties can be reduced by averaging multiple pulse-pairs. Such statistical uncertainties can be characterised with an Allan deviation (Allan, 1966) plot, as shown in Fig. C2. For an individual double-pulse measurement the standard deviation is about $100\,\text{ppb}$ or $5\%$. Averaging multiple measurements reduces the uncertainty with an inverse square-root law, as expected, until at about $1\,\text{s}$ of averaging. At this point the Allan deviation diverges from this noise-only expectation, which can be explained by actual gradients in form of the observed methane plumes. Residual uncertainties from systematic drifts or offsets have to be addressed using a different strategy.

Systematic uncertainties are related to inaccurate knowledge or deviation of certain system or meteorological parameters from the design/assumed values. These deviations typically change over time at very slow rates of the order of minutes to hours and are therefore highly correlated between individual pulse-pairs. Systematic uncertainties arise from various sources.



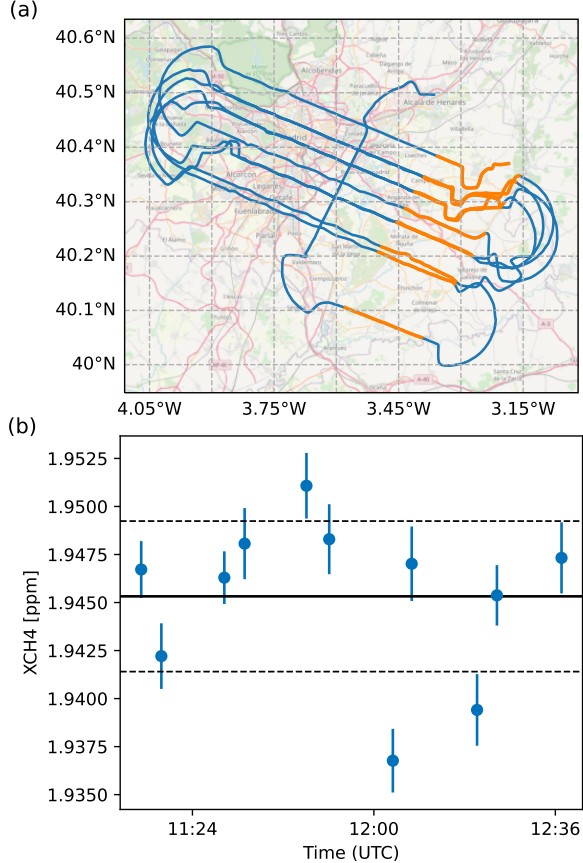

**Figure C3.** Uncertainties of the CHARM-F column concentration background, averaged over 100 second background segments. The flight-track segments that were used for the background-uncertainty study are shown in orange (**a**). The average values and statistical errors $X\mathrm{CH_4}$ for these segments are shown in panel (**b**), together with the overall mean value and standard standard-deviation band, drawn as solid and dashed lines respectively. Base map data © OpenStreetMap contributors, ODbL 1.0. Tiles accessed using Cartopy.

These include small misalignments in the optical setup, which could cause unequal clipping of light between the online and
offline pulses in the receiver. Additional factors contributing to uncertainties are errors in the tuning of the emitted wavelengths, spectroscopic inaccuracies and discrepancies between the numerical weather simulation model and the actual atmospheric conditions during measurement, which impact the calculation of the weighting function (Ehret et al., 2008).

In order to estimate relevant systematic uncertainties for our measurements, we define a background region, south-east of the landfills and also the city of Madrid. The region has been chosen such, that the aircraft typically stays within that region
for about $100\,\mathrm{s}$ or 5000 individual measurements (orange track segments in Fig. C3). This length/duration is comparable to the typical plume or background regions used for the flux measurements. For each overpass of the background region the average partial-column $X\mathrm{CH_4}$ is computed, resulting in a residual scatter of $4\,\mathrm{ppb}$ or $0.2\%$. From extrapolating the random-noise



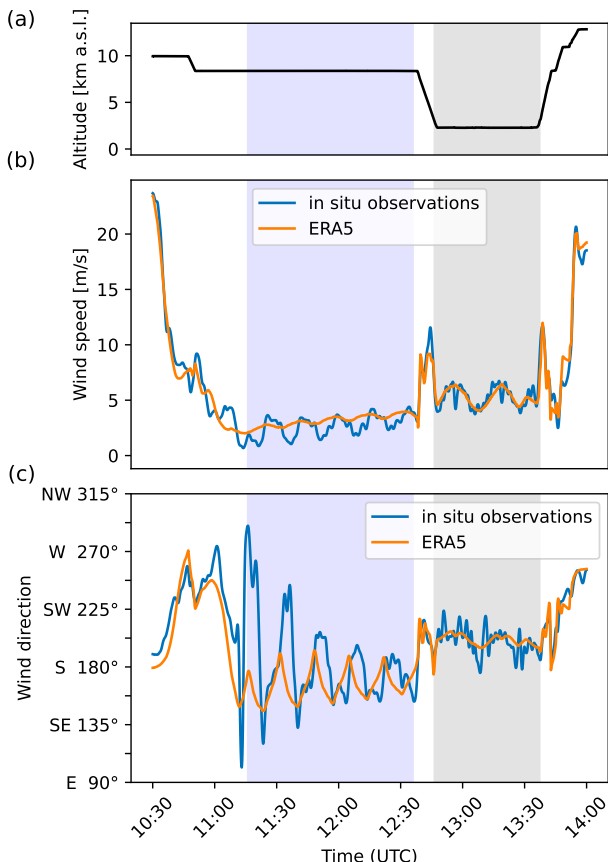

**Figure D1.** Flight altitude of the HALO aircraft **(a)**, wind speed **(b)** and direction **(c)** as modelled in ECMWF ERA5 vs. actual in situ measurements of BAHAMAS during the HALO flight over Madrid. The section of the flight that took place within the BL is highlighted in grey and the section for the remote sensing observations in light blue.

model in the Allan deviation plot, $2\,\mathrm{ppb}$ or $0.1\%$ are expected. We conclude that on the time scales that are relevant for our measurements, systematic effects lead to a doubling of the errors that would be expected from random noise only.

**Appendix D: Validation of ECMWF ERA5 Data by on-board Aircraft Measurements**

**D1    Validation of ERA5 Wind Data**

In order to confirm that the wind parameters, that are relevant for our flux or emission rate estimates, are correctly modelled in ECMWF ERA5, we compare them to on board wind in situ measurements on HALO from BAHAMAS (see Sect. 2.1.5). ERA5 data have been interpolated and evaluated along the flight track in space and time. The on-board measurements have

been smoothed with a $30\,\mathrm{s}$ Gaussian kernel to reduce fluctuations from turbulence, which is on a scale far below the resolution





of ERA5. The ERA5 model matches the BAHAMAS measurements very well over longer time frames, especially when neglecting the small scale turbulence still visible in Fig. D1. This is particularly valid for the part of the flight within the BL (gray region), where the plume is located in. The averaged difference between model and measurement for the wind speed and wind direction is $0.05\,\mathrm{m\,s^{-1}}$ and $0.8°$ in the BL (averages are actually quite independent from the applied smoothing kernel).
The only caveat is that the time period for which the good match can be demonstrated is up to around $2\,\mathrm{h}$ after the start of remote sensing section of the flight. Therefore, a similarly good match between model prediction and actual wind situation within the BL has to be assumed also for earlier times without explicit proof. The significantly larger mismatch between ERA5 and the measured winds in some segments of the flight at earlier times (not within the PBL) can most likely be explained with strong wind shear in a thin layer, exactly at about the flight altitude during the remote sensing part of the flight, in combination
with a relatively coarse vertical resolution of ECMWF there.

Interestingly, during the descent into the BL (at around 12:45 UTC) and the subsequent ascent around two hours later the wind speed peaks at values of up to $10\,\mathrm{m\,s^{-1}}$ without significant change in wind direction. This indicates a strong vertical gradient in wind speed in the BL in both model and measurement from top of the BL to at least $1.6\,\mathrm{km\,a.g.l.}$. This is actually confirmed by vertical profiles of the wind speed from ERA5 data (Fig. D2, a). They show a strong vertical gradient, which is
around $2\,\mathrm{m\,s^{-1}}$ at the ground and increases to $10\,\mathrm{m\,s^{-1}}$ at the top of BL for 12 UTC. Moreover, on that day the wind speed increases from 10 to 14 UTC in the BL. The wind direction (b) is relatively stable after 11 UTC with values between $200°$ and $220°$. However, before the stabilisation to south-south-west at 11 UTC (right before the measurement flight started), the wind had a strong easterly component.

## D2 Validation of ERA5 BLH Data

For the BLH, ERA5 shows an increase from $\sim 1.3\,\mathrm{km\,a.s.l.}$ ($0.7\,\mathrm{km\,a.g.l.}$) to $\sim 4.2\,\mathrm{km\,a.s.l.}$ ($3.6\,\mathrm{km\,a.g.l.}$) nicely illustrated in the potential temperature profiles for that location and time period (Fig. D2, c). To validate ERA5's BLH, we compare it to measured potential temperature profiles during the descent and ascent at $12:45\,\mathrm{UTC}$ ($40.345°\mathrm{N}$, $3.010°\mathrm{W}$) and $13:40\,\mathrm{UTC}$ ($40.322°\mathrm{N}$, $3.745°\mathrm{W}$), respectively, at the respective locations. The descent took place over a more hilly environment approx. $50\,\mathrm{km}$ east of the landfills and the ascent right over the city of Madrid $12\,\mathrm{km}$ west of the landfills.

Both profiles are shown in Fig. D3 (a). We estimate the BLH for both profiles to $\sim 3.2\,\mathrm{km\,a.s.l.}$ (considering topography would yield $2.4\,\mathrm{km\,a.g.l.}$ and $2.6\,\mathrm{km\,a.g.l.}$ for descend and ascent, respectively) due to the strong increase in potential temperatures at these altitudes. Comparing the BLHs from ERA5 at these positions and times yield significantly and consistently higher BLHs of $700\,\mathrm{m}$ relative to sea level (b, c). In order to correct for this discrepancy but also to transfer it to earlier time of the measurement, when the remote sensing observations were collected, we use the potential temperature profile from ERA5
and estimate a new BLH. We do that by using the temperature at the surface and then searching for the altitude level where this value is approached for the first time. The assumption behind this is that an air parcel, having a certain potential temperature, rises if the potential temperature is lower in the surrounding air masses and reaches an equilibrium (height) when the potential temperature of the surrounding is similar. This process is indicated in the panels (b) and (c) in Fig. D3 by the vertical red lines





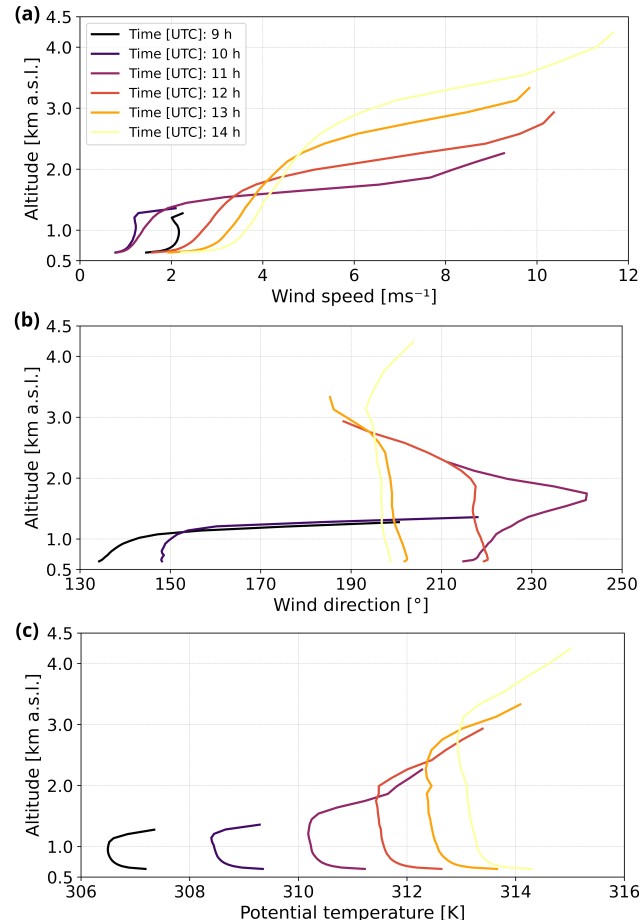

**Figure D2.** Vertical profiles for wind speed **(a)**, wind direction **(b)**, and potential temperature **(c)** between 9 and 14 UTC based on the ECMWF ERA5 model data. The height of the profiles is restricted to the BLH as given in ERA5. Time of the remote sensing overflight was between 11:00 and 12:40 UTC. In situ data collection within the BL was between 12:50 and 13:30 UTC. The profiles are representative for 40.292°N and 3.614°W, which is located between the two landfill complexes.

for the respective time steps of the model. If the leg, or in this case one profile, is collected in between two time steps, the newly computed BLH is linearly interpolated in time.

For the descending and ascending profiles, this method yields $\sim 3.33\,\mathrm{km\,a.s.l.}$ ($2.53\,\mathrm{km\,a.g.l.}$) and $\sim 3.65\,\mathrm{km\,a.s.l.}$ ($3.05\,\mathrm{km\,a.g.l.}$), respectively, which is up to $15\,\%$ lower than given directly by ERA5 with regards to $\mathrm{km\,a.s.l.}$.. On average over all leg positions and times, the BLHs decrease by around $17\,\%$, which also leads to a decrease in the computed wind speed of $24\,\%$. As conservative uncertainty estimate, we assume an error of $20\,\%$ in our BLH estimate with respect to its depth relative to the ground. A deeper BL would have a larger error. Applying this to the profiles would lead to absolute errors in the BLH for descent and





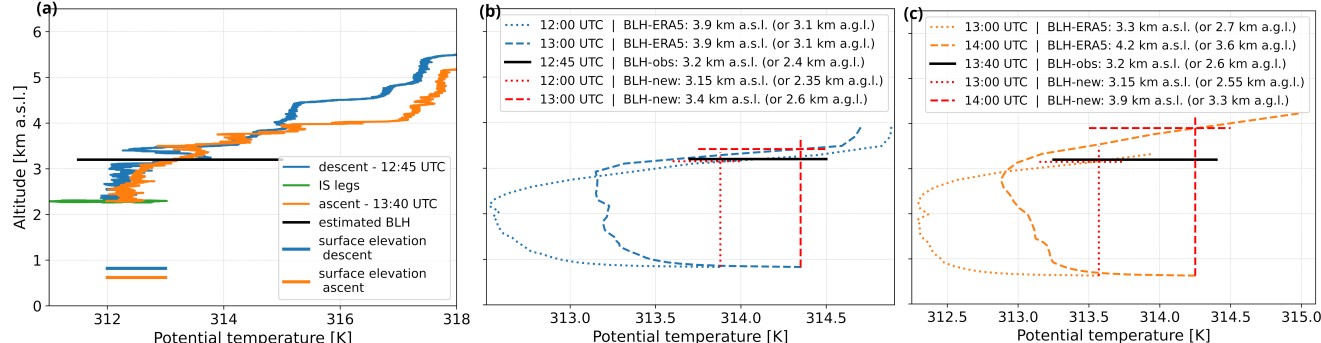

**Figure D3.** Comparison between HALO in situ measurements and ERA5 model data. The vertical profiles of potential temperature for the descent at 12:45 UTC (40.345°N, 3.010°W) and the ascent at 13:40 UTC (40.322°N, 3.745°W) as measured by BAHAMAS aboard HALO are shown in **(a)**. The dotted black horizontal line marks the BLH, estimated from these profiles to be 3.2 km a.s.l., in all sub-figures. Blue marks the profile during descent; the relevant ERA5 model profiles at that location and times (12, dotted, and 13, dashed, UTC) are then given in **(b)**. Orange marks the profile during ascent; the relevant EAR5 model profiles at that location and times (13, dotted, and 14, dashed, UTC) are then given in **(c)**. Red vertical and horizontal lines mark the newly estimated BLH for each time step in the model based on the method described in the main text. Different line types show different temporal affiliations. BHL values are given in the legends.

ascent of 0.5 km and 0.6 km, respectively. This uncertainty in BLH translates into a wind speed uncertainty of $0.8\,\mathrm{m\,s^{-1}}$ (or 20 %) on average for our flight legs. The wind direction is hardly influenced by a change in BLH.

## Appendix E:  Column and Flux Comparison MAMAP2DL and CHARM-F

Figure E1 summaries the comparison of cross-sections between MAMAP2DL and CHARM-F.

## Appendix F:  Errors of the Flux Estimates

### F1  General Error Handling

Assuming that the $CH_4$ emission plumes are well-mixed within the BL, major error sources for the cross-sectional flux computation are the wind speed and direction, the random and systematic errors of the retrieved column anomalies from the remote sensing instruments, the boundaries or limits of the background normalisation used to compute the real enhancements, and the

uncertainty in the estimate of the BLH from ERA5. To compute these errors, we use Gaussian error propagation of Eq. 2 where possible. All considered sources of uncertainty are summarised in Table F1 and their range is given in Table 1. Error propagation is applied to the error in wind speed ($\delta F_u$) and the error in the BLH ($\delta F_{\mathrm{blh}}$), which changes the part of the wind profiles over which the average is calculated, and remaining systematic errors of the columns ($\delta F_{\mathrm{col-ac}}$) and of the conversion factor ($\delta F_{\mathrm{col-cf}}$). Moreover, the random error of the columns ($\delta F_{\mathrm{col-pr}}$) is propagated. However, the single-column precision is addi-





**Figure E1.** Same as Fig. 6 but for the remaining cross-sectional fluxes of the different legs from MAMAP2DL and CHARM-F observations. Cross-section in Fig. 6 would correspond to **(g)**. Order of cross-sections is from upwind **(a)** to downwind **(j)**. The spike in the CHARM-F column anomalies at ∼5 km in **(i)** is an albedo artefact due to retro-reflecting road markings parallel to the flight leg.

tionally divided by $\sqrt{n}$, with $n$ being the number of measurements or ground scenes inside of the plume of one cross-section, taking into account its random nature (see for example Appendix C3). The error of the wind direction on the computed flux is



considered by varying the direction ($\alpha$) according to the respective directional uncertainty. The error in the flux is then given by the variation of the computed fluxes. In a similar way, we take into account the error of the background normalisation ($\delta F_{bg}$) by computing a set of fluxes based on reasonable variations of the background limits and calculating the standard deviation of
their differences from the main flux estimate.

**Table F1.** Summary of relevant error sources and quantities for the flux computation.

| Parameter | Description |
|---|---|
| $\delta F_u$ | error in wind speed |
| $\delta F_\alpha$ | error in wind direction |
| $\delta F_{blh}$ | error in boundary layer height |
| $\delta F_{bg}$ | error in the selected background region(s) |
| $\delta F_{col\text{-}pr}$ | column single measurement precision |
| $\delta F_{col\text{-}ac}$ | remaining systematic column offset (Appendix B2, only MAMAP2DL) |
| $\delta F_{col\text{-}cf}$ | error in the conversion factor (Appendix B3, only MAMAP2DL) |
| $\delta F_{M2D,\,cs}$ | error of one cross-section (MAMAP2DL) |
| $\delta F_{M2D,\,leg}$ | error of one leg |
| $\delta F_{M2D,\,css}$ | errors from the single cross-sections |
| $\delta F_{M2D,\,atm,\,css}$ | error due to atmospheric turbulence within one leg |
| $\delta F_{M2D,\,ar\text{-}aver}$ | error of one area (or areal average) |
| $\delta F_{M2D,\,legs}$ | error from the single legs |
| $\delta F_{M2D,\,atm,\,legs}$ | error due to atmospheric turbulence across area |
| $\delta F_{CHARM\text{-}F,\,leg}$ | error of one leg |
| $\delta F_{CHARM\text{-}F,\,ar\text{-}aver}$ | error of one area (or areal average) |
| $\delta F_{CHARM\text{-}F,\,legs}$ | errors from the single legs |
| $\delta F_{CHARM\text{-}F,\,atm,\,legs}$ | error due to atmospheric turbulence across area |

## F2    Error Handling for MAMAP2DL

In the case of MAMAP2DL the total error of the flux of one cross-section ($\delta F_{M2D,\,cs}$) is calculated by root sum squaring the error contributions:





$$\delta F_{\text{M2D, cs}} = \sqrt{\begin{array}{c} \delta F_u^2 + \delta F_\alpha^2 + \delta F_{\text{blh}}^2 + \delta F_{\text{bg}}^2 \\ + \delta F_{\text{col-pr}}^2(n) + \delta F_{\text{col-ac}}^2 + \delta F_{\text{col-cf}}^2 \end{array}} . \tag{F1}$$

If the emission source is constant, the actual uncertainty of the flux of one cross-section at a certain position is also influenced by atmospheric variability or turbulence in the atmosphere (Krautwurst et al., 2021; Wolff et al., 2021; Krautwurst et al., 2017; Matheou and Bowman, 2016). This is also visible in the imaging data and at this point not covered by the error $\delta F_{\text{M2D, cs}}$ (Eq. F1). However, this error component is reduced by taking spatially and/or temporally independent cross-sections. We estimate this factor as 1-sigma standard deviation (SD) from the cross-sections of one entire MAMAP2DL leg itself:

$$\delta F_{\text{M2D, atm, css}} = \frac{\text{SD}(F_{\text{M2D, cs},j})}{\sqrt{m_{\text{eff}}}} \tag{F2}$$

where $m_{\text{eff}}$ is the number of actual independent cross-sections across a MAMAP2DL flight leg. Additionally, the wind ($\delta F_u^2$ and $\delta F_\alpha^2$), BLH ($\delta F_{\text{blh}}^2$), background ($\delta F_{\text{bg}}^2$) and conversion factor ($\delta F_{\text{col-cf}}^2$) errors introduced in Eq. F1 must be considered as systematic errors[7]. For the average flux of one leg, they are computed by averaging over the respective errors for all cross-sections of one leg. Eventually, the error of one leg $\delta F_{\text{M2D, leg}}$ summarises therefore to:

$$\delta F_{\text{M2D, leg}} = \sqrt{\begin{array}{c} \delta F_{\text{M2D, css}}^2 + \delta F_{\text{M2D, atm, css}}^2 \\ + \delta F_u^2 + \delta F_\alpha^2 + \delta F_{\text{blh}}^2 + \delta F_{\text{bg}}^2 + \delta F_{\text{col-cf}}^2 \end{array}} \tag{F3}$$

where

$$\delta F_{\text{M2D, css}} = \frac{\sqrt{\sum_{j=0}^{l} \delta F_{\text{M2D, cs},j}^2}}{m} \tag{F4}$$

where

$$\delta F_{\text{M2D, cs},j} = \sqrt{\delta F_{\text{col-pr},j}^2(n) + \delta F_{\text{col-ac},j}^2} , \tag{F5}$$

where $\delta F_{\text{M2D, css}}$ is the contribution of the single cross-sections' errors. However, only random ($\delta F_{\text{col-pr}}^2(n)$) and remaining systematic ($\delta F_{\text{col-ac}}^2$) errors of the columns from Eq. F1 are included, as others are systematic for all cross-sections of one leg. $m$ is the actual number of cross-sections of one leg.

The argumentation above also applies if we compute a mean flux over a specific area, e.g., downwind of the Las Dehesas area or in between the two waste treatment areas, from the averaged MAMAP2DL legs:

$$\delta F_{\text{M2D, ar-aver}} = \sqrt{\begin{array}{c} \delta F_{\text{M2D, legs}}^2 + \delta F_{\text{M2D, atm, legs}}^2 \\ + \delta F_u^2 + \delta F_\alpha^2 + \delta F_{\text{blh}}^2 + \delta F_{\text{col-cf}}^2 \end{array}} , \tag{F6}$$

---

[7]I.e. not being reduced by averaging over several cross-sections.



where

$$\delta F_{\text{M2D, legs}} = \frac{\sqrt{\sum_{j=0}^{p} \delta F_{\text{M2D, leg},j}^2}}{p} \,, \tag{F7}$$

where

$$\delta F_{\text{M2D, leg},j} = \sqrt{\delta F_{\text{css},j}^2 + \delta F_{\text{M2D, atm, css},j}^2 + \delta F_{\text{bg},j}^2} \,, \tag{F8}$$

and

$$\delta F_{\text{M2D, atm, legs}} = \frac{\text{SD}(F_{\text{M2D, leg},j})}{\sqrt{p}} \,, \tag{F9}$$

where $\delta F_{\text{M2D, legs}}$ are the error contributions of the errors of the single legs of MAMAP2DL, however, excluding systematic uncertainties that are valid for the entire areas such as errors in wind, BLH, and the conversion factor. For example, the background error is not systematic for different legs anymore. $p$ is the number of (independent) legs.

## F3   Error Handling for CHARM-F

In contrast to MAMAP2DL, the CHARM-F instrument measures in one viewing direction so that the differentiation between cross-section and leg is not necessary. Furthermore, errors for $\delta F_{\text{col-ac}}$ and $\delta F_{\text{col-cf}}$ are negligible and Eq. F1 simplifies thus to

$$\delta F_{\text{CHARM-F, leg}} = \sqrt{\delta F_u^2 + \delta F_\alpha^2 + \delta F_{\text{blh}}^2 + \delta F_{\text{bg}}^2 + \delta F_{\text{col-pr}}^2(n)}. \tag{F10}$$

The error of the average over certain areas follows then modified versions of Eqs. F6 to F9.

$$\delta F_{\text{CHARM-F, ar-aver}} = \sqrt{\begin{array}{c} \delta F_{\text{CHARM-F, legs}}^2 + \delta F_{\text{CHARM-F, atm, legs}}^2 \\ + \delta F_u^2 + \delta F_\alpha^2 + \delta F_{\text{blh}}^2 \end{array}} \tag{F11}$$

where

$$\delta F_{\text{CHARM-F, legs}} = \frac{\sqrt{\sum_{j=0}^{p} \delta F_{\text{CHARM-F, leg},j}^2}}{p} \tag{F12}$$

where

$$\delta F_{\text{CHARM-F, leg},j} = \sqrt{\delta F_{\text{bg},j}^2 + \delta F_{\text{col-pr},j}^2(n)} \,, \tag{F13}$$

and

$$\delta F_{\text{CHARM-F, atm, legs}} = \frac{\text{SD}(F_{\text{CHARM-F, leg},j})}{\sqrt{p}} \,. \tag{F14}$$



*Author contributions.* Deriving the results presented here involved the contributions of a large team of scientists. This includes the development of hardware and software, the planning of the measurement campaign, data analysis and last but not least the preparation of the manuscript including a thorough internal review process.

SK performed the analysis of the MAMAP2DL data (from measured spectra to final fluxes) and adapted the algorithms accordingly. CF performed the analysis and conversion to column anomalies of the CHARM-F data and a preliminary flux estimate from the CHARM-F data. SK then performed the final flux inversion of CHARM-F data. MG processed the JIG and JAS data. CM performed the error analysis of the wind data from BAHAMAS.

SK developed the flight plan together with inputs from SW and MG. MQ and AF operated the CHARM-F system during the flight. OH
operated the MAMAP2DL system during the flight. AF was principal investigator of the CoMet 2.0 mission.

KG, with support from SK and JB, designed and built the MAMAP2DL instrument. JB designed and wrote the operating software for MAMAP2DL. AF, MQ, and MW developed the lidar system CHARM-F. MW designed and wrote the retrieval software for CHARM-F data.

SK, CF, and SW wrote the first draft version of the manuscript. All the authors critically assessed the results and corrected and improved the manuscript in an iterative process.

*Competing interests.* CG has a competing interest as he is ACP editor (and CoMet special issue editor). CK has a competing interest as he is an associate editor at AMT. The remaining authors declare that they have no conflict of interest.

*Acknowledgements.* HALO flights during the CoMet 2.0 Arctic mission have been supported by the State of Bremen, the Max Planck Society (MPG), and by the German Research Foundation (Deutsche Forschungsgemeinschaft, DFG) within the DFG Priority Program (SPP 1294) Atmospheric and Earth System Research with the Research Aircraft HALO (High Altitude and LOng Range Research Aircraft) under grant
BO 1731/1-1. MAMAP2DL was in parts funded by BMBF within the project AIRSPACE (01LK1701B), by the University of Bremen and by the State of Bremen (APF). The DLR part of CoMet 2.0 Arctic was funded by DLR headquarters, e.g., via the internal MABAK project.

The publication contains modified Copernicus Atmosphere Monitoring Service (CAMS) information (2023) downloaded from CAMS Atmosphere Data Store (ADS). Neither the European Commission nor ECMWF is responsible for any use that may be made of the Copernicus information or data it contains.

A special thanks goes to A. Schäfler from the DLR for the discussion and interpretation about the boundary layer development in that area on the day of our measurement flight. We also want to thank the weather forecast team (C. Kiemle, A. Schäfler, S. Wolff, S. Arnold, S. Groß, F. Reum, H. Zeis, M. Gutleben) whose input was essential for flight planning. A big thank you also goes to the entire HALO - Flight Experiments and - Operations teams who took care of the operation of the aircraft and analysed and provided all the data provided by HALO, and especially to A. Minikin, T. Sprünken, A. Hausold, and F. Betsche, who implemented our flight ideas and obtained the necessary permits
from the respective authorities. We would also like to thank our pilots S. Grillenbeck and M. Puskeiler realising our flight ideas during the Madrid flight. A big thank you also goes to R. Maser, C. Hallbauer, and D. Schell from enviscope, who took care of the certification of the MAMAP2DL instrument aboard HALO for the CoMet 2.0 mission. A final thank you goes to M. D. A. Hernández from the IUP Bremen who tried to contact the Spanish authorities and landfill operators via a Spanish acquaintance, unfortunately without success.



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
