# Peer review of "Identification and Quantification of CH4 Emissions from Madrid Landfills using Airborne Imaging Spectrometry and Greenhouse Gas Lidar"

_EGUsphere, 2024_

## Author Comment (AC1)

Referee/Reviewer #1

We would like to thank the anonymous referee #1 for the careful reading and valuable comments, which we address point-by-point.

In the following, the reviewer comments are in *italics*, the answers in plain text and the proposed changes / new text in blue. References to Sections or Figures in our answers refer to the discussion version of the manuscript.

*The manuscript by Krautwurst et al. deals with an airborne campaign to characterise methane emissions from two landfills close to the Madrid city. The MAMAP2DL and CHARM-F instruments (pushbroom imaging spectrometer and active lidar, resp.) were flown over the landfills to derive maps of methane column concentrations, which were used to estimate emission rates.*

*The manuscript is very well written and presented, the methods are sound, and the results are very solid. On the downside, perhaps the level of novelty is not great, given that the high methane emissions from those landfills are well known, and the two instruments and the corresponding processing methods are already described in a number of peer-reviewed publications. For example, I miss some synergistic application of the two instruments.*

*Having said that, I recommend publication of this manuscript in ACP because of the growing interest in the development of methods for the monitoring of methane emissions and of the great technical quality of the study.*

Thank you for the encouraging words. We note that there is only one peer-reviewed publication about the Madrid waste treatment areas' emissions of methane covering the period 2018 to 2020 [Tu et al., 2022]. With the data presented in our paper using different sensors at a different time (summer 2022), the magnitude of the emissions is emphasized once again and indicates consistently high values. Furthermore, as stated in our comment to Line 43 of Referee#2, the CH4 emissions of the Pinto landfill should have dropped by a factor of almost 40 from 2021 to 2022 according to the reporting.

We agree with the reviewer that both measurement principles and processing methods are well established in the community and have extensively been used in the past to quantify greenhouse gas emissions. However, to our knowledge, this is the first study which combines active and passive remote sensing observations for methane emission quantification. We emphasize that the analysis of the measurements was made independently and that 'only' the final products such as CH4 concentration anomalies or the emissions of methane are used in further comparisons and analysis. We contend that having two independently characterized simultaneous methane emission data products is the optimal synergetic approach.

Nevertheless, for future work, it may be worthwhile to combine information collected by active and passive instruments already on a CH4 column retrieval level. However, the focus of this study was to quantify the emissions of the two landfills and to assess whether they are as high as previously indicated by satellite (and ground-based) observations.

###################################################################

*Specific comments:*
*Below I am listing a number of minor points to be addressed at the authors' discretion.*
*P1, L18: the definition of the ERF is probably not needed*

Agreed. The sentence has been amended accordingly.

It has an effective radiative forcing of ~ 0.54 W m−1 or one quarter of that of CO2 (Forster et al., 2021).

####################################################################

*P2, L28: "is produced"?*

Agreed and amended accordingly.

####################################################################

*P3, L56-57: thermal imagers can also be "passive remote sensing imaging instruments"*

Agreed and amended accordingly.

Recently, passive remote sensing imaging instruments exploiting solar electromagnetic radiation in the near and short wave infrared have been deployed.

####################################################################

*P4, L115: "The non-operating ..." verb missing?*

Agreed and amended accordingly.

Additionally, it contains landfill sites such as the non-operating Valdemingómez landfill...

####################################################################

*P5, Fig 1 caption: "Spain" → "Iberian Peninsula"*

Agreed and amended accordingly.

####################################################################

*P7, Fig 2 caption (and elsewhere): I had never heard the term "ground scene size". I think is referring to "ground sampling distance" or "pixel size"?*

We would refrain from using the term "pixel size", as this can sometimes be confusing when also talking about the sensor (chip) or FPA (Focal Plane Array) itself, which in our case has 384 x 288 pixels with a specific size in the μm range. The term "ground scene size" has a long legacy (e.g. in Afe et al., 2004, or Gerilowski et al., 2011) and is also used by other groups and in related fields (e.g., Lin et al, 2018,

or Ren et al., 2022). In our case, it expresses the size of the two dimensions of an observation through the atmosphere at the ground, which are defined by the used optical fibers of the optical fiber bundle within the instrument, the exposure time, and an additional co-adding of observations along the flight direction.

###############################################################################

*P13, Eq. 4: please consider to use shorter variable names / subscripts for the emission rates F*

Thank you for bringing this up. In fact, there were also long internal discussions on the subject among co-authors, going back and forth between even longer subscripts or just one letter subscripts. We have opted for an intermediate solution to allow the reader to better understand the meaning of the different emission rates F without having to search for them in the text or in the separate table.

###############################################################################

*P15, L374: the "different opening angles of the two instruments" are provided as the main reason for the difference in the flux estimates by the two instruments. Can we expect that factor to be more important than potential differences in the column concentration retrievals (there are some in Fig. 6), and the different atmospheric paths sampled by the two instruments?*

The different viewing geometries or opening angles of the instruments result in observations of different but overlapping air masses below the aircraft. This can, however, only partly explain the observed difference in the column, for example if an air mass has highly varying methane column amounts over short distances, as indicated in the manuscript. Differences also arise from different atmospheric light paths. For the passive remote sensing MAMAP2DL measurement, the sun first passes through the entire atmosphere at a given angle before being reflected from the ground towards the aircraft. In comparison, light from the CHARM-F lidar is emitted from the instrument itself on board the aircraft, being reflected and scattered at the ground before passing through the atmosphere and being collected again at the aircraft. Furthermore, differences could also originate from the different column retrievals, e.g., how they deal with variable surface reflectivity etc. However, the extent to which each factor contributes would require a dedicated study.

We have slightly expanded the paragraph in the main text accordingly.

These differences may be due to (a) different but overlapping opening angles of the two instruments and the resultant spatial resolution of the ground scene widths of 110 m and 10 m, respectively, or (b) different paths through the atmosphere of the electromagnetic radiation used to measure methane absorption, or (c) differences in the algorithms used to retrieve the columns, e.g. how they deal with variable surface reflectivity etc.

###############################################################################

*P23, L533: I don't think that is true. For example, Frankenberg et al. mapped methane missions with the AVIRIS-NG (optical) and HyTes (thermal) imaging spectromenters https://doi.org/10.1073/pnas.1605617113*

Thanks for pointing that out. While writing this paragraph, we were addressing only the two different remote sensing techniques, "active" and "passive", and not two different passive remote sensing instruments. We have amended the sentence accordingly.

This was the first time that emissions were observed and quantified simultaneously by two different and independent active and passive remote sensing techniques.

###############################################################################

*P24, L562: "influence"*

Agreed and amended accordingly.

###############################################################################

References:

Afe, O. T., A. Richter, B. Sierk, F. Wittrock, and J. P. Burrows: BrO emission from volcanoes: A survey using GOME and SCIAMACHY measurements, *Geophys. Res. Lett.*, 31, L24113, doi:10.1029/2004GL020994, 2004.

Lin, C., Tang, S., Zhang, L., and Guo, P.: Focusing High-Resolution Airborne SAR with Topography Variations Using an Extended BPA Based on a Time/Frequency Rotation Principle. Remote Sensing, 10(8), 1275, https://doi.org/10.3390/rs10081275, 2018.

Ren, Y., Tang, S., Dong, Q., Sun, G., Guo, P., Jiang, C., Han, J., and Zhang, L.: An Improved Spatially Variant MOCO Approach Based on an MDA for High-Resolution UAV SAR Imaging with Large Measurement Errors. Remote Sensing, 14(11), 2670, https://doi.org/10.3390/rs14112670, 2022.

[revised manuscript text omitted]

 Active remote sensing instruments are independent of sunlight because they use a laser as their own source of electromagnetic radiation. In contrast to airborne and satellite-borne passive instruments, they can measure during day and night, across all seasons and latitudes. They provide ranging capabilities resulting from the precise measurement of the propagation time of the emitted light and, due to their narrow field of view, measure between clouds. IPDA (integrated-path differential absorption) lidars potentially provide highly accurate measurements without varying biases: the exceptions are those introduced by small differences in the scattering and reflectivity of the ground scene and any inaccuracies in the knowledge of the absorption cross-sections. However, Wolff et al. (2021) showed that under turbulent conditions, the spatial distribution of enhanced concentrations within an exhaust plume may be highly heterogeneous. As a result, a single overflight may sample sections with stronger or weaker enhancements purely as a result of the local variability. In some cases, the true emission signal only emerges after averaging over a high number of overflights.

To account for this potential limitation, in the analysis, we combine active lidar with passive imaging spectrometry, both designed to capture atmospheric $CH_4$ column gradients . Thus, we obtain both high-precision transects and spatial context, which supports a more robust interpretation of the observed $CH_4$ column enhancements. Moreover, these remote sensing measurements are complemented by auxiliary in situ measurements of $CH_4$, $CO_2$ and 3D winds in support of the remote sensing data. It was the first time that this payload was flown aboard the same aircraft acquiring spatially and

temporally collocated active and passive remote sensing measurements side-by-side for the acquisition of atmospheric $CH_4$ column observations. The greenhouse gas lidar CHARM-F ($CO_2$ and $CH_4$ Atmospheric Remote Monitoring Flugzeug) is an airborne demonstrator for the future satellite mission MERLIN

90 (MEthane Remote Sensing LIdar missioN, Ehret et al., 2017). The passive imaging MAMAP2DL (Methane Airborne Mapper 2D - Light) remote sensing instrument demonstrates the applicability of the $CH_4$ proxy retrieval (see below) at scales probed by CO2M (Copernicus Anthropogenic Carbon Dioxide Monitoring, Sierk et al., 2021) and TANGO (Twin Anthropogenic Greenhouse Gas Observer, SRON, 2024). The observations were collected in 2022 as part of the CoMet 2.0 (Carbon Dioxide and Methane) Arctic mission in Canada (CoMet, 2022). Prior to the transfer to Canada, an initial research flight was carried

95 out to test all the instruments. This test flight was performed on the 4th August over Madrid to investigate the unexpected high landfill emission rates reported by Tu et al. (2022) and in a webstory from the European Space Agency (ESA) from October 2021 (ESA, 2021).

In Sect. 2, we provide a brief summary of the CoMet 2.0 mission and introduce the main instruments MAMAP2DL and CHARM-F used in this study (Sect. 2.1). This also includes a description of the algorithms used to infer $CH_4$ columns from

100 the measurements (Sect. 2.2), additional steps necessary to achieve comparability between the passive and active observations (Sect. 2.3), and the cross-sectional flux method, which is used to quantify the $CH_4$ emissions (Sect. 2.4). Section 3 describes the observed $CH_4$ plumes over Madrid from both remote sensing instruments. This data is used to pin-point the exact source locations within the landfill area (Sect. 3.1), followed by a rigorous comparison of the active and passive data (Sect. 3.2), the resulting emission fluxes (Sect. 3.3), and a comprehensive discussion of potential uncertainties (Sect. 3.4). We close the paper

105 by discussing our fluxes in a broader context (Sect. 4) and summarising our findings (Sect. 5).

**2 Methods and Data**

**2.1 Campaign and Instrumentation**

 Below we describe the

110 waste treatment facilities

**2.1.1**

115  that were the targets of the research flights, and the flight strategy used to derive their methane emission rates. We then describe the instruments used, which were installed aboard the German research aircraft HALO (High Altitude and Long Range Research Aircraft, operated

by the DLR, Deutsches Zentrum für Luft- und Raumfahrt, type: Gulfstream G550).  The retrieval algorithms used to derive the $CH_4$  columns are explained next. This is followed by a description of the observed $CH_4$ ~~concentrations and of the basic data acquisition system of HALO, BAHAMAS (Basic Halo Measurement and Sensor System) including SHARC (Sophisticated Hygrometer for Atmospheric ResearCh) were used. The HALO basic data acquisition suite collects various atmospheric parameters such as the 3D wind field, temperature, humidity, and aircraft attitude data that is used for interpretation of the remote sensing data. The main instrumentation and the flight strategy are described in more detail below.~~ columns. Finally, we derive the methane fluxes, i.e. the methane emission rates, using the plume cross-sections.

**2.1.1 Target Description and Flight Strategy**

The targets under consideration were the Mancomunidad del Sur landfill in the municipality Pinto (40.264°N, 3.633°W; hereafter: Pinto landfill) and the Valdemingómez technology park (VTP, 40.332°N, 3.586°W) in the south-east of Madrid, Spain. The latter is a waste treatment complex  that accepts around 1,222 kt (Madrid, 2022) of waste, of which around 140 kt were deposited at the Las Dehesas landfill site in 2022 (Spanish-PRTR, 2025a), and houses several waste treatment facilities including the largest biomethane plant in Spain (Calero et al., 2023), which is also one of the largest in Europe (UABIO, 2022). Additionally, it contains landfill sites  such as the non-operating Valdemingómez landfill (40.331°N, 3.580°W) equipped with a gas recovery system and an active landfill site (40.325°N, 3.591°W; hereafter: Las Dehesas landfill) next to the waste treatment plant Las Dehesas. The northern half of the Las Dehesas landfill, where certain areas (i.e. cells) are already full and therefore closed, are also equipped with a gas recovering system (Sánchez et al., 2019). The two landfills in the technology park spread over an area of $\sim 0.9$ and $0.6\,km^2$ for the inactive Valdemingómez and the active Las Dehesas sites, respectively. More details about the different facilities are in the Annual Report for 2022 for the VTP (Madrid, 2022). The Pinto landfill, further to the south, stretches over $\sim 1.5\,km^2$. It opened in 1987, is still operational with around  53 kt of waste being dumped in 2022 (Spanish-PRTR, 2025b), and the already closed parts of the landfills are equipped with gas recovering system (MdS, 2024). The topography around these landfill sites shows some variability, ranging from about 550 to 700 m a.g.l. with a small valley between the two sites and a steep rise just south of the VTP according, to Google Earth.

According to the Spanish PRTR (Spanish-PRTR, 2025c), the combined annual reported 2022 $CH_4$ emissions for the two facilities "VERTRESA-URBASER, S.A. UTE

[Figure]

**Figure 1.** Top-view of the flight path of the HALO aircraft during the test flight over Madrid. An overview map in Google Earth of  the Iberian Peninsula is shown in **(a)** and Madrid is marked by the white cross.  Panel **(b)** shows a zoom-in of the Pinto landfill, outlined with a solid cyan line. Panel **(d)** displays the Valdemingómez Technology Park (VTP), marked with a solid magenta line. In the same panel, the closed Valdemingómez  landfill  is shown in pink and  open  Las Dehesas landfill is highlighted in purple. The flight path is shown in **(c)** whereby bluish colours represent the remote sensing (RS) part at $\sim 7.7$ km a.g.l. and greenish colours the in situ (IS) part at $\sim 1.6$ km a.g.l. (above ground level) of the flight. For better visualisation, the greenish in situ part is slightly shifted to the north-west because otherwise part of the legs would be hidden by RS legs. The map underneath is provided by Google Earth (Image © Landsat/Copernicus, Maxar Technologies).

(UTE LAS DEHESAS)" [3] and "DEPOSITO CONTROLADO DE RESIDUOS URBANOS DE PINTO"[4]

150 are $0.2$ t h$^{-1}$. Both sites are classified as "landfills" according to the European-Parliament (2006, Regulation (EC) 166/2006 E-PRTR). We assume that these reported values are representative for the two  investigated areas, which include landfills and waste treatment plants, as  other listed sources would not contribute significantly to the emissions according to the Spanish PRTR (Spanish-PRTR, 2025c). According to the European-Commission (2006), both landfills appear not to use strict IPCC reporting methods. They report the methods 'OTH' (for other measurement

155 or calculation methodology) and 'C' (for calculation) using 'issue factors', and 'CRM' (for measurement methodology by means of certified reference materials) and 'M' (for measurement) using 'electrochemical cells' for the Las Dehesas and Pinto landfills, respectively, in 2022. In addition, the reporting method for the Pinto landfill changed from 2021 to 2022, while 'OTH' and 'C' using 'an American EPA (Environmental Protection Agency) calculation model' was applied in 2021 instead of CRM as in 2022.
* * *
[3]E-PRTRSectorCode / Name: 5 / Waste and wastewater managements, mainActivityCode: 5.(d), landfills, EU-Registry Code: 003510000, PRTR Code: 3510 (Spanish-PRTR, 2025a).

[4]E-PRTRSectorCode / Name: 5 / Waste and wastewater managements, mainActivityCode: 5.(d), landfills, EU-Registry Code: 001636000, PRTR Code: 1636 (Spanish-PRTR, 2025b).

To properly investigate emissions from these two landfills, dedicated flight patterns where the aircraft is levelled (so called flight legs) were aligned perpendicular to the forecasted wind direction (Fig. 1). The overflight time was between 13:00 and 15:40 local time (11:00 to 13:40 UTC) on the 4[th] August 2022. This time window was chosen using knowledge of the weather forecast predicting stable winds around noon, which also favoured the observations by the passive remote sensing instrument due to the high position of the sun.

Prevailing wind direction during the flight was from approximately SSW, aligned with the two waste treatment areas (Fig. D2 in the Appendix D1 shows time-resolved wind profiles for the time measured in the measurement area). 
[revised manuscript text omitted]

where $n$ is the number of cross-sections of one leg. The errors of the fluxes for one cross-section $F_{\mathrm{M2D, cs}}$ and of the average flux of one leg $F_{\mathrm{M2D, leg}}$ are then computed by error propagation using the errors on the individual parameters used in Eq. 2, as

360   explained in Appendix F.

However, the approach above does not allow for a 1-to-1 comparison of fluxes between those determined form the measurements made by the imaging MAMAP2DL and those by the 1D CHARM-F instruments. Both datasets are actually distorted by the aircraft movement (i.e. predominantly the aircraft roll). The straight cross-sections introduced above for MAMAP2DL do not follow the distorted CHARM-F ground track. However, as seen in Fig. 4 (a), the CHARM-F ground track follows one

365  fixed  MAMAP2DL viewing angle approximately in the middle of the swath because the effect of the distortion is the same for both instruments. Therefore, Eq. 1 is directly applied to the measurements, with each parameter being evaluated individually for one measurement i.e. the wind speed, direction and length segment are not constant anymore. The resulting CHARM-F flux is then representative for this one leg. Definition of plume and background areas remain the same.

Independent of the applied approach for MAMAP2DL or CHARM-F data described above, the fluxes from several legs,
370  computed by Eq. 3, are then again averaged to derive the mean emission rates $F_{\text{M2D, ar-aver}}$ and $F_{\text{CHARM-F, ar-aver}}$[7] of certain areas in the measurement area for the respective instrument:

$$F_{\text{M2D or CHARM-F, ar-aver}} = \frac{\sum_i F_{\text{M2D or CHARM-F, leg}_i}}{p} \frac{\sum_{i=1}^{p} F_{\text{M2D or CHARM-F, leg}_i}}{p}, \tag{4}$$

where $p$ is the number of legs. This applies, for example, to the area in the lee of the two waste treatment areas, which is representative of the total emissions from the measurement area.

**3  Results**

**3.1  Observed Column Enhancements over Madrid and Source Attribution**

Figure 5 visualises the retrieved and orthorectified $CH_4$ column anomaly maps derived from the MAMAP2DL measurements (a, as described in Sect. 2.2.1) and the $XCH_4$, given as 3 seconds averages, derived from CHARM-F data (d, as described in Sect. 2.2.2) for the different remote sensing legs acquired at a flight altitude of $\sim 7.7\,\text{km a.g.l.}$. Both data sets clearly show
380  $CH_4$ enhancements (in red) located at or downwind of the waste treatment areas, whereas upwind or south-west of the Pinto landfill in the bottom left corner, there are no indications of inflow of external enhanced $CH_4$ in the measurement area. These observations are also confirmed by two legs flown in along wind direction at two different flight altitudes (see Appendix A and Fig. A1 for details). Especially for the Pinto landfill, there is a clear plume visible in both overflights, $\sim 2.5\,\text{h}$ apart from each other.

385  The highest $CH_4$ concentrations are observed at or close to landfills. $CH_4$ hot spots, with peak enhancements of around $17\,\%$, are located at the eastern part of the Pinto landfill according to the MAMAP2DL imaging data. This hot spot is also captured by the CHARM-F instrument with $XCH_4$ of up to $2.28\,\text{ppm}$.

The insets (b) and (c) show more details of the individual landfills including the locations of the highest column anomalies, which were identified in different overflights. Marked regions in the southeast of the landfills are areas which are, most prob-
390  ably, responsible for a large fraction of the observed emissions. They were selected by analysing the flight legs (those flown perpendicular to the wind direction as well as those flown in wind direction) over the landfills to find the ground scenes with the highest column enhancements (only ground scenes above a certain threshold are shown, see Fig. 5 for more details). The assumption is that the $CH_4$ is most concentrated just above or very close to a source, as it is not yet dispersed (horizontally),
* * *
[7]'ar-aver' stands for areal averages.

[Figure]

**Figure 5.** Retrieval results from the airborne remote sensing instruments. **(a)** and **(d)** show the retrieved $CH_4$ column anomalies from MAMAP2DL and the $XCH_4$ from CHARM-F, respectively. **(b)** and **(c)** are zoomed pictures of the two landfills including the MAMAP2DL  ground scenes in red with the largest anomalies only (only those larger than $\sim 4\%$ for VTP or Las Dehesas in **(b)**,  and $\sim 8\%$ for Pinto in **(c)**, see main text for details). The different colours of the borders around those ground scenes in **(b)** and **(c)** mark the enhancements observed in different flight legs. E.g., yellow represents the leg flown in along wind direction shown in Fig. A1, a. The small insets in **(b)** and **(c)** zoom in further detailing some activities across the areas with the largest observed enhancements. The shown Google Imagery was recorded in August 2022. The  waste treatment areas are encircled by different coloured solid lines: Cyan for the Pinto landfill;  magenta for the VTP;  pink for the Valdemingómez landfill; purple for the Las Dehesas landfill. The map underneath is provided by Google Earth (Image © Landsat/Copernicus, Maxar Technologies).

leading to the highest observed column enhancements. However, there is a residual uncertainty associated with this method, as by chance, high column enhancements could also be observed further away due to turbulent transport. Columns are also modulated by the prevailing wind speed at the time of release into the atmosphere and by local atmospheric turbulence. Both can change during a measurement flight. However, if the highest column anomalies during multiple over-flights point to the same region, there is a high probability that this region is acting as a source.

The Google Earth imagery recorded in August 2022 (the same month and year as our flight) clearly shows that these hot spots are directly located over active landfill areas where waste is deposited. The $CH_4$ plumes clearly begin over these areas of the Pinto and Las Dehesas landfills (see Figs. 5 and A1 and discussion on the step-wise increase of the fluxes in Sect. 3.3). However, we cannot exclude that other parts,  closed cells of the landfills or facilities located in these waste treatment areas, also contribute (weakly) to the observed $CH_4$ plume, but are partly masked by $CH_4$ released

further upwind. For example, in the north western part of the VTP, there are also hot spots identified (two ground scenes outlined in yellow in Fig. 5, b) from the along wind leg (Fig. A1, a), which could be advected there or released from the waste treatment plants (PLANT OF BIOMEIZATION La Paloma and Las Dehesa) immediately to the south. However, as a second overflight (Fig. A1, b) shows no enhancements, it is probably the former.

**3.2  Column Comparison between MAMAP2DL and CHARM-F**

Figure 5 shows a good visual agreement between the column anomalies of the passive MAMAP2DL and the $X\mathrm{CH_4}$ of the active CHARM-F instruments. In order to perform a more rigorous comparison between the two types of atmospheric $\mathrm{CH_4}$ columns, we convert the $X\mathrm{CH_4}$ partial columns derived from CHARM-F to total column anomalies (see Sect. 2.3). We then identify the ground scene in the MAMAP2DL swath which corresponds to the CHARM-F measurements, which are approximately located in the middle of the MAMAP2DL swath. This procedure ensures the selection of observations, where both instruments see similar ground scenes and air masses.

Figure 6 shows a typical example comparison for one leg. The two different types of observations have been processed as explained previously, i.e the plume anomalies have been processed as described in Sect. 2.3 and the $\mathrm{CH_4}$ fluxes have been estimated as described in Sect. 2.4. The shown background normalised column anomalies agree well within their respective errors inside and outside of the plume. Even more pronounced structures in the $\mathrm{CH_4}$ concentration, as encountered on the right hand side ($\sim 6$ to $15\,\mathrm{km}$ distance), are identified by both instruments. The fluxes from the two shown cross-sections deviate by only $0.1\,\mathrm{t\,h^{-1}}$ or $1\,\%$.

More generally, when comparing fluxes estimated using measurements of MAMAP2DL with those derived from CHARM-F observations from all flight legs (see Fig. E1), the averaged absolute difference between them is $\sim 1.2\,\mathrm{t\,h^{-1}}$ or $\sim 13\,\%$ excluding the flight legs upwind and directly over the Pinto landfill (see Sect. 3.3 for reasoning).  These differences may be due to (a) different but overlapping opening angles of the two instruments and the resultant spatial resolution of the ground scene widths of $110\,\mathrm{m}$  and $10\,\mathrm{m}$ , respectively, or (b) different paths through the atmosphere of the electromagnetic radiation used to measure methane absorption, or (c) differences in the algorithms used to retrieve the columns, e.g. how they deal with variable surface reflectivity etc. Consequently, observed air masses are different.

Typically, the errors of the fluxes are around or below $30\,\%$ of the respective flux and are similar for MAMAP2DL and CHARM-F.

**3.3  Derived Landfill Emission Rates**

Using imaging MAMAP2DL observations, we also computed the fluxes within the different legs (see Sect. 2.4). The results are summarised in Fig. 7, which also includes the cross-sectional fluxes derived from the CHARM-F instrument already computed and introduced in Sect. 3.2.

Based on the MAMAP2DL observations, the fluxes exhibit a step-wise increase at the location of landfills as expected (from left - upwind, to right - downwind). The upwind leg at $-5\,\mathrm{km}$ shows no significant inflow of enhanced $\mathrm{CH_4}$ and a steep increase

[Figure]

**Figure 6.** The background normalised $CH_4$ column anomalies for CHARM-F (orange) and the co-located MAMAP2DL (blue) observations for one flight leg collected between 11:42 and 11:46 UTC are shown. Vertical dotted lines separate the plume and background areas. Shaded areas represent the random error (single measurement precision) of the retrieved column anomalies of the respective instrument. The computed fluxes for the cross-sections according to Eq. 1 and the corresponding errors (MAMAP2DL: Eq. F1, and CHARM-F: Eq. F10) are given by the text insets. For graphical presentation only, the MAMAP2DL data has been smoothed by a 500 m kernel to match the spatial resolution of CHARM-F in along flight direction. Flux, error and uncertainty range is, however, based on the $\sim 110 \times 110 \, m^2$ data.

directly over the Pinto landfill. Between the Pinto and the VTP the flux or emission rate stabilises at $4.2 \, t \, h^{-1}$ ($\pm 38 \, \%$) before increasing to around $12.1 \, t \, h^{-1}$ ($\pm 27 \, \%$) on average at and after the Las Dehesas landfill. However, the cross-sectional fluxes show some variability from flight leg to flight leg (see the bold horizontal coloured lines, representing averaged values over one MAMAP2DL leg) and variability within one leg (see the thin solid coloured lines). Furthermore, the retrieved column anomalies in Fig. 5 as well as the cross-sectional fluxes in Fig. 7 show no sign of accumulation of $CH_4$ as, for example, in the valley between the two landfills (see Sect. 2.1.1 for a brief discussion of the local topography). Adding the fluxes derived from the CHARM-F observations to the figure (coloured stars) reveals very good agreement between active and passive remote sensing (thin solid coloured lines) data as already indicated in Sect. 3.2. Computing average fluxes or emission rates from the CHARM-F observations alone yields $5.2 \, t \, h^{-1}$ ($\pm 37 \, \%$) for the Pinto landfill and $13.3 \, t \, h^{-1}$ ($\pm 26 \, \%$) for both waste treatment areas combined.

The first two flight legs were deliberately designed so that, in the blue leg, CHARM-F sampled background conditions upwind of the Pinto landfill, while MAMAP2DL already partially covered the source area. Conversely, during the green leg, CHARM-F measured directly over the Pinto landfill, whereas the MAMAP2DL swath still included parts of the upwind background. For the averaged flux between the two landfills, the flight leg directly over the Pinto landfill (i.e. the green lines and star in Fig. 7) has been omitted. There, the plume might be still restricted to the surface and the wind speed is highly biased due to the strong vertical wind gradient (see Fig. D2, a). Over the Las Dehesas landfill, although there are new emissions

[revised manuscript text omitted]

Our analysis shows the importance of knowledge and understanding of the characteristics of the BL during a measurement. While we had the privilege to compare in situ wind measurements with model data, even though at a later time of the day, emission estimates based on satellite data rely on atmospheric parameters from models. Moreover, there is usually no possibility to validate the conditions during measurement times. Systematic errors such as the BLH in combination with the strong vertical wind speed gradient, influence the estimated emission rates. They need to be identified and taken into account to minimise their impact.

Our calculated emission rates are in good agreement with previous top-down estimates, even though, strictly speaking, they are only valid for the time of the overflight. The prevailing winds in combination with the vertical distribution of the $CH_4$ emissions in the BL could introduce a common error in our emission rate estimate but not to an extent that we approach reported values assuming constant emission throughout the year, at least on the 4[th] August 2022 during our flight. The fact that our emission estimates are a factor of 40 to 50 higher than reported values (assuming constant emissions) supports the inference that a major part of the emissions are unreported, especially as the reported emissions in E-PRTR fell by a factor of 10 from 2021 to 2022.

The methods used in this work are also applicable to planned satellite missions such as CO2M and MERLIN. Nevertheless, the generally coarser resolution on the ground will lead to a reduced sensitivity for emission rates, particularly for somewhat dispersed sources. Also, the combination of active and passive remote sensing on a single satellite platform would show promise for the future, as the advantages of both methods can be synergistically exploited.

[revised manuscript text omitted]

$S_{\text{on}}$, $S_{\text{off}}$ are backscattered  signals and $E_{\text{on}}$, $E_{\text{off}}$ are internal energy reference  measurements at the on- and offline wavelength, respectively. $\Delta\tau$ can on the other hand also be described in terms of the molecular absorption cross-section at both wavelengths ($\sigma_{\text{on}}$ and $\sigma_{\text{off}}$) and the number density $n_{\text{CH}_4}$ or the dry-air mixing ratio of methane $r_{\text{CH}_4}$.

$$\Delta\tau = \int_{h_0}^{h_1} n_{\text{CH}_4} \underbrace{(\sigma_{\text{on}} - \sigma_{\text{off}})}_{=\Delta\sigma}\ \mathrm{d}h, \tag{C2}$$

$$= \int_{h_0}^{h_1} \frac{r_{\text{CH}_4}}{(1 + r_{\text{H}_2\text{O}})} \cdot n_{\text{air}} \cdot \Delta\sigma\ \mathrm{d}h. \tag{C3}$$

The integral runs over the air column between aircraft ($h_1$) and ground ($h_0$). In Eq. C3 the number density of greenhouse-gas molecules has been expressed in terms of the number density of air molecules $n_{\text{air}}$ and the dry-air molar mixing ratio of the greenhouse-gas species, while also accounting for the dry-air molar mixing ratio of water

$$n_{\text{air}} = (1 + r_{\text{H}_2\text{O}}) \cdot n_{\text{dry air}}. \tag{C4}$$

Using the general gas equation ($n_{\text{air}} = \frac{p}{k_{\text{B}} \cdot T}$) $n_{\text{air}}$ can be expressed in terms of pressure $p$ and temperature $T$. Furthermore, the integral over altitude can be transformed into a pressure integral ($\mathrm{d}h = \frac{\mathrm{d}h}{\mathrm{d}p} \cdot \mathrm{d}p = -\frac{k_{\text{B}} \cdot N_{\text{A}} \cdot T}{p \cdot M_{\text{air}} \cdot g}\ \mathrm{d}p$).

$$\Delta\tau = \int_{p_1}^{p_0} \frac{r_{\text{CH}_4} \cdot N_{\text{A}}}{(1 + r_{\text{H}_2\text{O}}) \cdot M_{\text{air}} \cdot g} \cdot \Delta\sigma\ \mathrm{d}p. \tag{C5}$$

Here, $k_{\text{B}}$ is the Boltzmann constant, $N_{\text{A}}$ the Avogadro constant, $g$ the gravitational acceleration and $M_{\text{air}}$ the average molar mass of air. Next, molar mass is expressed as molecular mass ($m_{\text{air}} = \frac{M_{\text{air}}}{N_{\text{A}}}$) and dry air is discriminated from water vapour.

$$\Delta\tau = \int_{p_1}^{p_0} \frac{r_{\text{CH}_4}}{(1 + r_{\text{H}_2\text{O}})} \cdot \frac{\Delta\sigma}{\underbrace{m_{\text{air}}}_{=\frac{m_{\text{dry air}} + r_{\text{H}_2\text{O}}\, m_{\text{H}_2\text{O}}}{1 + r_{\text{H}_2\text{O}}}}} \cdot g\ \mathrm{d}p, \tag{C6}$$

$$= \int_{p_1}^{p_0} \frac{r_{\text{CH}_4}}{(m_{\text{dry air}} + r_{\text{H}_2\text{O}} \cdot m_{\text{H}_2\text{O}})} \cdot \frac{\Delta\sigma}{g}\ \mathrm{d}p. \tag{C7}$$

Finally, $r_{CH_4}$ is pulled out of the integral by replacing it with a (weighted) column average $XCH_4$, which is thus defined.

$$\Delta\tau = XCH_4 \int_{p_1}^{p_0} \frac{\sigma_{on}(p,T) - \sigma_{off}(p,T)}{g\,(m_{dry\ air} + r_{H_2O} \cdot m_{H_2O})}\ dp, \tag{C8}$$

$$= XCH_4 \int_{p_1}^{p_0} W(p,T)\ dp\,. \tag{C9}$$

The quotient remaining in the integral is the so-called weighting function

$$W(p,T) = \frac{\sigma_{on}(p,T) - \sigma_{off}(p,T)}{g \cdot (m_{dry\ air} + m_{H_2O} \cdot r_{H_2O})}, \tag{C10}$$

which carries the terms that are pressure/altitude dependent but known to some degree. In the standard data analysis routines of CHARM-F, the absorption cross-sections are calculated based on the spectroscopic data sets GEISA20 and Vasilchenko (Delahaye et al., 2021; Vasilchenko et al., 2023), and the state of the atmosphere (vertical structure at measurement location) is extracted from the ECMWF Integrated Forecasting System (IFS).

The partial-column weighted-average molar mixing ratio is therefore described by

$$XCH_4 = \frac{\Delta\tau}{\int_{p_1}^{p_0} W(p,T)\ dp}. \tag{C11}$$

**C2 Column Anomalies from CHARM-F Measurements**

This section describes the details of how we convert $XCH_4$, as measured by CHARM-F, into a column anomaly of the dry-air $CH_4$ molar mixing ratio for the total column. The reference for the calculation of anomaly is the methane concentration from the Copernicus Atmosphere Monitoring Service (CAMS) global inversion model (Segers and Houweling, 2023). Based on the dry-air molar mixing ratio from CAMS ($r_{CH_4,CAMS}$) we calculate a column-averaged molar mixing ratio between surface (sfc) and flight altitude (flh), using number density of air molecules from CAMS as weight $n_{air}$:

$$XCH_{4\,CAMS} = \frac{\int_{sfc}^{flh} r_{CH_4,CAMS} \cdot n_{air}\ dz}{\int_{sfc}^{flh} n_{air}\ dz}. \tag{C12}$$

On this basis, the partial column anomaly $A_{pc}$ is calculated.

Due to the temperature and pressure dependence of the spectroscopic properties of methane, $XCH_{4\,CHARM-F}$ and $XCH_{4\,CAMS}$ are weighted somewhat differently along the column (see Fig. C1). Nevertheless, in the hypothetical case, where methane is homogeneously distributed along the surveyed column, they give the exact same result by definition of the weighting function. To compensate for this small bias (few percent), which only affects the anomalous part of the column concentration, based on the CHARM-F weighting function $W(p,T)$, we calculate a correction factor.

$$C_W = \frac{\frac{\int_{sfc}^{flh} W(p,T) n_{air}\ dz}{\int_{sfc}^{flh} n_{air}\ dz}}{\frac{\int_{sfc}^{blh} W(p,T) n_{air}\ dz}{\int_{sfc}^{blh} n_{air}\ dz}}. \tag{C13}$$

[revised manuscript text omitted]
}}^2(n) + \delta F_{\mathrm{col\text{-}ac}}^2 + \delta F_{\mathrm{col\text{-}cf}}^2\end{aligned}} \sqrt{\begin{aligned}&\delta F_u^2 + \delta F_\alpha^2 + \delta F_{\mathrm{blh}}^2 + \delta F_{\mathrm{bg}}^2 \\ &+ \delta F_{\mathrm{col\text{-}pr}}^2(k) + \delta F_{\mathrm{col\text{-}ac}}^2 + \delta F_{\mathrm{col\text{-}cf}}^2\end{aligned}}. \tag{F1}$$

**Table F1.** Summary of relevant error sources and quantities for the flux computation.

| Parameter | Description |
| --- | --- |
| $\delta F_u$ | error in wind speed |
| $\delta F_\alpha$ | error in wind direction |
| $\delta F_{\mathrm{blh}}$ | error in boundary layer height |
| $\delta F_{\mathrm{bg}}$ | error in the selected background region(s) |
| $\delta F_{\mathrm{col\text{-}pr}}$ | column single measurement precision |
| $\delta F_{\mathrm{col\text{-}ac}}$ | remaining systematic column offset (Appendix B2, only MAMAP2DL) |
| $\delta F_{\mathrm{col\text{-}cf}}$ | error in the conversion factor (Appendix B3, only MAMAP2DL) |
| $\delta F_{\mathrm{M2D,\,cs}}$ | error of one cross-section (MAMAP2DL) |
| $\delta F_{\mathrm{M2D,\,leg}}$ | error of one leg |
| $\delta F_{\mathrm{M2D,\,css}}$ | errors from the single cross-sections |
| $\delta F_{\mathrm{M2D,\,atm,\,css}}$ | error due to atmospheric turbulence within one leg |
| $\delta F_{\mathrm{M2D,\,ar\text{-}aver}}$ | error of one area (or areal average) |
| $\delta F_{\mathrm{M2D,\,legs}}$ | error from the single legs |
| $\delta F_{\mathrm{M2D,\,atm,\,legs}}$ | error due to atmospheric turbulence across area |
| $\delta F_{\mathrm{CHARM\text{-}F,\,leg}}$ | error of one leg |
| $\delta F_{\mathrm{CHARM\text{-}F,\,ar\text{-}aver}}$ | error of one area (or areal average) |
| $\delta F_{\mathrm{CHARM\text{-}F,\,legs}}$ | errors from the single legs |
| $\delta F_{\mathrm{CHARM\text{-}F,\,atm,\,legs}}$ | error due to atmospheric turbulence across area |

If the emission source is constant, the actual uncertainty of the flux of one cross-section at a certain position is also influenced by atmospheric variability or turbulence in the atmosphere (Krautwurst et al., 2021; Wolff et al., 2021; Krautwurst et al., 2017; Matheou and Bowman, 2016). This is also visible in the imaging data and at this point not covered by the error $\delta F_{\mathrm{M2D,\,cs}}$ (Eq. F1). However, this error component is reduced by taking spatially and/or temporally independent cross-sections. We estimate this factor as 1-sigma standard deviation (SD) from the cross-sections of one entire MAMAP2DL leg itself:

$$\delta F_{\mathrm{M2D,\,atm,\,css}} = \frac{\mathrm{SD}(F_{\mathrm{M2D,\,cs},j})}{\sqrt{m_{\mathrm{eff}}} } \frac{\mathrm{SD}(F_{\mathrm{M2D,\,cs},j})}{\sqrt{n_{\mathrm{eff}}}} \tag{F2}$$

[revised manuscript text omitted]

---

## Author Comment (AC2)

Referee/Reviewer #2

We would like to thank the anonymous referee #2 for the careful reading and valuable comments, which we address point-by-point.

In the following, the reviewer comments are in *italics*, the answers in plain text and the proposed changes / new text in blue. References to Sections or Figures in our answers refer to the discussion version of the manuscript.

*This manuscript by Krautwurst et al. (2024) shows the first use of both active lidar and passive remote sensing on aircraft to estimate landfill methane in Madrid with verification using in situ instruments. The manuscript is generally well written and thoroughly explains the instrumentation and methodologies used. The described research reflects important work to compare instrument capabilities, especially for complex emission sources like landfills. There is growing interest in tracking emissions reductions, and the work by Krautwurst et al. will be valuable for establishing the methodologies available for accurate quantification of methane emissions.*

*The manuscript currently provides thorough description of instruments, retrievals, and flux estimates, and I recommend publication. The main recommended revision is to add some discussion of how sources on the landfill were identified. Currently, there is little discussion of how sources at the facility were geolocated and attributed and the uncertainties in these methodologies. As the identification of the sources at the landfill is a major conclusion, there needs to be some additional text added to the body of the manuscript explaining how this was done. Detailed comments:*

See our answers to the comments below.

###########################################################################

*Line 34: Is this EU number for landfills? Solid waste? Or total waste sector? And can the actual reported number be added in addition to the relative 24% (as this would allow comparison to other regions of the world)?*

The 24% are representative of the total waste sector (IPCC category: 5). Solid waste disposal (IPCC category: 5A) would account for 18 %.

We have amended the sentence as follows:

… reported $CH_4$ emissions from waste (IPCC sector 5) are 97 Mt $CO_2$,eq yr−1 (or 3.5 Mt yr−1)[1] and still account for $\sim$ 24 % (or $\sim$ 18 % if only solid waste disposal, IPCC sector 5A, is considered) of the anthropogenic $CH_4$ emissions in the European Union in 2022 (EEA, 2024).

[1]Converting $CO_2$,eq to $CH_4$ by using a factor of 28.

###########################################################################

*Line 43: If not specified later in the manuscript, it would be helpful to know if the Madrid facilities, specifically, use IPCC-type methods.*

We agree that some additional information on the landfill reporting may be useful. For example, the literature review revealed that the Pinto landfill changed its reporting methodology from 2021 to 2022, which may have been the reason for the almost 40-fold decrease in reported emissions over the same period.

We have added the information in Sect. 2.1.2, where the targets are described in more detail and with additional context in Sect. 4, where reporting in recent years is discussed.

According to the Spanish PRTR (Spanish-PRTR, 2025c), the combined annual reported 2022 $CH_4$ emissions for the two facilities "VERTRESA-URBASER, S.A. UTE (UTE LAS DEHESAS)"[3] and "DEPOSITO CONTROLADO DE RESIDUOS URBANOS DE PINTO"[4] are 0.2 t h−1. Both sites are classified as "landfills" according to the European-Parliament (2006, Regulation (EC) 166/2006 E-PRTR, Annex I). We assume that these reported values are representative for the two investigated areas, which include landfills and waste treatment plants, as other listed sources would not contribute significantly to the emissions according to the Spanish PRTR (Spanish-PRTR, 2025c). According to the European-Commission (2006), both landfills appear not to use strict IPCC reporting methods. They report the methods 'OTH' (for other measurement or calculation methodology) and 'C' (for calculation) using 'issue factors', and 'CRM' (for measurement methodology by means of certified reference materials) and 'M' (for measurement) using 'electrochemical cells' for the Las Dehesas and Pinto landfills, respectively, in 2022. In addition, the reporting method for the Pinto landfill changed from 2021 to 2022, while 'OTH' and 'C' using 'an American EPA (Environmental Protection Agency) calculation model' was applied in 2021 instead of CRM as in 2022.

[3]E-PRTRSectorCode / Name: 5 / Waste and wastewater managements, mainActivityCode: 5.(d), landfills, EU-Registry Code: 003510000, PRTR Code: 3510 (Spanish-PRTR, 2025a).
[4]E-PRTRSectorCode / Name: 5/Waste and wastewater managements, mainActivityCode: 5.(d), landfills, EU-Registry Code: 001636000, PRTR Code: 1636 (Spanish-PRTR, 2025b).

Interestingly, the reported emissions of the Pinto landfill site decreased by a factor of almost 40 from 2021 to 2022, which could potentially be related to the change in reporting methodology (see Sect. 2.1.1).

###############################################################################

*Line 63: Some discussion of the value of active remote sensing in comparison to passive remote sensing would be valuable.*

Thank you for raising this important point. We have amended the paragraph from line 63 as follows:

Active remote sensing instruments are independent of sunlight because they use a laser as their own source of electromagnetic radiation. In contrast to airborne and satellite-borne passive instruments, they can measure during day and night, across all seasons and latitudes. They provide ranging capabilities resulting from the precise

measurement of the propagation time of the emitted light and, due to their narrow field of view, measure between clouds. IPDA (integrated-path differential absorption) lidars potentially provide highly accurate measurements without varying biases: the exceptions are those introduced by small differences in the scattering and reflectivity of the ground scene and any inaccuracies in the knowledge of the absorption cross-sections.

##############################################################################

*Section 2.1.1: This description of CoMet 2.0 Arctic seems unrelated to the manuscript focus.*

We agree and have deleted the paragraph as the information on the used instrumentation used is given later in Section 2 (and was partly outlined earlier). However, the information on the HALO aircraft has been moved to Section 2.1.

Below we describe the waste treatment facilities that were the targets of the research flights, and the flight strategy used to derive their methane emission rates. We then describe the instruments used, which were installed aboard the German research aircraft HALO (High Altitude and Long Range Research Aircraft, operated by the DLR, Deutsches Zentrum für Luft- und Raumfahrt, type: Gulfstream G550). The retrieval algorithms used to derive the CH4 columns are explained next. This is followed by a description of the observed CH4 columns. Finally, we derive the methane fluxes, i.e. the methane emission rates, using the plume cross-sections.

##############################################################################

*Figure 1: A color key in the figure would be helpful.*

It has been added.

##############################################################################

*Line 121: Adding waste-in-place metrics for the 2 landfills and adding activity per year rates for both landfills (in the same units) would contextualize the landfills better.*

We are not sure what the referee exactly means by "activity per year rates". We assume that he is referring to the waste dumped over a certain period. We added these values for the VTP (and the Las Dehesas landfill) and the Pinto landfill site. We could, however, not find information or data on waste-in-place or the mass of waste material deposited.

The latter [VTP] is a waste treatment complex that accepts around 1,222 kt (Madrid, 2022) of waste, of which around 140 kt were deposited at the Las Dehesas landfill site in 2022 (Spanish-PRTR, 2025a), ...

It [Pinto] opened in 1987, is still operational with around 53 kt of waste being dumped in 2022 (Spanish-PRTR, 2025b), …

##############################################################################

*Line 125: Clarify if, for readers unfamiliar with EPRTR, there is potential for the facility reported numbers to be missing methane sources. For example, are some sources not reported due to regulatory limitations, or should we expect the reported number to be comparable to measurements?*

Very good point. For CH4 emissions from landfills, operators have to report their emissions if they emit more than 100 tCH4/year or receive more than 10 t/d or have a capacity of more than 25 kt [European Parliament, 2006]. Other source types also have specific thresholds or reporting requirements, which can be found in European Parliament [2006]. As a result, potential CH4 sources, e.g., emitting smaller amounts of CH4 than specified in the Regulation, are not covered by reporting. Just for clarification, the sentence in line 125 refers to additional sources over the Pinto landfill or the VTP (such as waste treatment plants or biogas plants), not over the entire measurement area. Outside the coloured areas (Fig. 1), there are some additional CH4 sources according to the Spanish PRTR that we could interfere with our observed plume [Spanish-PRTR, 2025], but their emissions are too small to contribute significantly to the calculated emission rate, if they emit CH4 at all. Furthermore, our data does not indicate a significant contribution from these potential sources either, or any other sources unknown to PRTR, except for the northeastern most signal discussed in Sect. 3.4.2.

The question as to whether we would expect the measurements to give similar emission rates to those reported is tricky. If the regulations and therefore the reporting are correct, we would expect similar values. However, as we know, reporting, especially when based on bottom-up methods, can yield uncertain estimates. However, in this case, we expected higher values than reported based on the ESA web story released the year before.

We have added the reporting regulations to the introduction, where we introduce the (E-)PRTR (Line 43):

Landfill facility emissions must be reported to the authorities (E-PRTR) in the European Union to comply with the objectives of EU directives if they meet certain criteria, such as emitting more than 100 kt CH4 yr$^{-1}$ or receiving more than 10 t d$^{-1}$ or having a total capacity of more than 25 kt (European-Parliament, 2006).

################################################################################

*Line 132: Further information on winds would be valuable in this section. For example, visuals showing the wind speeds and directions over time and space for the study area (can be added to Appendix rather than body of text). Some discussion of topography may also be valuable. For example, is it possible that the second landfill location is a topographic low and the methane from upwind is just pooling there?*

We agree and have added a reference in line 132 to the temporally and vertically resolved winds in the measurement area shown in Fig. D2 in Appendix D. The topography in this area shows some variability. Based on SRTM, the surface elevation along the entire flight path over Madrid is around 500 to 1000 m a.s.l. with a mean of around 670 m a.s.l. and a 1-sigma standard deviation of 80 m. Based on Google Earth topography, the highest elevation areas are tentatively located at the eastern and western locations of the flight pattern and have no direct influence on the

plume. The surface elevation around the landfills and the observed plume signal is much less variable, ranging between 550 and 700 m a.s.l. Between the Pinto landfill and the VTP, there is a small valley (about 70 m lower than the surroundings) with a steep increase at the start of the VTP and then remaining relatively stable between 600 and 650 m a.s.l. until the position of the northernmost leg. The most likely location for methane accumulation, if any, would be upwind of the VTP, but there is no evidence for this in our measurements. It is likely that the high solar insolation has already transported the main plume to higher altitudes.

We have added some context concerning the topographical situation around the landfills to the manuscript in Sect. 2.1.2 and a statement on the pooling in Sect. 3.3.

 ... (Fig. D2 in the Appendix D1 shows time-resolved wind profiles for the time measured in the measurement area).

The topography around these landfill sites shows some variability, ranging from about 550 to 700 m a.g.l. with a small valley between the two sites and a steep rise just south of the VTP, according to Google Earth.

Furthermore, the retrieved column anomalies in Fig. 5 as well as the cross-sectional fluxes in Fig. 7 show no sign of accumulation of CH4 as, for example, in the valley between the two landfills (see Sect. 2.1.1 for a brief discussion of the local topography).

###################################################

*Line 297: It's unclear what the authors are defining as a "plume" versus "enhancement" and how this relates to the differentiation between "flux" and "emission rate".*

In general, we retrieve from the measured spectra the CH4 column anomalies relative to the local background. These may be positive or negative. Positive anomalies or deviations are called "enhancements". If these enhancements (visible in multiple ground scenes) are caused by a CH4 source, are connected and have a plume-like shape, they are also called a "plume". Such plumes are always CH4 enhancements, but a CH4 enhancement does not have to be a plume. In Eq. 1, we then integrate over contiguous CH4 enhancements or "plumes", along a straight line (i.e. a cross-section) to calculate the mass flow rate of CH4 molecules or the flux through this cross-section. If this flux can be associated with a source, we refer to it as the methane emission rate of that source (see also line 291ff).

We went through the document and unified the terminology where appropriate.

##############################################################################

*Figure 5: It would be helpful to separate 5a-b from 5c-d and provide more explanation of what 5c-d is meant to show. Keys for the color outlines are needed. Were the 5c-d enhancements the highest enhancements of all observed methane, or just those above the landfills? Why do the shapes of enhancement regions vary and how were they delineated? This comment relates to the above comment about providing more details on how source locations were identified on the landfills.*

We are not quite sure why 5a-b should be separated from 5c-d. 5c actually belongs to 5a (as does 5b) – all subplots show MAMAP2DL observations, in one form or another. You could separate 5d (showing CHARM-F observations) from 5a-c, but then you would lose the direct comparison between the MAMAP2DL and CHARM-F columns. Presumably, the reviewer is referring to 5b-c when talking about the isolated enhancements.

The shown enhancements are the highest for all observations, i.e., not limited to landfills. There are no enhancements larger than ~8% downwind of the Pinto landfill and no enhancements larger than 4% downwind of the VTP.

The basic assumption for the source identification based on the observed column enhancements is that just above or very close to a potential source, the CH4 is most concentrated as it has not yet dispersed (horizontally), leading to the highest observed column anomalies. Of course, by chance, high column concentrations could also be observed elsewhere because of turbulent transport, but if the highest column anomalies during multiple overflights point to the same region, there is a high probability that this region is acting as a source. In addition, the observed column enhancement is also modulated by the actual wind speed at which the CH4 is released into the atmosphere, leading to smaller CH4 columns (also at the source) for higher wind speeds, and by the strength of local turbulence.

The colour of the outlines is less important and should only show that the observations belong to different legs (i.e., the yellow one belongs to the flight leg in wind direction actually not shown in Fig. 5 but in the Appendix in Fig. A1, a).

With respect to the question "... shapes of enhancements vary": As mentioned above, the different colour contours indicate different legs, which were acquired during different overflights at different times. In reality, two different observations of an emission plume will never give the same concentration measurements or the same plume shape due to turbulence, which is spatially and temporally variable. Therefore, even for an idealised emission source (with constant emissions, which may not need to be the case for the landfills studied), the plume (location, shape, enhancements) will change from overflight to overflight. In addition, the position of the emission may also vary with the activity across the landfill.

Regarding the question "... delineated": MAMAP2DL provides approx. square ground scene sizes. In Fig. 5b-c only those ground scenes are shown that are above a certain threshold as defined in the caption.

We have amended Sect. 3.1 and the caption of Fig. 5 accordingly.

They were selected by analysing the flight legs (those flown perpendicular to the wind direction as well as those flown in wind direction) over the landfills to find the ground scenes with the highest column enhancements (only ground scenes above a certain threshold are shown, see Fig. 5 for more details). The assumption is that the CH4 is most concentrated just above or very close to a source, as it is not yet dispersed (horizontally), leading to the highest observed column enhancements. However, there is a residual uncertainty associated with this method, as by chance, high column enhancements could also be observed further away due to turbulent transport. Columns are also modulated by the prevailing wind speed at the time of release into the atmosphere

and by local atmospheric turbulence. Both can change during a measurement flight. However, if the highest column anomalies during multiple overflights point to the same region, there is a high probability that this region is acting as a source.

The CH4 plumes clearly begin over these areas of the Pinto and Las Dehesas landfills (see Figs. 5 and A1 and discussion on the step-wise increase of the fluxes in Sect. 3.3). However, we cannot exclude that other parts, closed cells of the landfills or facilities located in these waste treatment areas, also contribute (weakly) to the observed CH4 plume, but are partly masked by CH4 released further upwind. For example, in the north western part of the VTP, there are also hot spots identified (two ground scenes outlined in yellow in Fig. 5, b) from the along wind leg (Fig. A1, a), which could be advected there or released from the waste treatment plants (PLANT OF BIOMEIZATION La Paloma and Las Dehesa) immediately to the south. However, as a second overflight (Fig. A1, b) shows no enhancements, it is probably the former.

###################################################################################

*Line 354-357: These two sentences are confusing and contradictory. The authors point out a source but then say it may not be that source and could be any part of the landfill. Is this due some aspect of method uncertainty (e.g., geolocation of source, wind direction, identification of enhanced pixels)?*

This is a good and important point. To unambiguously identify sources over landfills, one would need ground scene sizes in the sub-metre range. Based on the plume shape alone (assuming at least some wind to have a reasonable transport of CH4 away from the source in one direction), the source of the emissions could usually be identified. For MAMAP2DL, the ground scene size is about 110 m, and the accuracy of the orthorectification has also been estimated to be about one ground scene size (see Sect. B4) limiting the exact geolocation capabilities. For the Pinto landfill in the south, the start of the plume, and therefore at least 'one source', is in the south-eastern part of this landfill, as shown in Figs. 5a and A1 a, b. This is also supported by the already available argumentation in Sect. 3.3, line 381ff, discussing the step-wise increase in fluxes over the landfills. However, it cannot be excluded that other parts of the landfill are also emitting CH4, which would then be covered/masked by the previously emitted CH4. The same reasoning applies to areas outside the landfill, but is much less likely if no potential sources on the ground can be identified. Analysing the CH4 column anomalies at the VTP (Fig. A1), there is again a plume starting in the south-eastern part, which corresponds to the highest observed enhancements (yellow outlined ground scenes in Fig. 5, b) and is over the active landfill. However, there is also a cloud of enhanced CH4 over the north-western part with smaller enhancements. This cloud could have been advected from the active landfill part, but could have also originated at this location from the waste treatment plants located there.

Overall, there is strong evidence that the active parts of the landfills contribute significantly to the observed emissions, but it can neither be confirmed nor refuted that there are also contributions from other parts of the waste treatment areas. See also the comment to Fig. 5 above for more detailed adaptations in the manuscript.

###################################################################################

*Figure 7: The flux estimates above the Pinto landfill seem to show that the passive remote sensing is seeing a combination of background and enhancement from landfill whereas active lidar is only seeing the enhancement. Is this the influence of the narrower instrument view? Some discussion on the potential interpretation difficulties if the active lidar "misses" the enhancement when there are steep gradients (and/or need for tight flight patterns to avoid this) would be valuable.*

We thank the reviewer for highlighting this important aspect. We agree, and indeed this point has been discussed in the literature in the context of both methane and carbon dioxide emission quantification (e.g., Krautwurst et al., 2021; Wolff et al., 2021). In particular, Wolff et al. (2021) showed that under turbulent atmospheric conditions, the spatial distribution of enhanced concentrations within a plume can be highly heterogeneous. As a result, individual overflights may intersect either strongly enhanced or only weakly enhanced sections, depending on the local variability. This can lead to significant differences in the retrieved fluxes/emission rates despite constant emissions, unless results are averaged over multiple plume crossings.

To address this potential limitation, we have added a brief discussion to the Introduction section of the revised manuscript.

However, Wolff et al. (2021) showed that under turbulent conditions, the spatial distribution of enhanced concentrations within an exhaust plume may be highly heterogeneous. As a result, a single overflight may sample sections with stronger or weaker enhancements purely as a result of the local variability. In some cases, the true emission signal only emerges after averaging over a high number of overflights.

To account for this potential limitation, in the analysis, we combine active lidar with passive imaging spectrometry, both designed to capture atmospheric $CH_4$ column gradients. Thus, we obtain both high-precision transects and spatial context, which supports a more robust interpretation of the observed $CH_4$ column enhancements. Moreover, these remote sensing measurements are complemented by auxiliary in situ measurements of CH4, CO2 and 3D winds in support of the remote sensing data.

There, we now explicitly state that one motivation for the synergistic deployment of active lidar and passive imaging spectrometry in our campaign is to overcome the interpretation challenges arising from plumes having spatial inhomogeneities. The combination enables the analysis to benefit from the high-precision transects of the lidar while using the wide swath of the imaging spectrometer to provide spatial context and thereby support a more robust interpretation of the observed CH4 column enhancements.

As the reviewer also correctly notes, the narrower field of view of CHARM-F limits its spatial sampling. Specifically, CHARM-F measures a single ground scene approximately in the middle of the MAMAP2DL swath, which itself consists of 28 cross-track ground scenes. In the case of the Pinto landfill observations discussed in Figure 7, this difference is further influenced by the flight pattern: the flight legs were deliberately designed so that, during the first leg (blue), CHARM-F sampled upwind of the landfill to assess potential background conditions, while MAMAP2DL already partially covered the source area. Conversely, the second leg (green) directly overflew the Pinto landfill

with CHARM-F, while the MAMAP2DL swath still included parts of the upwind background. This difference in spatial coverage is now briefly discussed in the manuscript (line 393). Specifically, we added the following text about Fig. 7:

The first two flight legs were deliberately designed so that, in the blue leg, CHARM-F sampled background conditions upwind of the Pinto landfill, while MAMAP2DL already partially covered the source area. Conversely, during the green leg, CHARM-F measured directly over the Pinto landfill, whereas the MAMAP2DL swath still included parts of the upwind background.

############################################################################

*Section 3.4.2: Would it not be possible for the other methane sources in the region to be significantly contributing but not be individually detectable? For example, the enhancement from a single smaller source may not be detectable over that source but as multiple of these sources emit in your domain, could their combined methane lead to detectable enhancements downwind if the winds are stable? It would be valuable to provide some metrics (e.g., emission rate) of the expected potential contribution of the other methane sources in the domain.*

We agree with the reviewer that weak sources, if present in very large numbers, can contribute to the observed emission rate. However, to contribute significantly to the calculated emission rate, the sources would have to be within the range of observed $CH_4$ enhancements and not outside in in the background area. We have attempted to estimate these (additional) emissions by analysing the $CH_4$ emissions reported in EDGAR v8.0 (Crippa et al., 2023) over the entire area covered by our measurement flight, by aggregating all the emissions reported in EDAGR in that area. Figure 1 shows the $CH_4$ emissions according to EDGAR in the area, where our meas-

[Figure]

*Figure 1: The coloured contours show the position of the Pinto (south) and VTP (north) waste treatment areas. Left: Excluding landfills yielding emissions of ~2 t/h. Right: All CH4 emissions yielding ~6 t/h.*

urements were acquired. All categories are shown except for the 'solid waste landfills' (4A, 4B) sectors, as this is the one we are targeting in our analysis. The sum of the emissions is about 2 t/h (left) or about 15% of our total estimated emission rate. If these emissions are located within our plume region, they would add to the landfill emissions, but if they were only in the background region, they would actually reduce the landfill emissions, as a higher background would then be removed from the landfill plume

signal. Alternatively, if they were evenly distributed between the plume and the background area, then they would have a negligible effect. Please also read the comment above to "Line 125" and the additional sources mentioned according to the Spanish PRTR.

We have amended Sect. 3.4.2 accordingly with a short paragraph.

Furthermore, there is the possibility, that other CH4 sources located in the measurement area could affect the observed emission rates. Depending on whether these sources are in the plume area or the background area, they either contribute to the emissions or reduce them. If they were evenly distributed around the area, the effect on the estimated emission rate would be negligible. We estimated this effect by analysing the CH4 emissions reported in EDGAR v8.0 (Emission Database for Global Atmospheric Research, Crippa et al., 2023). We aggregated all CH4 emissions in our measurement area except for the source categories 4A and 4B (solid waste disposal and biological treatment of solid waste), which are our targets. Consequently, the resulting impact on our estimated emission rates could be approximately up to 2 t h$^{-1}$ at maximum or around 15 % of our total emission rate estimated according to EDGAR v8.0. Additionally, no other sources stand out in our column observations.

###############################################################################

*Line 448: This seems like a very important potential limitation of these methodologies, so it would be valuable to move the discussion of the wind changes into the body of the paper rather than the Appendix. A visual showing the potential location of these "puffs" in comparison to the mapped "plumes" would be helpful for understanding the impact this wind direction change could have on the calculated fluxes (this visual does not necessarily need to go in the body of the text though).*

Certainly, the mass balance approach is strictly applicable only for constant and persistent wind conditions. We are well aware of these limitations and are therefore currently carrying out model simulations to elucidate the evolution of the wind field. However, this is beyond the scope of this paper at hand and will be the subject of a subsequent paper that is currently in preparation. For this reason, we would like to leave this discussion to the forthcoming publication. In the manuscript at hand, we believe that we have at least mentioned all the possible reasons that could lead to deviations between "real" and "mass-balance derived" emissions rates.

###############################################################################

*Line 499: Why does the unpredictability of locations impact process-based modeling? In some cases, landfill operators have well documented information on which cells are being filled at different points in time. For the Madrid landfills, is it that this information is not being collected or just not publicly available to researchers?*

Our statement is not related to the process-based modelling or unpredictability of locations, but to the emissions which might be modulated by activities on the landfill. Usually, there is a lot of activity on weekdays, when waste is being added. In contrast, during weekends, less waste is shifted and less vehicles drive across the landfill, reducing the pressure on the subsurface waste and disfavour CH4 release due to compression. Thus, CH4 might be trapped more eaily and is less likely to reach the atmosphere across the active area of the landfill on weekends (or during night, if no work is

done). Information on which cells are active or inactive is available and also included in the paper (see Sect. 2.1.2).

###########################################################################

*Line 525: It seems inaccurate to say the two landfill sites were "well separated" when the downwind site flux could not be isolated.*

We meant that the separation of the landfills was sufficient to quantify the emissions from the upwind (Pinto) independently from the Las Dehesas/VTP area. In fact, because our measurements were performed in the least favorable conditions (with plume emitted by upwind source crossing over the downwind one), it is logical to assume that in the perpendicular wind situation would make independent estimation even easier. Thus, we believe that the areas are, in fact, well separated.

We have amended Line 525 accordingly.

The emissions from the two landfill sites were sufficiently separated for our methods by the two remote sensing instruments with an observed emission rate…

###########################################################################

*Line 546-548: This implies confidence in the ability of these instruments to identify sub-facility locations of sources. If this is a major conclusion of this manuscript, it does not seem like there was enough explanation of the methodology for identifying source locations and the associated uncertainty in those locations in the body of the paper.*

We have expanded the explanation of the source identification methodology and potential uncertainties and limitations as described in previous comments. (e.g., comment to "Fig. 5" and comment to "Line 354 - 357").

We have amended the statement taking the discussions above into account.

The determination of the exact source location is limited by a combination of the ground scene size of ∼ 110 x 110 m2, the accuracy of the orthorectification process itself being estimated to be better than 110 m, and modulation by local winds. The highest column enhancements and the 'start' of plumes, indicating the origin of the emissions, were observed over active parts of the landfills, ...

###########################################################################

*Section 5: Given the mention of the coming satellite in the introduction, it may be valuable to have more discussion on how the information gained from this study can inform future methods for the satellite.*

We have amended Sect. 5 as follows:

The methods used in this work are also applicable to planned satellite missions such as CO2M and MERLIN. Nevertheless, the generally coarser resolution on the ground will lead to a reduced sensitivity for emission rates, particularly for somewhat dispersed

sources. Also, the combination of active and passive remote sensing on a single satellite platform would show promise for the future, as the advantages of both methods can be synergistically exploited.

####################################################################

[revised manuscript text omitted]

 Active remote sensing instruments are independent of sunlight because they use a laser as their own source of electromagnetic radiation. In contrast to airborne and satellite-borne passive instruments, they can measure during day and night, across all seasons and latitudes. They provide ranging capabilities resulting from the precise measurement of the propagation time of the emitted light and, due to their narrow field of view, measure between clouds. IPDA (integrated-path differential absorption) lidars potentially provide highly accurate measurements without varying biases: the exceptions are those introduced by small differences in the scattering and reflectivity of the ground scene and any inaccuracies in the knowledge of the absorption cross-sections. However, Wolff et al. (2021) showed that under turbulent conditions, the spatial distribution of enhanced concentrations within an exhaust plume may be highly heterogeneous. As a result, a single overflight may sample sections with stronger or weaker enhancements purely as a result of the local variability. In some cases, the true emission signal only emerges after averaging over a high number of overflights.

To account for this potential limitation, in the analysis, we combine active lidar with passive imaging spectrometry, both designed to capture atmospheric $CH_4$ column gradients. Thus, we obtain both high-precision transects and spatial context, which supports a more robust interpretation of the observed $CH_4$ column enhancements. Moreover, these remote sensing measurements are complemented by auxiliary in situ measurements of $CH_4$, $CO_2$ and 3D winds in support of the remote sensing data. It was the first time that this payload was flown aboard the same aircraft acquiring spatially and

temporally collocated active and passive remote sensing measurements side-by-side for the acquisition of atmospheric $CH_4$ column observations. The greenhouse gas lidar CHARM-F ($CO_2$ and $CH_4$ Atmospheric Remote Monitoring Flugzeug) is an airborne demonstrator for the future satellite mission MERLIN  (MEthane Remote Sensing LIdar missioN, Ehret et al., 2017). The passive imaging MAMAP2DL (Methane Airborne Mapper 2D - Light) remote sensing instrument demonstrates the applicability of the $CH_4$ proxy retrieval (see below) at scales probed by CO2M (Copernicus Anthropogenic Carbon Dioxide Monitoring, Sierk et al., 2021) and TANGO (Twin Anthropogenic Greenhouse Gas Observer, SRON, 2024). The observations were collected in 2022 as part of the CoMet 2.0 (Carbon Dioxide and Methane) Arctic mission in Canada (CoMet, 2022). Prior to the transfer to Canada, an initial research flight was carried out to test all the instruments. This test flight was performed on the 4th August over Madrid to investigate the unexpected high landfill emission rates reported by Tu et al. (2022) and in a webstory from the European Space Agency (ESA) from October 2021 (ESA, 2021).

In Sect. 2, we provide a brief summary of the CoMet 2.0 mission and introduce the main instruments MAMAP2DL and CHARM-F used in this study (Sect. 2.1). This also includes a description of the algorithms used to infer $CH_4$ columns from the measurements (Sect. 2.2), additional steps necessary to achieve comparability between the passive and active observations (Sect. 2.3), and the cross-sectional flux method, which is used to quantify the $CH_4$ emissions (Sect. 2.4). Section 3 describes the observed $CH_4$ plumes over Madrid from both remote sensing instruments. This data is used to pin-point the exact source locations within the landfill area (Sect. 3.1), followed by a rigorous comparison of the active and passive data (Sect. 3.2), the resulting emission fluxes (Sect. 3.3), and a comprehensive discussion of potential uncertainties (Sect. 3.4). We close the paper by discussing our fluxes in a broader context (Sect. 4) and summarising our findings (Sect. 5).

**2  Methods and Data**

**2.1  Campaign and Instrumentation**

 Below we describe the waste treatment facilities

**2.1.1**

~~The analysed data was collected in the framework of the Carbon Dioxide and Methane Mission CoMet 2.0 Arctic, executed in Canada in 2022. The main objective of CoMet 2.0 was to investigate the influence of the contribution of major Arctic wetlands to the greenhouse gas budget in summer and how these compare to anthropogenic emissions from, e.g., fossil fuel exploitation and production sites or landfills. To achieve this goal, a comprehensive suite of instruments was~~ that were the targets of the research flights, and the flight strategy used to derive their methane emission rates. We then describe the instruments used, which were installed aboard the German research aircraft HALO (High Altitude and Long Range Research Aircraft, operated

by the DLR, Deutsches Zentrum für Luft- und Raumfahrt, type: Gulfstream G550).
2022,  The retrieval algorithms used to derive the CH$_4$  columns are explained next. This is followed by a description of the observed CH$_4$ ~~concentrations and of the basic data acquisition system of HALO, BAHAMAS (Basic Halo Measurement and Sensor System) including SHARC (Sophisticated Hygrometer for Atmospheric ResearCh) were used. The HALO basic data acquisition suite collects various atmospheric parameters such as the 3D wind field, temperature, humidity, and aircraft attitude data that is used for interpretation of the remote sensing data. The main instrumentation and the flight strategy are described in more detail below.~~ columns. Finally, we derive the methane fluxes, i.e. the methane emission rates, using the plume cross-sections.

**2.1.1 Target Description and Flight Strategy**

The targets under consideration were the Mancomunidad del Sur landfill in the municipality Pinto (40.264°N, 3.633°W; hereafter: Pinto landfill) and the Valdemingómez technology park (VTP, 40.332°N, 3.586°W) in the south-east of Madrid, Spain. The latter is a waste treatment complex  that accepts around 1,222 kt (Madrid, 2022) of waste, of which around 140 kt were deposited at the Las Dehesas landfill site in 2022 (Spanish-PRTR, 2025a), and houses several waste treatment facilities including the largest biomethane plant in Spain (Calero et al., 2023), which is also one of the largest in Europe (UABIO, 2022). Additionally, it contains landfill sites  such as the non-operating Valdemingómez landfill (40.331°N, 3.580°W) equipped with a gas recovery system and an active landfill site (40.325°N, 3.591°W; hereafter: Las Dehesas landfill) next to the waste treatment plant Las Dehesas. The northern half of the Las Dehesas landfill, where certain areas (i.e. cells) are already full and therefore closed, are also equipped with a gas recovering system (Sánchez et al., 2019). The two landfills in the technology park spread over an area of $\sim 0.9$ and $0.6\,\text{km}^2$ for the inactive Valdemingómez and the active Las Dehesas sites, respectively. More details about the different facilities are in the Annual Report for 2022 for the VTP (Madrid, 2022). The Pinto landfill, further to the south, stretches over $\sim 1.5\,\text{km}^2$. It opened in 1987, is still operational with around  53 kt of waste being dumped  in 2022 (Spanish-PRTR, 2025b), and the already closed parts of the landfills are equipped with gas recovering system (MdS, 2024).  The topography around these landfill sites shows some variability, ranging from about 550 to 700 m a.g.l. with a small valley between the two sites and a steep rise just south of the VTP according, to Google Earth.

According to the Spanish PRTR (Spanish-PRTR, 2025c), the combined annual reported 2022 CH$_4$ emissions for the two facilities "VERTRESA-URBASER, S.A. UTE

[Figure]

**Figure 1.** Top-view of the flight path of the HALO aircraft during the test flight over Madrid. An overview map in Google Earth of  the Iberian Peninsula is shown in **(a)** and Madrid is marked by the white cross.  Panel **(b)** shows a zoom-in of the Pinto landfill, outlined with a solid cyan line. Panel **(d)** displays the Valdemingómez Technology Park (VTP), marked with a solid magenta line. In the same panel, the closed Valdemingómez  landfill  is shown in pink and  open  Las Dehesas landfill is highlighted in purple. The flight path is shown in **(c)** whereby bluish colours represent the remote sensing (RS) part at $\sim 7.7$ km a.g.l. and greenish colours the in situ (IS) part at $\sim 1.6$ km a.g.l. (above ground level) of the flight. For better visualisation, the greenish in situ part is slightly shifted to the north-west because otherwise part of the legs would be hidden by RS legs. The map underneath is provided by Google Earth (Image © Landsat/Copernicus, Maxar Technologies).

(UTE LAS DEHESAS)"[3] and "DEPOSITO CONTROLADO DE RESIDUOS URBANOS DE PINTO"[4]

150 are $0.2$ t h$^{-1}$. Both sites are classified as "landfills" according to the European-Parliament (2006, Regulation (EC) 166/2006 E-PRT). We assume that these reported values are representative for the two  investigated areas, which include landfills and waste treatment plants, as  other listed sources would not contribute significantly to the emissions according to the Spanish PRTR (Spanish-PRTR, 2025c). According to the European-Commission (2006), both landfills appear not to use strict IPCC reporting methods. They report the methods 'OTH' (for other measurement

155 or calculation methodology) and 'C' (for calculation) using 'issue factors', and 'CRM' (for measurement methodology by means of certified reference materials) and 'M' (for measurement) using 'electrochemical cells' for the Las Dehesas and Pinto landfills, respectively, in 2022. In addition, the reporting method for the Pinto landfill changed from 2021 to 2022, while 'OTH' and 'C' using 'an American EPA (Environmental Protection Agency) calculation model' was applied in 2021 instead of CRM as in 2022.
* * *
[3]E-PRTRSectorCode / Name: 5 / Waste and wastewater managements, mainActivityCode: 5.(d), landfills, EU-Registry Code: 003510000, PRTR Code: 3510 (Spanish-PRTR, 2025a).

[4]E-PRTRSectorCode / Name: 5 / Waste and wastewater managements, mainActivityCode: 5.(d), landfills, EU-Registry Code: 001636000, PRTR Code: 1636 (Spanish-PRTR, 2025b).

160     To properly investigate emissions from these two landfills, dedicated flight patterns where the aircraft is levelled (so called flight legs) were aligned perpendicular to the forecasted wind direction (Fig. 1). The overflight time was between 13:00 and 15:40 local time (11:00 to 13:40 UTC) on the 4th August 2022. This time window was chosen using knowledge of the weather forecast predicting stable winds around noon, which also favoured the observations by the passive remote sensing instrument due to the high position of the sun.

165     Prevailing wind direction during the flight was from approximately SSW, aligned with the two waste treatment areas (Fig. D2 in the Appendix D1 shows time-resolved wind profiles for the time measured in the measurement area). 
[revised manuscript text omitted]

where $n$ is the number of cross-sections of one leg. The errors of the fluxes for one cross-section $F_{\mathrm{M2D,\,cs}}$ and of the average flux of one leg $F_{\mathrm{M2D,\,leg}}$ are then computed by error propagation using the errors on the individual parameters used in Eq. 2, as

360 explained in Appendix F.

  However, the approach above does not allow for a 1-to-1 comparison of fluxes between those determined form the measurements made by the imaging MAMAP2DL and those by the 1D CHARM-F instruments. Both datasets are actually distorted by the aircraft movement (i.e. predominantly the aircraft roll). The straight cross-sections introduced above for MAMAP2DL do not follow the distorted CHARM-F ground track. However, as seen in Fig. 4 (a), the CHARM-F ground track follows one

365 fixed  MAMAP2DL viewing angle approximately in the middle of the swath because the effect of the distortion is the same for both instruments. Therefore, Eq. 1 is directly applied to the measurements, with each parameter being evaluated individually for one measurement i.e. the wind speed, direction and length segment are not constant anymore. The resulting CHARM-F flux is then representative for this one leg. Definition of plume and background areas remain the same.

Independent of the applied approach for MAMAP2DL or CHARM-F data described above, the fluxes from several legs, 370 computed by Eq. 3, are then again averaged to derive the mean emission rates $F_{\text{M2D, ar-aver}}$ and $F_{\text{CHARM-F, ar-aver}}$[7] of certain areas in the measurement area for the respective instrument:

$$F_{\text{M2D or CHARM-F, ar-aver}} = \frac{\sum_i F_{\text{M2D or CHARM-F, leg}_i}}{p} \frac{\sum_{i=1}^{p} F_{\text{M2D or CHARM-F, leg}_i}}{p}, \tag{4}$$

where $p$ is the number of legs. This applies, for example, to the area in the lee of the two waste treatment areas, which is representative of the total emissions from the measurement area.

**3 Results**

**3.1 Observed Column Enhancements over Madrid and Source Attribution**

Figure 5 visualises the retrieved and orthorectified $CH_4$ column anomaly maps derived from the MAMAP2DL measurements (a, as described in Sect. 2.2.1) and the $XCH_4$, given as 3 seconds averages, derived from CHARM-F data (d, as described in Sect. 2.2.2) for the different remote sensing legs acquired at a flight altitude of $\sim 7.7\,\text{km a.g.l.}$. Both data sets clearly show 380 $CH_4$ enhancements (in red) located at or downwind of the waste treatment areas, whereas upwind or south-west of the Pinto landfill in the bottom left corner, there are no indications of inflow of external enhanced $CH_4$ in the measurement area. These observations are also confirmed by two legs flown in along wind direction at two different flight altitudes (see Appendix A and Fig. A1 for details). Especially for the Pinto landfill, there is a clear plume visible in both overflights, $\sim 2.5\,\text{h}$ apart from each other.

385 The highest $CH_4$ concentrations are observed at or close to landfills. $CH_4$ hot spots, with peak enhancements of around $17\,\%$, are located at the eastern part of the Pinto landfill according to the MAMAP2DL imaging data. This hot spot is also captured by the CHARM-F instrument with $XCH_4$ of up to $2.28\,\text{ppm}$.

The insets (b) and (c) show more details of the individual landfills including the locations of the highest column anomalies, which were identified in different overflights. Marked regions in the southeast of the landfills are areas which are, most prob-390 ably, responsible for a large fraction of the observed emissions. They were selected by analysing the flight legs (those flown perpendicular to the wind direction as well as those flown in wind direction) over the landfills to find the ground scenes with the highest column enhancements (only ground scenes above a certain threshold are shown, see Fig. 5 for more details). The assumption is that the $CH_4$ is most concentrated just above or very close to a source, as it is not yet dispersed (horizontally),
* * *
[7]'ar-aver' stands for areal averages.

[Figure]

**Figure 5.** Retrieval results from the airborne remote sensing instruments. **(a)** and **(d)** show the retrieved $CH_4$ column anomalies from MAMAP2DL and the $XCH_4$ from CHARM-F, respectively. **(b)** and **(c)** are zoomed pictures of the two landfills including the MAMAP2DL  ground scenes in red with the largest anomalies only (only those larger than ~$4\%$ for VTP or Las Dehesas in **(b)**,  and ~$8\%$ for Pinto in **(c)**, see main text for details). The different colours of the borders around those ground scenes in **(b)** and **(c)** mark the enhancements observed in different flight legs. E.g., yellow represents the leg flown in along wind direction shown in Fig. A1, a. The small insets in **(b)** and **(c)** zoom in further detailing some activities across the areas with the largest observed enhancements. The shown Google Imagery was recorded in August 2022. The  waste treatment areas are encircled by different coloured solid lines: Cyan for the Pinto landfill;  magenta for the VTP;  pink for the Valdemingómez landfill; purple for the Las Dehesas landfill. The map underneath is provided by Google Earth (Image © Landsat/Copernicus, Maxar Technologies).

leading to the highest observed column enhancements. However, there is a residual uncertainty associated with this method,
395  as by chance, high column enhancements could also be observed further away due to turbulent transport. Columns are also modulated by the prevailing wind speed at the time of release into the atmosphere and by local atmospheric turbulence. Both can change during a measurement flight. However, if the highest column anomalies during multiple over-flights point to the same region, there is a high probability that this region is acting as a source.

The Google Earth imagery recorded in August 2022 (the same month and year as our flight) clearly shows that these hot
400  spots are directly located over active landfill areas where waste is deposited. The $CH_4$ plumes clearly begin over these areas of the Pinto and Las Dehesas landfills (see Figs. 5 and A1 and discussion on the step-wise increase of the fluxes in Sect. 3.3). However, we cannot exclude that other parts,  closed cells of the landfills or facilities located in these waste treatment areas, also contribute (weakly) to the observed $CH_4$ plume, but are partly masked by $CH_4$ released

further upwind. For example, in the north western part of the VTP, there are also hot spots identified (two ground scenes outlined in yellow in Fig. 5, b) from the along wind leg (Fig. A1, a), which could be advected there or released from the waste treatment plants (PLANT OF BIOMEIZATION La Paloma and Las Dehesa) immediately to the south. However, as a second overflight (Fig. A1, b) shows no enhancements, it is probably the former.

**3.2 Column Comparison between MAMAP2DL and CHARM-F**

Figure 5 shows a good visual agreement between the column anomalies of the passive MAMAP2DL and the $X\mathrm{CH_4}$ of the active CHARM-F instruments. In order to perform a more rigorous comparison between the two types of atmospheric $\mathrm{CH_4}$ columns, we convert the $X\mathrm{CH_4}$ partial columns derived from CHARM-F to total column anomalies (see Sect. 2.3). We then identify the ground scene in the MAMAP2DL swath which corresponds to the CHARM-F measurements, which are approximately located in the middle of the MAMAP2DL swath. This procedure ensures the selection of observations, where both instruments see similar ground scenes and air masses.

Figure 6 shows a typical example comparison for one leg. The two different types of observations have been processed as explained previously, i.e the plume anomalies have been processed as described in Sect. 2.3 and the $\mathrm{CH_4}$ fluxes have been estimated as described in Sect. 2.4. The shown background normalised column anomalies agree well within their respective errors inside and outside of the plume. Even more pronounced structures in the $\mathrm{CH_4}$ concentration, as encountered on the right hand side ($\sim 6$ to $15\,\mathrm{km}$ distance), are identified by both instruments. The fluxes from the two shown cross-sections deviate by only $0.1\,\mathrm{t\,h^{-1}}$ or $1\,\%$.

More generally, when comparing fluxes estimated using measurements of MAMAP2DL with those derived from CHARM-F observations from all flight legs (see Fig. E1), the averaged absolute difference between them is $\sim 1.2\,\mathrm{t\,h^{-1}}$ or $\sim 13\,\%$ excluding the flight legs upwind and directly over the Pinto landfill (see Sect. 3.3 for reasoning).  These differences may be due to (a) different but overlapping opening angles of the two instruments and the resultant spatial resolution of the ground scene widths of $110\,\mathrm{m}$  and $10\,\mathrm{m}$, respectively, or (b) different paths through the atmosphere of the electromagnetic radiation used to measure methane absorption, or (c) differences in the algorithms used to retrieve the columns, e.g. how they deal with variable surface reflectivity etc. Consequently, observed air masses are different.

Typically, the errors of the fluxes are around or below $30\,\%$ of the respective flux and are similar for MAMAP2DL and CHARM-F.

**3.3 Derived Landfill Emission Rates**

Using imaging MAMAP2DL observations, we also computed the fluxes within the different legs (see Sect. 2.4). The results are summarised in Fig. 7, which also includes the cross-sectional fluxes derived from the CHARM-F instrument already computed and introduced in Sect. 3.2.

Based on the MAMAP2DL observations, the fluxes exhibit a step-wise increase at the location of landfills as expected (from left - upwind, to right - downwind). The upwind leg at $-5\,\mathrm{km}$ shows no significant inflow of enhanced $\mathrm{CH_4}$ and a steep increase

[Figure]

**Figure 6.** The background normalised $CH_4$ column anomalies for CHARM-F (orange) and the co-located MAMAP2DL (blue) observations for one flight leg collected between 11:42 and 11:46 UTC are shown. Vertical dotted lines separate the plume and background areas. Shaded areas represent the random error (single measurement precision) of the retrieved column anomalies of the respective instrument. The computed fluxes for the cross-sections according to Eq. 1 and the corresponding errors (MAMAP2DL: Eq. F1, and CHARM-F: Eq. F10) are given by the text insets. For graphical presentation only, the MAMAP2DL data has been smoothed by a $500\,\mathrm{m}$ kernel to match the spatial resolution of CHARM-F in along flight direction. Flux, error and uncertainty range is, however, based on the $\sim 110 \times 110\,\mathrm{m}^2$ data.

directly over the Pinto landfill. Between the Pinto and the VTP the flux or emission rate stabilises at $4.2\,\mathrm{t\,h}^{-1}$ ($\pm 38\,\%$) before increasing to around $12.1\,\mathrm{t\,h}^{-1}$ ($\pm 27\,\%$) on average at and after the Las Dehesas landfill. However, the cross-sectional fluxes show some variability from flight leg to flight leg (see the bold horizontal coloured lines, representing averaged values over one MAMAP2DL leg) and variability within one leg (see the thin solid coloured lines). Furthermore, the retrieved column anomalies in Fig. 5 as well as the cross-sectional fluxes in Fig. 7 show no sign of accumulation of $CH_4$ as, for example, in the valley between the two landfills (see Sect. 2.1.1 for a brief discussion of the local topography). Adding the fluxes derived from the CHARM-F observations to the figure (coloured stars) reveals very good agreement between active and passive remote sensing (thin solid coloured lines) data as already indicated in Sect. 3.2. Computing average fluxes or emission rates from the CHARM-F observations alone yields $5.2\,\mathrm{t\,h}^{-1}$ ($\pm 37\,\%$) for the Pinto landfill and $13.3\,\mathrm{t\,h}^{-1}$ ($\pm 26\,\%$) for both waste treatment areas combined.

The first two flight legs were deliberately designed so that, in the blue leg, CHARM-F sampled background conditions upwind of the Pinto landfill, while MAMAP2DL already partially covered the source area. Conversely, during the green leg, CHARM-F measured directly over the Pinto landfill, whereas the MAMAP2DL swath still included parts of the upwind background. For the averaged flux between the two landfills, the flight leg directly over the Pinto landfill (i.e. the green lines and star in Fig. 7) has been omitted. There, the plume might be still restricted to the surface and the wind speed is highly biased due to the strong vertical wind gradient (see Fig. D2, a). Over the Las Dehesas landfill, although there are new emissions

[revised manuscript text omitted]

and their resulting flux is $\sim 3.5\,\text{t}\,\text{h}^{-1}$. This flux was assigned to the Valdemingómez waste plant. Satellite data were analysed over the period May 2018 to December 2020. Estimated emission rates are $7.1\,\text{t}\,\text{h}^{-1}$ ($\pm 0.6\,\text{t}\,\text{h}^{-1}$) for the entire area.

A ground-based investigation in that area was undertaken from the 1$^{\text{st}}$ to 3$^{\text{rd}}$ March in 2016. Sánchez et al. (2019) used specifically designed flux chambers to measure $CH_4$ emission from the already full and closed parts (or cells) of the Las Dehesas landfill north of the still active area. They have estimated $1.1\,\text{t}\,\text{h}^{-1}$ on average for this part which accounts for approximately half of the total designated landfill area of $\sim 0.6\,\text{km}^2$. The values for the $95\,\%$ confidence interval are given with 0.4 to $2.8\,\text{t}\,\text{h}^{-1}$. Their averaged value would correspond to around $9\,\%$ of our total emission rate, however, derived for the entire area also including the Pinto landfill.

Over the past years, all these estimates indicate consistently high emission rates of up to 7 to $9\,\text{t}\,\text{h}^{-1}$ for both waste treatment areas, although they are made over short periods (with the exception of the estimates using satellite observations[8]). Our estimated emission rate for the two areas are at the upper end of this range ($12.7\,\text{t}\,\text{h}^{-1}$ or $9.3\,\text{t}\,\text{h}^{-1}$ if the $CH_4$ puff hypothesis is applicable) and also indicates disagreement with the reported values in E-PRTR (see Table 3). Interestingly, the reported emissions of the Pinto landfill site decreased by a factor of almost 40 from 2021 to 2022, which could potentially be related to the change in reporting methodology (see Sect. 2.1.1).

[revised manuscript text omitted]

$S_{\text{on}}$, $S_{\text{off}}$ are backscattered  signals and $E_{\text{on}}$, $E_{\text{off}}$ are internal energy reference  measurements at the on- and offline wavelength, respectively. $\Delta\tau$ can on the other hand also be described in terms of the molecular absorption cross-section at both wavelengths ($\sigma_{\text{on}}$ and $\sigma_{\text{off}}$) and the number density $n_{\text{CH}_4}$ or the dry-air mixing ratio of methane $r_{\text{CH}_4}$.

$$\Delta\tau = \int_{h_0}^{h_1} n_{\text{CH}_4} \underbrace{(\sigma_{\text{on}} - \sigma_{\text{off}})}_{=\Delta\sigma} \, dh, \tag{C2}$$

$$= \int_{h_0}^{h_1} \frac{r_{\text{CH}_4}}{(1 + r_{\text{H}_2\text{O}})} \cdot n_{\text{air}} \cdot \Delta\sigma \, dh. \tag{C3}$$

The integral runs over the air column between aircraft ($h_1$) and ground ($h_0$). In Eq. C3 the number density of greenhouse-gas molecules has been expressed in terms of the number density of air molecules $n_{\text{air}}$ and the dry-air molar mixing ratio of the greenhouse-gas species, while also accounting for the dry-air molar mixing ratio of water

$$n_{\text{air}} = (1 + r_{\text{H}_2\text{O}}) \cdot n_{\text{dry air}}. \tag{C4}$$

Using the general gas equation ($n_{\text{air}} = \frac{p}{k_{\text{B}} \cdot T}$) $n_{\text{air}}$ can be expressed in terms of pressure $p$ and temperature $T$. Furthermore, the integral over altitude can be transformed into a pressure integral ($dh = \frac{dh}{dp} \cdot dp = -\frac{k_{\text{B}} \cdot N_{\text{A}} \cdot T}{p \cdot M_{\text{air}} \cdot g} \, dp$).

$$\Delta\tau = \int_{p_1}^{p_0} \frac{r_{\text{CH}_4} \cdot N_{\text{A}}}{(1 + r_{\text{H}_2\text{O}}) \cdot M_{\text{air}} \cdot g} \cdot \Delta\sigma \, dp. \tag{C5}$$

Here, $k_{\text{B}}$ is the Boltzmann constant, $N_{\text{A}}$ the Avogadro constant, $g$ the gravitational acceleration and $M_{\text{air}}$ the average molar mass of air. Next, molar mass is expressed as molecular mass ($m_{\text{air}} = \frac{M_{\text{air}}}{N_{\text{A}}}$) and dry air is discriminated from water vapour.

$$\Delta\tau = \int_{p_1}^{p_0} \frac{r_{\text{CH}_4}}{(1 + r_{\text{H}_2\text{O}})} \cdot \frac{\Delta\sigma}{\underbrace{m_{\text{air}}}_{=\frac{m_{\text{dry air}} + r_{\text{H}_2\text{O}} \, m_{\text{H}_2\text{O}}}{1 + r_{\text{H}_2\text{O}}}}} \cdot g \, dp, \tag{C6}$$

$$= \int_{p_1}^{p_0} \frac{r_{\text{CH}_4}}{(m_{\text{dry air}} + r_{\text{H}_2\text{O}} \cdot m_{\text{H}_2\text{O}})} \cdot \frac{\Delta\sigma}{g} \, dp. \tag{C7}$$

Finally, $r_{CH_4}$ is pulled out of the integral by replacing it with a (weighted) column average $XCH_4$, which is thus defined.

$$\Delta\tau = XCH_4 \int_{p_1}^{p_0} \frac{\sigma_{\mathrm{on}}(p,T) - \sigma_{\mathrm{off}}(p,T)}{g\left(m_{\mathrm{dry\ air}} + r_{H_2O} \cdot m_{H_2O}\right)}\ \mathrm{d}p, \tag{C8}$$

[revised manuscript text omitted]